# Zero-Shot Visual Generalization in Model-Based Reinforcement Learning via Latent Consistency

## Abstract

Model-based reinforcement learning (MBRL) has shown remarkable success in pixel-based control by planning within learned latent dynamics. However, its robustness degrades significantly when test-time observations deviate from the training distribution due to unseen distractions such as shadows, viewpoint, or background variations. In this paper, we propose **Vi**sual **G**eneralization in **MO**del-based RL (**ViGMO**), a novel framework that achieves zero-shot generalization to unseen visual distractions while preserving high sample efficiency. ViGMO integrates three key components: (i) a *mixed weak-to-strong augmentation* strategy to balance efficient learning with robustness, (ii) *latent-consistency learning* to enforce stable transition predictions under distribution shifts, and (iii) *encoder regularization* to preserve task-relevant features and prevent representational collapses. Extensive evaluations on the DeepMind Control suite and Robosuite with challenging unseen distractions demonstrate that ViGMO outperforms state-of-the-art model-free and model-based baselines, improving zero-shot generalization by up to $13\%$ over the strongest baseline while maintaining the hallmark efficiency of latent-space MBRL.

## 1 Introduction

Visual reinforcement learning (visual RL) enables the training of control policies directly from raw pixel observations, eliminating the need for task-specific state estimation within hand-engineered perception pipelines (Laskin et al., 2020b; Yarats et al., 2021a). This end-to-end approach simplifies a deployment to real-world applications and has demonstrated impressive success in domains such as locomotion (Schulman et al., 2017; Haarnoja et al., 2018; Fujimoto et al., 2018) and manipulation (Kalashnikov et al., 2018; Nair et al., 2018; Hansen et al., 2024). However, visual RL faces a fundamental limitation: the visual input channel is highly sensitive to *unseen* visual distractions. Even minor shifts in shadows, viewpoints, or background textures can push test-time observations outside the training image distribution, leading to substantial policy degradation.

Addressing this vulnerability requires *visual generalization*, wherein an agent must reliably handle task-irrelevant distractions in the visual input. Recent model-free RL (MFRL) approaches have proposed various strategies, including learning robust visual representations (Yuan et al., 2022; Nair et al., 2023; Wang et al., 2023; Yang et al., 2024), applying data augmentations (Hansen et al., 2021; Yarats et al., 2022; Huang et al., 2022), or stabilizing value function learning (Hansen et al., 2021; Liu et al., 2023; Huang et al., 2024). While these methods have shown promising results, they typically rely on incremental policy updates and consequently suffer from limited sample efficiency (Botvinick et al., 2019). Moreover, challenges are further exacerbated under conditions of broken randomness (Xu et al., 2024) and high-dimensional state–action spaces (Yarats et al., 2021b).

By contrast, recent advances in model-based RL (MBRL) achieve superior sample efficiency by planning in a learned latent dynamics space (Hansen et al., 2024; Hafner et al., 2025). However, achieving visual generalization in MBRL is markedly more challenging: an MBRL agent must not only encode observations invariantly, but also ensure that its *latent dynamics* model—the backbone of planning—produces accurate and reliable rollouts under visual perturbations. Any perturbation-induced drift in latent representations can corrupt entire synthetic trajectories, leading to catastrophic decision errors. Consequently, overlooking these factors diminishes predictive accuracy and undermines the inherent sample efficiency advantages of MBRL. Figure 1 illustrates that directly deploying a standard MBRL agent under unseen visual distractions can indeed result in substantial failures.

Figure 1: **Challenges of unseen visual distractions in MBRL.** During training (top), clean observations are encoded into latents $z_t \in \mathcal{Z}^{\text{train}}$ and the dynamics model $d_\theta$ correctly predicts $z_{t+1} \in \mathcal{Z}^{\text{train}}$ where $\mathcal{Z}^{\text{train}} \subset \mathcal{Z}$. At test time (bottom), observations with unseen, task-irrelevant distractions (red) yield latents $z'_t \notin \mathcal{Z}^{\text{train}}$, forcing $d_\theta$ to extrapolate and producing erroneous rollouts $z'_{t+1}$ that deviate from the expected dynamics on $\mathcal{Z}^{\text{train}}$. ViGMO alleviates this issue by aligning out-of-distribution transitions $(z'_t, z'_{t+1})$ with their in-distribution counterparts $(z_t, z_{t+1})$, thereby preserving consistent latent rollouts and enabling robust generalization.

In this paper, we propose **Vi**sual **G**eneralization in **MO**del-based RL (ViGMO), a novel MBRL framework that enables zero-shot visual generalization to unseen distractions while maintaining high sample efficiency. ViGMO incorporates three core components: (i) a mixed weak-to-strong augmentation (MA) strategy, which maintains a balance of soft and hard perturbations throughout training, enabling sample-efficient learning while enhancing robustness to diverse distractions; (ii) latent-consistency learning (LC), which builds upon MA by enforcing the latent dynamics model to produce stable next-state predictions across MA-generated views of a trajectory segment, thereby improving stability under distribution shifts; and (iii) encoder regularization (ER), which constrains the visual encoder to preserve task-relevant features across augmentations and prevent representational collapses, ensuring reliable representations throughout training.

We comprehensively evaluate ViGMO on challenging visual control tasks in the DeepMind Control (DMC) suite (Tassa et al., 2018) and Robosuite (Zhu et al., 2020), following rigorous zero-shot generalization protocols (Yuan et al., 2024). After training solely on clean environments, ViGMO is tested under settings with unseen and complex distractions—including object-, and background-color shifts, natural video backgrounds, and scene texture changes. In this challenging regime, ViGMO surpasses state-of-the-art MFRL and MBRL baselines: it improves zero-shot generalization by up to $13\%$ over the strongest baseline while maintaining the hallmark sample efficiency of latent-space MBRL. Ablation studies further confirm that the individual components of ViGMO, as well as its overall design, are essential for achieving robust generalization.

The key contributions of this study are summarized as follows:

- We propose ViGMO, a novel MBRL framework that enables zero-shot generalization to unseen visual distractions while maintaining high sample efficiency. This is achieved by effectively integrating MA, LC, and ER into the training pipeline of standard world models.
- Through extensive experiments on the DMC suite and Robosuite, we show that ViGMO achieves a superior efficiency–generalization trade-off, improving zero-shot generalization by up to $13\%$ over the strongest baseline while preserving high sample efficiency.

## 2 RELATED WORK

### 2.1 VISUAL MODEL-BASED REINFORCEMENT LEARNING

Latent-space MBRL improves sample efficiency by learning a compact world model that supports planning directly in latent space (Ha & Schmidhuber, 2018; Hafner et al., 2019; 2020; 2025; Hansen et al., 2022; 2024; Zhao et al., 2023). The Dreamer series adopts a *decoder-based* world model that reconstructs pixel observations alongside learning latent dynamics and rewards (Hafner et al., 2020; 2021; 2025), whereas the TD-MPC series uses a *decoder-free* world model that discards the reconstruction and performs planning directly in the latent space (Hansen et al., 2022; 2024). A shared limitation of these approaches is the *i.i.d. assumption*: a test image set is expected to follow the same distribution as the training image set. In practice, novel backgrounds, shadows, or textures readily violate this assumption, driving latent representations off the training manifold and ultimately causing rollout failures (Figure 1). ViGMO addresses this vulnerability by equipping

latent-space MBRL with three key components—MA, LC, and ER—in a manner compatible with *any* latent-space MBRL backbone. Further discussions on visual MBRL are provided in Appendix F.

## 2.2 VISUAL GENERALIZATION IN MODEL-BASED REINFORCEMENT LEARNING

Visual distractions are particularly harmful in MBRL. They not only distort encoder outputs but also destabilize downstream dynamics, leading to compounding errors in long-horizon planning.

One line of work, often referred to as *invariant MBRL*, aims to learn representations that discard task-irrelevant visual factors while preserving task-relevant dynamics. Representative approaches (Zhang et al., 2021; Wang et al., 2022; Zhu et al., 2023; Zhou et al., 2025) encourage encoders to ignore nuisance variables such as textures or colors, thereby improving robustness under distribution shifts. However, most of these methods rely on *test-time adaptation*, requiring additional interactions in the target environment to update the encoder before reliable deployment. By contrast, ViGMO targets the stricter setting of *zero-shot generalization*, where no adaptation data are available at test time and robustness must hold immediately upon deployment. A complementary direction is Dr. G (Ha et al., 2023), which extends Dreamer (Hafner et al., 2021) with dual contrastive objectives and an inverse-dynamics loss. Unlike invariant MBRL methods that focus on representation learning with adaptation, Dr. G explicitly targets *zero-shot generalization*, improving robustness against unseen distractions. However, this improvement comes at the expense of additional data and reduced sample efficiency. Building upon TD-MPC2 (Hansen et al., 2024), ViGMO advances this line of work by integrating three key components—MA, LC, and ER—which jointly enable strong zero-shot generalization to unseen distractions while preserving the hallmark sample efficiency of latent-space MBRL, without requiring contrastive negatives or inverse-dynamics losses.

## 3 VIGMO

In this section, we introduce ViGMO, a framework for zero-shot MBRL that achieves strong visual generalization to unseen distractions while retaining high sample efficiency. ViGMO enhances a standard latent-space world model with three main components: (i) **MA**, which balances efficient training with robustness by exposing the agent to both soft and hard perturbations via a structured augmentation strategy; (ii) **LC**, which builds upon MA by enforcing consistent transition predictions across the MA-generated views of a trajectory segment, thereby stabilizing latent rollouts under distribution shifts; and (iii) **ER**, which preserves task-relevant features and prevents representational collapses. Because these components operate solely through the input pipeline and auxiliary loss terms, ViGMO can be seamlessly integrated into *any* latent-space MBRL algorithm that employs a visual encoder and latent dynamics model, without modifying its planner or optimization procedures. An overview of ViGMO is illustrated in Figure 2.

### 3.1 MIXED WEAK-TO-STRONG AUGMENTATION STRATEGY

We distinguish between *weak* augmentations—minor, task-irrelevant perturbations that preserve most visual information—and *strong* augmentations that impose substantial, task-relevant visual changes. Weak augmentations promote stable and sample-efficient training, whereas strong augmentations enhance robustness by exposing the agent to diverse distractions. To combine these complementary benefits, we adopt *random-shift* (Yarats et al., 2022) as the weak transform $\tau^w$ and *random-overlay* (Hansen & Wang, 2021) as the strong transform $\tau^s$.

Our MA strategy proceeds as follows: weak augmentations are applied to the entire mini-batch, and strong augmentations are further applied to a subset of these weakly augmented samples. This produces a mixture of weak-only and weak-to-strong samples, which serve as the foundation for LC. Importantly, this design avoids the computational overhead of encoding two views per sample: instead of doubling the batch size, MA achieves robustness by partitioning the batch, thereby retaining the efficiency of standard training pipelines. Its detailed discussion is provided in Appendix D.3.

**Batch split.** Given a mini-batch $\mathcal{B}$ with index set $\mathcal{I} = \{1, \ldots, |B|\}$, we first apply the weak transform to all samples and then split the batch uniformly at random into two complementary sub-batches, $B^w$ and $B^{ws}$, indexed by

$$\mathcal{I}^{ws} = \mathcal{I} \setminus \mathcal{I}^w, \qquad \zeta = |\mathcal{I}^w|/|\mathcal{I}^{ws}|.$$

Here, $B^w$ (with indices $\mathcal{I}^w$) contains weak-only samples, while $B^{ws}$ (with indices $\mathcal{I}^{ws}$) contains weak-to-strong samples. In our experiments, we set $\zeta = 1$ (half weak-only, half weak-to-strong).

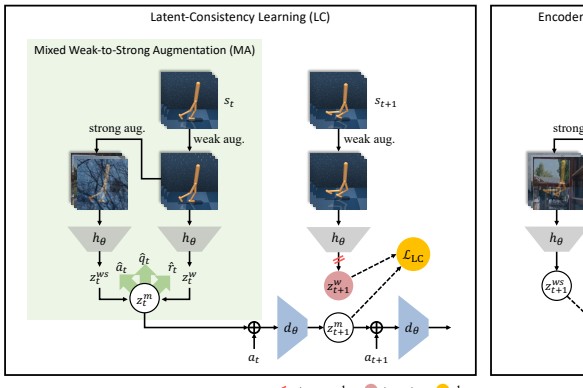

Figure 2: **An overview of ViGMO. Left:** Within LC, MA generates complementary weak-only and weak-to-strong views of the same frame, which are then encoded as $z_t^w$ and $z_t^{ws}$ and combined into a mixed latent representation $z_t^m = z_t^w \oplus z_t^{ws}$. The dynamics model is trained to align its prediction from this mixed latent with the weakly augmented target $z_{t+1}^w$, ensuring stable latent rollouts under distractions. **Right:** ER constrains the encoder by aligning weak-only and weak-to-strong encodings of the same frame, preserving task-relevant features while discarding nuisance factors.

**Augmented representations.** At the first time step of each trajectory segment with horizon $H$, we construct weak-only and weak-to-strong image stacks,

$$s_t^w = \tau^w\big(\{s_{t,i}\}_{i \in \mathcal{I}^w}, \upsilon^w\big), \qquad s_t^{ws} = \tau^s\big(\tau^w\big(\{s_{t,j}\}_{j \in \mathcal{I}^{ws}}, \upsilon^w\big), \upsilon^s\big),$$

which are then encoded by the shared encoder $h_\theta$:

$$z_t^w = h_\theta(s_t^w), \qquad z_t^{ws} = h_\theta(s_t^{ws}).$$

Here, $\upsilon^w \sim \Upsilon^w$ and $\upsilon^s \sim \Upsilon^s$ parameterize the weak and strong augmentation functions, and $\{s_{t,i}\}_{i \in \mathcal{I}^w}$ and $\{s_{t,j}\}_{j \in \mathcal{I}^{ws}}$ are disjoint subsets of mini-batch frames assigned to weak-only or weak-to-strong perturbations. The resulting latents are concatenated along the batch dimension to form the mixed representation

$$z_t^m = z_t^w \oplus z_t^{ws},$$

which serves as the input to the world model and is recursively rolled forward by the latent dynamics model over the prediction horizon.

## 3.2 LATENT-CONSISTENCY LEARNING

Building upon the MA strategy, we introduce the LC loss. The key idea is that task-relevant dynamics should remain stable under unseen visual distractions (Figure 1); thus, the world model should produce consistent next-state predictions across the MA-generated views of a trajectory segment. The LC loss enforces this invariance in the latent space.

Formally, after encoding an observation $s_t$ into a latent $z_t = h_\theta(s_t)$, the dynamics model $d_\theta$ predicts the successor latent $\hat{z}_{t+1} = d_\theta(z_t, a_t)$ given action $a_t$. Under the standard objective, $d_\theta$ is trained by regressing its prediction toward the latent of the true next observation, $z_{t+1} = h_\theta(s_{t+1})$, via a mean-squared-error loss:

$$\mathcal{L}_{\text{dyn}}(\theta) = \text{MSE}\big(\hat{z}_{t+1}, \text{sg}(z_{t+1})\big) = \big\|d_\theta(z_t, a_t) - \text{sg}(h_\theta(s_{t+1}))\big\|_2^2,$$

where $\text{sg}(\cdot)$ denotes the stop-gradient operator, preventing gradients from flowing into the target encoder path.

For the LC loss, the dynamics model is conditioned on the mixed latent representation $z_t^m = z_t^w \oplus z_t^{ws}$ generated by MA. As a stable training signal, LC employs the weakly augmented successor $s_{t+1}^w = \tau^w(s_{t+1}, \upsilon^w)$, encoded as $z_{t+1}^w = h_\theta(s_{t+1}^w)$, as the target. The dynamics model is trained to align its prediction $\hat{z}_{t+1}^m = d_\theta(z_t^m, a_t)$ with this weak target:

$$\mathcal{L}_{\text{LC}}(\theta) = \text{MSE}\big(\hat{z}_{t+1}^m, \text{sg}(z_{t+1}^w)\big) = \big\|d_\theta(z_t^m, a_t) - \text{sg}(h_\theta(s_{t+1}^w))\big\|_2^2.$$

Because the weak target $z_{t+1}^w$ is nearly noise-free, the model receives a stable supervisory signal, yet it must learn to map the next-state prediction, derived from the perturbed mixed input ($z_t^m$

constructed via MA), onto that target. In doing so, the LC loss enforces *augmentation-invariant* latent dynamics, thereby improving out-of-distribution (OOD) generalization while also enhancing rollout stability and long-horizon consistency—critical properties for effective planning in unseen environments.

### 3.3 Encoder Regularization

While the LC loss enforces invariance at the dynamics level, the encoder itself may still produce inconsistent features across augmentations. If left unconstrained, this can yield unstable latents under unseen distractions, corrupting the entire rollout and undermining the planning process. To address this, we introduce an auxiliary ER loss, inspired by SODA (Hansen & Wang, 2021).

At each time step of a trajectory segment, we generate a weak-only augmented view $s_t^w$ and a weak-to-strong augmented view $s_t^{ws}$ from the same image stack $s_t$. Passing these through the shared encoder yields $z_t^w = h_\theta(s_t^w)$ and $z_t^{ws} = h_\theta(s_t^{ws})$. The encoder is then trained by minimizing the $\ell_2$ distance between their $\ell_2$-normalized representations:

$$\mathcal{L}_{\text{ER}}(\theta) = \left\| \frac{z_t^{ws}}{\|z_t^{ws}\|_2} - \frac{\text{sg}(z_t^w)}{\|\text{sg}(z_t^w)\|_2} \right\|_2^2.$$

This objective explicitly aligns weak-only and weak-to-strong encodings of the same frame, ensuring that the encoder captures task-relevant features rather than augmentation-specific artifacts. Together with the LC loss, the ER loss stabilizes representation learning and produces more reliable rollouts, leading to stronger generalization under unseen visual distractions.

### 3.4 Overall Training Objective

Building upon the standard world model (here, TD-MPC2 (Hansen et al., 2024)), ViGMO augments the training objective with two auxiliary terms: the LC loss $\mathcal{L}_{\text{LC}}$, which enforces augmentation-invariant latent dynamics, and the ER loss $\mathcal{L}_{\text{ER}}$, which stabilizes encoder representations. For a horizontal trajectory segment $\Gamma = (s_t, a_t, r_t, s_{t+1})_{t:t+H}$ sampled from the replay buffer $\mathcal{B}$, the overall objective is defined as:

$$\mathcal{L}_{\text{total}}(\theta) = \mathbb{E}_{\Gamma \sim \mathcal{B}} \left[ \mathbb{E}_{v^w \sim \Upsilon^w, v^s \sim \Upsilon^s} \left[ \sum_{i=t}^{t+H} \lambda^{i-t} \underbrace{\mathcal{L}_{\text{TD-MPC2}}(\theta; \mathcal{L}_{\text{rew}}, \mathcal{L}_Q, \mathcal{L}_{\text{LC}}, \Gamma_i)}_{\text{world-model losses}} + \underbrace{\mathcal{L}_{\text{ER}}(\theta; v^w, v^s)}_{} \right] \right],$$

$$\underbrace{\phantom{\mathcal{L}_{\text{total}}(\theta) = \mathbb{E}_{\Gamma \sim \mathcal{B}} \left[ \mathbb{E}_{v^w \sim \Upsilon^w, v^s \sim \Upsilon^s} \left[ \sum_{i=t}^{t+H} \lambda^{i-t} \mathcal{L}_{\text{TD-MPC2}}(\theta; \mathcal{L}_{\text{rew}}, \mathcal{L}_Q, \mathcal{L}_{\text{LC}}, \Gamma_i) + \mathcal{L}_{\text{ER}}(\theta; v^w, v^s) \right]}}_{\mathcal{L}_{\text{ViGMO}}(\theta; v^w, v^s)}$$

where $\mathcal{L}_{\text{TD-MPC2}}$ denotes the standard world model learning objective, consisting of a reward prediction loss ($\mathcal{L}_{\text{rew}}$) and a Q-function loss ($\mathcal{L}_Q$), further augmented with the LC loss ($\mathcal{L}_{\text{LC}}$) and the ER loss ($\mathcal{L}_{\text{ER}}$) for stabilized, augmentation-invariant representations.

The terms highlighted in red are the only additions to the original TD-MPC2 objective; consequently, ViGMO can be seamlessly integrated into *any* latent-space MBRL framework without modifying its planner or policy update mechanisms. The training procedure is provided in Algorithm 1 (Appendix A), implementation details including hyperparameters are given in Appendix B, and a detailed discussion of each component is provided in Appendix E.

## 4 Experiments

To validate the efficacy of ViGMO, we compare its zero-shot generalization performance with state-of-the-art MFRL and MBRL methods on six continuous control tasks from the DMC suite (Tassa et al., 2018) and four manipulation tasks from Robosuite (Zhu et al., 2020). All agents are trained exclusively on environments with clean backgrounds and evaluated under complex, unseen visual distractions, including natural video overlays (Figures 3, 8, and 9). Our experiments aim to address the following key research questions:

**Q1.** Does ViGMO achieve superior zero-shot visual generalization while maintaining the sample efficiency of latent-space MBRL, compared with state-of-the-art MFRL and MBRL methods under unseen distractions?

**Q2.** How do the LC and ER losses individually contribute to the effectiveness of ViGMO?

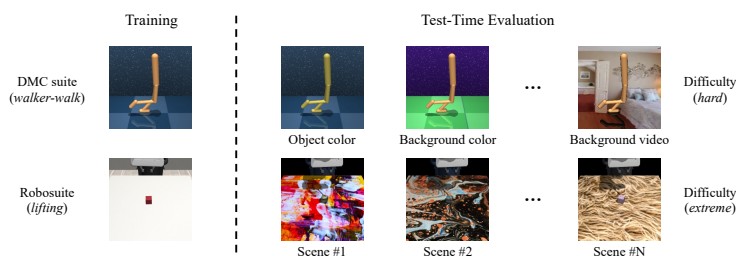

Figure 3: **Training and evaluation scenes.** Agents are trained in the default clean environment (leftmost column) and evaluated *zero-shot* under challenging unseen visual distractions, including color shifts, background videos, and scene variations. The top row shows examples from the DMC suite, and the bottom row shows examples from Robosuite.

**Q3.** How do different design choices—such as the randomness of augmentations over the horizon, the selection of strong augmentation techniques, and the choice of an auxiliary task for representation learning—affect ViGMO's generalization performance?

**Q4.** Does enforcing latent-level consistency mitigate the model collapse and preserve rollout accuracy over long horizons under visual shifts?

## 4.1 EXPERIMENTAL SETUP

**Environments.** We benchmark ViGMO on six DMC suite tasks—*walker-walk*, *finger-spin*, *cheetah-run*, *walker-stand*, *reacher-easy*, and *cartpole-swingup*—and four Robosuite manipulation tasks—*door-opening*, *nut-assembly*, *peg-in-hole*, and *lifting*. Detailed environment specifications are provided in Appendix C.

**Baselines.** To ensure a fair and comprehensive comparison, we evaluate ViGMO against seven strong visual RL baselines spanning both model-free and model-based paradigms: **MFRL:** (i) SVEA (Hansen et al., 2021), which stabilizes off-policy Q-learning through augmentation; (ii) DrQ-v2 (Yarats et al., 2022), a highly sample-efficient method leveraging data augmentation; (iii) SGQN (Bertoin et al., 2022), which integrates self-supervised learning with attribution-based Q-value regularization; and (iv) SRM (Huang et al., 2022), which enhances robustness to spatial corruption via spectrum augmentations. **MBRL:** (v) Dr. G (Ha et al., 2023), which targets zero-shot visual generalization via dual contrastive learning and recurrent inverse dynamics; (vi) TD-MPC2 (Hansen et al., 2024), which achieves state-of-the-art performance via latent-space planning without reconstructions; and (vii) DreamerV3 (Hafner et al., 2025), another state-of-the-art method based on latent imagination with pixel-level reconstruction.

## 4.2 EXPERIMENTAL RESULTS

We evaluate ViGMO on two key metrics—*zero-shot generalization* and *sample efficiency*—which together directly address (**Q1**). All generalization scores are averaged over five random seeds, and sample efficiency scores are averaged over ten seeds.

**Zero-shot generalization.** All agents are trained exclusively in a default clean setting and evaluated *zero-shot* on environments with unseen, complex distractors. These distractors include object- and background-color shifts, natural video backgrounds, and scene-texture changes, as shown in Figures 3, 8, and 9. The evaluation spans three difficulty levels: *easy*, *hard*, and an additional *extreme* level for Robosuite only. Table 1 summarizes the aggregated results across all tasks and distraction levels, with per-task and per-level breakdowns provided in Appendix D.1.

To ensure fair and reliable comparisons, we report three complementary metrics: mean, median, and the inter-quantile mean (IQM). The IQM averages the central 50% of returns, reducing sensitivity to random seeds and outliers (Agarwal et al., 2021). Together, these metrics provide a comprehensive view of performance, capturing both overall central tendency and robustness across random seeds.

Across both benchmarks, ViGMO consistently achieves the strongest overall zero-shot generalization. On the DMC suite, it outperforms all baselines across all three metrics (mean: 817.7, median: 879.3, IQM: 896.1), surpassing the next best method (SVEA) by a clear margin. On Robosuite, where most baselines collapse, ViGMO achieves substantially higher returns (mean: 116.5, median: 115.1, IQM: 54.2), establishing robustness advantages in complex manipulation tasks.

Table 1: **Zero-shot generalization scores.** Scores report mean, median, and IQM episode returns, averaged across all tasks and distraction levels. ViGMO consistently achieves the strongest overall zero-shot generalization. Values in brackets represent the lower and upper bounds of 95% stratified bootstrap confidence intervals (CIs). Boldface denotes the best score per metric.

| ENV | METRIC | SVEA | DRQ-v2 | SGQN | SRM | DR. G | TD-MPC2 | DREAMERV3 | ViGMO (OURS) |
|---|---|---|---|---|---|---|---|---|---|
| DMC SUITE | MEAN | 764.1, [755.3, 772.8] | 515.4, [503.3, 527.4] | 524.7, [516.7, 532.7] | 662.7, [653.9, 671.4] | 579.9, [567.7, 591.8] | 654.9, [642.8, 667.3] | 645.0, [632.9, 657.1] | **817.7, [812.1, 823.1]** |
| | MEDIAN | 810.4, [789.1, 824.0] | 559.3, [540.6, 577.7] | 550.2, [542.1, 558.0] | 812.8, [794.3, 827.9] | 590.0, [568.9, 614.4] | 707.5, [687.6, 730.0] | 660.9, [637.1, 684.4] | **879.3, [868.6, 889.6]** |
| | IQM | 867.8, [860.6, 874.7] | 533.7, [513.3, 553.4] | 549.5, [537.7, 561.2] | 772.9, [759.5, 785.3] | 634.2, [614.6, 653.1] | 762.1, [744.2, 779.1] | 762.4, [743.0, 780.8] | **896.1, [891.9, 899.7]** |
| ROBOSUITE | MEAN | 81.4, [73.9, 89.0] | 35.6, [33.6, 37.8] | 71.9, [67.6, 76.5] | 103.5, [92.3, 115.1] | 48.5, [46.4, 50.7] | 43.3, [41.1, 45.5] | 83.5, [74.9, 92.5] | **116.5, [103.1, 130.1]** |
| | MEDIAN | 49.5, [37.9, 61.8] | 2.4, [1.1, 4.6] | 40.7, [33.9, 48.4] | 85.1, [66.1, 104.6] | 10.8, [8.0, 13.9] | 3.3, [2.6, 4.2] | 61.4, [44.3, 79.3] | **115.1, [90.7, 130.9]** |
| | IQM | 37.2, [29.0, 46.2] | 2.1, [1.3, 3.3] | 43.9, [37.5, 50.7] | 49.4, [41.2, 58.4] | 10.5, [7.9, 13.6] | 3.7, [2.9, 4.5] | 35.5, [27.7, 43.9] | **54.2, [44.5, 65.0]** |

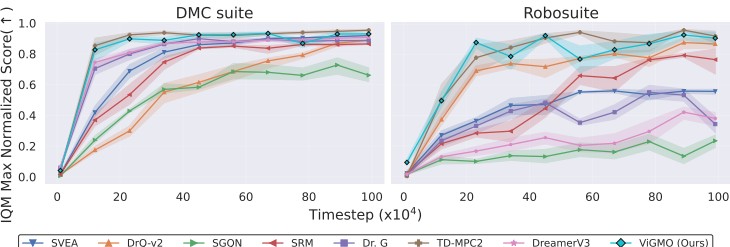

Figure 4: **Sample efficiency results.** Learning curves on the DMC suite and Robosuite, reported as IQM scores normalized by the maximum return across methods. ViGMO matches the efficiency of strong baselines while maintaining robustness to unseen distractions. Shaded regions denote 95% stratified bootstrap CIs.

**Sample efficiency.** To assess sample efficiency, we analyze learning curves of ViGMO and baselines under clean training conditions, reporting IQM scores normalized by the maximum return across all methods. Figure 4 presents aggregated results for both the DMC suite and Robosuite, with detailed per-task learning curves and statistical analyses provided in Appendix D.2.

ViGMO achieves strong sample efficiency on both benchmarks. On the DMC suite, ViGMO matches the rapid convergence of TD-MPC2 while requiring far fewer samples than most baselines. On Robosuite, the gains are even more pronounced: ViGMO learns faster and achieves higher asymptotic returns, especially compared with SGQN, SVEA, DreamerV3, and Dr. G. These results demonstrate that ViGMO preserves the hallmark efficiency of latent-space MBRL while substantially enhancing robustness to unseen distractions.

**Summary.** ViGMO achieves the best efficiency–robustness trade-off among all baselines, combining near-TD-MPC2 efficiency with significantly stronger zero-shot generalization under severe distribution shifts.

### 4.3 ABLATION STUDY

To further investigate the sources of ViGMO's performance gains, we conduct ablation studies addressing **(Q2)** and **(Q3)**. Each factor is analyzed in its own paragraph below, and the *summary* paragraph reports the aggregated findings.

**Individual contributions of LC and ER losses.** We ablate the two auxiliary losses to assess their individual contributions in addressing **(Q2)** (Table 2). On the DMC suite, removing the LC loss reduces performance to $84\%$ of ViGMO, while excluding ER lowers it to $96\%$, confirming that both components are important for stability and robustness. On Robosuite, where distribution shifts are substantially more severe, the necessity of these components becomes even clearer: removing LC causes performance to collapse to nearly zero ($1\%$ of ViGMO), while removing ER retains partial robustness ($51\%$) but with very high variance. These results indicate that LC is crucial for maintaining stable latent dynamics under extreme perturbations, whereas ER is indispensable for preventing representational collapse and ensuring consistent performance.

**Dynamic vs. consistent augmentations.** In MBRL, synthetic rollouts span a prediction horizon $H$, but it remains unclear whether augmentations should remain fixed across this horizon or be resampled dynamically. ViGMO applies both weak and strong augmentations at every step to support latent consistency (Section 3.2) and encoder regularization (Section 3.3). For a trajectory segment $s_{t:t+H}$, we define

$$s_k^w = \tau_k^w(s_k, v_k^w), \qquad s_k^{ws} = \tau_k^s(\tau_k^w(s_k, v_k^w), v_k^s), \qquad k = t, \ldots, t+H.$$

Table 2: **Zero-shot generalization scores.** Ablation on LC and ER losses.

| TASK | DIFFICULTY | W/O_LC_LOSS & W/O_ER_LOSS | W/O_LC_LOSS | W/O_ER_LOSS | ViGMO (OURS) |
|---|---|---|---|---|---|
| REACHER-EASY | EASY | $857.4 \pm 294.7$ | $926.5 \pm 174.8$ | $960.2 \pm 91.4$ | $\mathbf{982.4 \pm 10.5}$ |
| | HARD | $579.8 \pm 441.9$ | $678.7 \pm 380.7$ | $899.4 \pm 172.5$ | $\mathbf{946.7 \pm 154.4}$ |
| | AVERAGE | $718.6 \pm 400.0 \ (75\%)$ | $810.4 \pm 315.0 \ (84\%)$ | $929.8 \pm 141.2 \ (96\%)$ | $\mathbf{964.5 \pm 110.8 \ (100\%)}$ |
| DOOR-OPENING | EASY | $0.9 \pm 0.4$ | $2.0 \pm 4.0$ | $236.9 \pm 224.8$ | $\mathbf{374.2 \pm 207.3}$ |
| | HARD | $0.9 \pm 0.9$ | $1.6 \pm 3.3$ | $34.9 \pm 82.7$ | $\mathbf{123.5 \pm 204.6}$ |
| | EXTREME | $0.9 \pm 0.3$ | $0.7 \pm 0.0$ | $33.1 \pm 87.8$ | $\mathbf{99.3 \pm 193.6}$ |
| | AVERAGE | $0.9 \pm 0.6 \ (1\%)$ | $1.4 \pm 3.0 \ (1\%)$ | $101.7 \pm 174.5 \ (51\%)$ | $\mathbf{199.0 \pm 235.5 \ (100\%)}$ |

If $v_t^{w,s} = v_{t+k}^{w,s} \ \forall k \in [0, H]$, the scheme is called *consistent*; otherwise it is *dynamic*. By default, ViGMO adopts the dynamic setting ($v_t^{w,s} \neq v_{t+k}^{w,s}$), whereas the consistent variant, evaluated in our ablations, is denoted `ViGMO_CONST_AUG`.

**Choice of strong augmentation.** ViGMO employs *random-overlay* as its strong augmentation, while prior approaches have shown that alternative transformations can also improve generalization (Lee et al., 2020; Laskin et al., 2020b; Hansen et al., 2021). To evaluate this design choice, we compare against a convolution-based alternative, *random-conv*. Formally,

$$\tau^{s,\text{overlay}}(o, \tilde{o}) = (1 - \delta) \, o + \delta \, \tilde{o}, \qquad \tau^{s,\text{conv}}(o, w) = \text{CONV}(o, w),$$

where $o$ denotes the input image, $\tilde{o} \sim \mathcal{D}$ is a task-irrelevant overlaying image sampled from a distractor dataset, $\delta = 0.5$ is the blending coefficient, and $w \sim \mathcal{N}(0, 1)$ is a random convolution kernel. Both transformations are applied to every frame in the stacked state $s_t = \{o_t, o_{t-1}, \ldots, o_{t-k+1}\}$.

We adopt *random-overlay* as the default since it better mimics natural visual distractions encountered in deployment, such as dynamic textures or background clutters. In contrast, *random-conv* applies synthetic pixel-level perturbations, providing a complementary stress test for representation robustness. The ablation variant with *random-conv* in place of *random-overlay* is denoted `ViGMO_CONV`.

**Contrastive auxiliary loss.** A robust visual encoder is essential for both sample efficiency and generalization in visual RL. To this end, recent investigations augment the policy optimization with self-supervised objectives for representation learning (He et al., 2020; Laskin et al., 2020a; Bertoin et al., 2022; Nair et al., 2023). Motivated by these findings, we compare two contrastive losses:

- **SODA** (Hansen & Wang, 2021), the default regularizer in ViGMO.
- **CURL** (Laskin et al., 2020a), defined as:

$$\mathcal{L}_{\text{CURL}} = \log \frac{\exp\left(q^\top W k_+\right)}{\exp\left(q^\top W k_+\right) + \sum_{j=1}^{B} \exp\left(q^\top W k_j\right)},$$

where the anchor $q = h_\theta(\tau^w(s_t, v^q))$ and key $k = h_{\theta^-}(\tau^w(s_t, v^k))$ are two weakly augmented views of the same image, $W$ is a learnable bilinear projection, and $v^q, v^k \sim \Upsilon^w$.

The variant `ViGMO_CURL` replaces the SODA loss with CURL, i.e., $\mathcal{L}_{\text{ER}} = \mathcal{L}_{\text{CURL}}$ instead of $\mathcal{L}_{\text{SODA}}$.

**Summary.** Figure 5 summarizes the ablation results addressing **(Q3)** on *finger-spin*, *cheetah-run*, and *walker-walk*. ViGMO consistently outperforms its ablated variants—`ViGMO_CONST_AUG`, `ViGMO_CONV`, and `ViGMO_CURL`—in both zero-shot generalization and sample efficiency. These results highlight three key findings: (i) dynamic augmentations across horizons are crucial for stable rollouts, (ii) *random-overlay* provides stronger robustness than convolution-based perturbations, and (iii) SODA regularization better complements MA and LC than CURL. Together, these observations confirm that each design choice plays an indispensable role in enabling ViGMO's strong generalization performance while maintaining high sample efficiency. Detailed results are provided in Appendix D.4.

### 4.4 LATENT-SPACE CONSISTENCY ANALYSIS

To address **(Q4)**, we analyze whether the proposed consistency loss stabilizes latent dynamics and prevents collapse under long-horizon predictions with visual perturbations. We compare ViGMO with its backbone TD-MPC2 and the competitive baseline Dr. G. For each agent, we roll out the respective learned world models from identical initial inputs and record the predicted latent states $z_{t+1} = d_\theta(z_t, a_t)$ along with environment rewards $r_t$. To visualize these high-dimensional embeddings, we apply UMAP (McInnes et al., 2018) and plot the resulting 2D trajectories (Figure 6), with

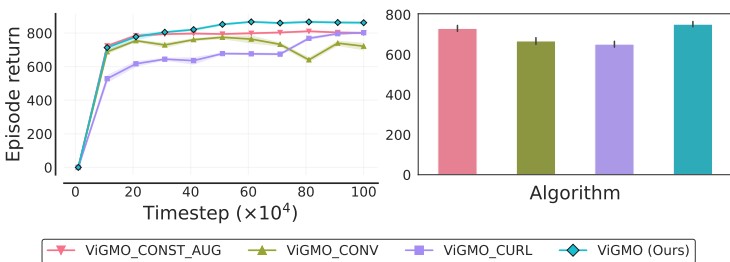

Figure 5: **Ablation study on design choices. Left:** Learning curves (sample efficiency). **Right:** Zero-shot generalization performance. Across both metrics, ViGMO consistently outperforms its ablated variants (`ViGMO_CURL`, `ViGMO_CONST_AUG`, `ViGMO_CONV`), highlighting the importance of each design choice. Shaded regions and error bars denote the $95\%$ confidence intervals and standard errors.

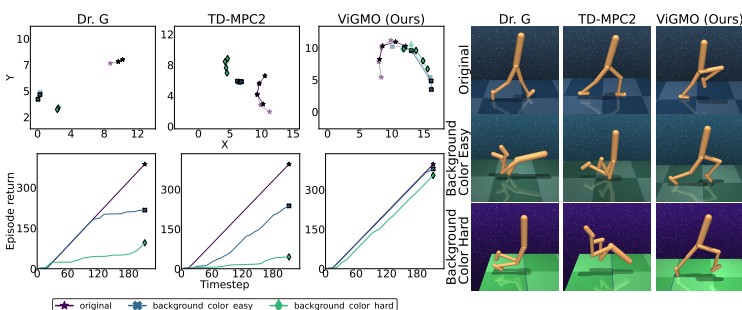

Figure 6: **Latent-space consistency under visual perturbations. Left:** UMAP projections of latent embeddings (top) and episode returns (bottom). **Right:** Environment snapshots across evaluation types. Markers denote difficulty levels ($\star$: *original*, $\times$: *easy*, $\diamond$: *hard*), and arrows indicate temporal progression within each rollout segment $s_{t:t+H}$. ViGMO preserves aligned latent manifolds and stable performance across difficulty levels, while TD-MPC2 and Dr. G show divergence and degraded execution under perturbations.

faded points indicating earlier time steps. For additional intuition, we also show episode returns and environment snapshots across different difficulty levels.

Figure 6 presents the results for the *walker_walk* task under the *background-color* perturbation. ViGMO maintains a single, well-aligned latent manifold across difficulty levels, consistently aligning perturbed trajectories with their clean counterparts. This structural consistency translates into stable task performance, whereas TD-MPC2 and Dr. G yield scattered or divergent rollouts under perturbations, leading to degraded returns and failed executions as evident in the snapshots. These findings corroborate our hypothesis (Figure 1) that mapping OOD inputs back onto the in-domain latent manifold is critical for generalization, and demonstrate that ViGMO uniquely enforces temporal and structural consistency in latent space—a property not observed in the baselines. Additional visualizations are provided in Appendix D.5.

## 5 CONCLUSION

We introduced **ViGMO**, a novel MBRL framework that achieves strong zero-shot visual generalization while preserving high sample efficiency. On challenging OOD benchmarks, ViGMO outperforms state-of-the-art MFRL and MBRL baselines, improving zero-shot generalization by up to $13\%$ over the strongest baseline while maintaining the hallmark sample efficiency of latent-space MBRL. These gains stem from the integration of MA, LC, and ER, which together equip the world model with invariance to visual perturbations and robustness under severe distribution shifts, thereby achieving the best efficiency–robustness trade-off among all baselines.

**Limitations and future work.** The empirical success of ViGMO opens up interesting theoretical questions regarding model-based generalization (Ghugare et al., 2023; Lyu et al., 2024). Further theoretical analysis could provide deeper insights into its effectiveness. Moreover, current benchmarks primarily evaluate visual distribution shifts. Extending ViGMO to broader dynamics- or task-level shifts (Seo et al., 2020; Beukman et al., 2023), as well as validating it in real-robot deployments, are important directions for building more robust and practically deployable RL agents.

## REPRODUCIBILITY STATEMENT

The training pseudocode is presented in Appendix A, and implementation details, including the code base, computational resources, and hyperparameters, are provided in Appendix B. Evaluation details for zero-shot generalization, covering visual distraction categories, evaluation protocols, and evaluation scenes, are described in Appendix C. Supplementary results, including per-task and per-distraction breakdowns as well as extensive ablation studies, are reported in Appendix D.

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

## APPENDIX CONTENTS

## A  TRAINING PROCEDURE

This section details the training procedure of ViGMO. Since ViGMO leaves the underlying planning mechanism unchanged, we focus on world model learning and omit planner updates in Algorithm 1.

For a trajectory segment $\Gamma = (s_t, a_t, r_t, s_{t+1})_{t:t+H}$ sampled from the replay buffer $\mathcal{B}$, the reward and Q-function losses are defined as:

$$\mathcal{L}_{\text{rew}}(\theta) = \text{CE}(\hat{r}_t, r_t), \qquad \mathcal{L}_Q(\theta) = \text{CE}(\hat{q}_t, q_t),$$

where $\hat{r}_t = R_\theta(z_t, a_t)$ and $\hat{q}_t = Q_\theta(z_t, a_t)$ denote the predicted reward and Q-value, respectively. The TD target at time $t$ is given by $q_t = r_t + \bar{Q}_\theta(z_{t+1}, \pi_\theta(z_{t+1}))$, where $\bar{Q}_\theta$ is an exponential moving average of the Q-function. Here, $z_t = h_\theta(s_t)$ is the latent representation produced by the encoder, $\pi_\theta$ is a parameterized policy trained via entropy maximization, and CE denotes the cross-entropy loss. For additional details on the world model architecture and planning procedure, we refer readers to Hansen et al. (2024).

---

**Algorithm 1** World model learning in ViGMO

---

**Input:** Replay buffer $\mathcal{B}$; Horizon $H$; Weak and strong augmentation functions $\tau^w, \tau^s$; Learning rate $\eta$; Target update rate $\delta$

**while** not converged **do**
    **for** gradient step $t_g = 1, 2, \ldots, T_g$ **do**
        $\Gamma = (s_t, a_t, r_t, s_{t+1})_{t:t+H} \sim \mathcal{B}$         {Sample a trajectory segment from the replay buffer}
        $L \leftarrow 0$         {Initialize cumulative loss}
        **for** $i = t, t+1, \ldots, t+H$ **do**
            $\upsilon^w \sim \Upsilon^w, \upsilon^s \sim \Upsilon^s$         {Sample augmentation parameters}
            $L \leftarrow L + \lambda^{i-t} \mathcal{L}_{\text{ViGMO}}(\theta; \upsilon^w, \upsilon^s)$         {Calculate ViGMO loss}
        **end for**
        $\theta \leftarrow \theta + \eta \frac{1}{H} \nabla_\theta L$         {Update online network parameters}
        $\theta^- \leftarrow (1-\delta)\theta^- + \delta\theta$         {Update target network parameters}
    **end for**
**end while**

---

# B  IMPLEMENTATION DETAILS

## B.1  CODE BASE

We evaluate ViGMO against four MFRL baselines—SVEA (Hansen et al., 2021), DrQ-v2 (Yarats et al., 2022), SGQN (Bertoin et al., 2022), and SRM (Huang et al., 2022)—and three MBRL baselines: Dr. G (Ha et al., 2023), TD-MPC2 (Hansen et al., 2024), and DreamerV3 (Hafner et al., 2025). For the MFRL baselines, we adopt the reference implementations from RL-ViGen (Yuan et al., 2024)[1], which provide a unified and reliable benchmark. For the MBRL baselines, we use the official implementations of Dr. G[2] and DreamerV3[3], while our implementation of ViGMO is built on the official TD-MPC2 repository[4], with modifications restricted to the additional components introduced in Section 3.

## B.2  COMPUTATIONAL RESOURCES

All experiments were conducted on a single NVIDIA A5000 GPU. On average, a complete training and evaluation run took approximately 48 hours for the DMC suite tasks and 72 hours for the Robosuite tasks.

## B.3  HYPERPARAMETERS

To ensure a fair comparison, we reuse the same task-specific settings (e.g., action repeat, frame stack) and common hyperparameters across all baselines whenever possible. Table 3 summarizes the shared hyperparameters used in both DMC suite and Robosuite experiments, while algorithm-specific settings are reported separately in Tables 4 and 5. For ViGMO, we adopt TD-MPC2 defaults (Hansen et al., 2024) unless otherwise noted, and introduce a small set of additional parameters highlighted in blue (Table 5). This separation clarifies which components are inherited and which are novel to ViGMO, ensuring reproducibility and transparent comparison.

Table 3: **Common hyperparameters.** Shared settings used across all baselines (including ViGMO) in both DMC suite and Robosuite experiments.

| Hyperparameter | Value |
| --- | --- |
| Discount factor $\gamma$ | 0.99 |
| Replay buffer size $B$ | Unlimited (same with $T_g$) |
| Action repeats | 2 (except Dreamer family and Dr. G, which use task-specific values) |
| Frame stack $k$ | 3 (1 for Dreamer family) |
| Pixel RGB image space | $o_t \in \mathcal{O}^{64 \times 64 \times 3}$ (model-based), $o_t \in \mathcal{O}^{84 \times 84 \times 3}$ (model-free) |
| Maximum episode length | 1,000 (DMC suite), 500 (Robosuite) |
| Batch size | 16 (DreamerV3), 50 (Dr. G), (512 (*walker-{walk,stand}* tasks), 256 (otherwise) (TD-MPC2, ViGMO, and model-free baselines)) |
| Total gradient steps $T_g$ | 1,000,000 |
| Total seeding steps | 2,500, 1,250 (model-based; DMC suite and Robosuite), 4,000 (model-free) |
| Periodic evaluation steps during training | 10,000 |
| $N$ steps for TD target | 1 (model-based), 3 (model-free) |
| MLP hidden layer dimension | 1024 (DreamerV3), 512 (TD-MPC2 and ViGMO), 200 (Dr. G), 1024 (model-free) |
| Latent dimension | 512 (TD-MPC2 and ViGMO), 30 (Dr. G), 50 (model-free) |
| Activation function | LayerNorm + Mish (TD-MPC2 and ViGMO), RMSNorm + SiLU (DreamerV3), ReLU + ELU (Dr. G), ReLU (model-free) |
| Target network EMA weight | 5e-2 (Dr. G), 2e-2 (DreamerV3), 1e-2 (otherwise) |

For MFRL baselines, we compute an $N$-step TD target for value function learning:

$$r_t + r_{t+1} + \cdots + r_{t+N} + \gamma Q(s_{t+N}, \pi(s_{t+N})).$$

Training and evaluation use image observations of size $84 \times 84 \times 9$, where each state $s_t = \{o_t, o_{t-1}, o_{t-2}\}$ is constructed by stacking three consecutive RGB frames $o_t \in \mathcal{O}^{84 \times 84 \times 3}$ along the channel axis. Action repeat is applied following prior work, using a fixed number of repeated actions sampled from the policy or model planner. Algorithm-specific hyperparameters are listed in Table 4.

---

[1] https://github.com/gemcollector/RL-ViGen
[2] https://github.com/JeongsooHa/DrG
[3] https://github.com/danijar/dreamerv3
[4] https://github.com/nicklashansen/tdmpc2

Table 4: **MFRL baseline hyperparameters.** Algorithm-specific settings for MFRL baselines used in DMC suite and Robosuite experiments. Common hyperparameters are provided in Table 3.

| Hyperparameter | Value |
|---|---|
| Periodic critic target network ($\theta^-$) update steps | 1 |
| Clip constant for the stochastic actor | 3e-2 |
| Learning rate for the auxiliary task | 3e-4 (SGQN) |
| Attribution mask quantile | 0.95 (SGQN) |

For ViGMO, we retain all TD-MPC2 defaults unless otherwise stated, and add a small set of parameters specific to our framework. These ViGMO-specific hyperparameters are highlighted in blue in Table 5.

Table 5: **ViGMO hyperparameters.** Full set of hyperparameters used in ViGMO, which builds upon TD-MPC2 defaults. ViGMO-specific parameters are highlighted in blue.

| Hyperparameter | Value |
|---|---|
| **MPC planning** | |
| Planning Horizon $H$ | 3 |
| Std. range | $\sigma \in [0.05, 2]$ |
| Population size | 512 |
| Elite fraction | 64 |
| Iterations | 6 |
| Policy prior samples | 24 |
| Sampling temperature | 0.5 |
| | |
| **Model learning** | |
| Temporal coefficient $\lambda$ | 0.5 |
| Reward loss coefficient $c_1$ | 0.1 |
| Q-value loss coefficient $c_2$ | 0.1 |
| Latent consistency loss coefficient $c_3$ | 20 |
| | |
| **Optimization** | |
| Learning rate $\eta$ | 3e-4 |
| Periodic target network ($\theta^-$) update steps $\delta$ | 1 |
| Optimizer | Adam ($\beta_1 = 0.9$, $\beta_2 = 0.999$) |
| Exploration schedule (std) | Linear (0.5, 0.05, 25,000 steps) |
| Planning horizon schedule | Linear (1, 5, 25,000 steps) |
| | |
| **ViGMO-specific hyperparameters** | |
| Weak augmentation $\tau^w$ | *random-shift*: padding $p = 4$ |
| Strong augmentation $\tau^s$ | *random-overlay*: $\begin{cases} \text{linear interpolation} \quad \delta = 0.5 \\ \text{Image dataset} \quad \mathcal{D} = \text{Places (Zhou et al., 2017)} \end{cases}$ |
| Augmentation ratio $\zeta$ | 1.0 (half weak-only, half weak-to-strong) |

# C EVALUATION DETAILS

We follow the zero-shot evaluation protocol of Yuan et al. (2024) and define **fourteen** distraction types for the DMC suite and **three** for Robosuite. Each DMC distraction includes two difficulty levels (*easy*, *hard*), while Robosuite includes three levels (*easy*, *hard*, *extreme*). These distractions induce distribution shifts in color, shadows, background, and camera viewpoint, providing a rigorous testbed for visual generalization.

## C.1 ENVIRONMENTS AND TASKS

We benchmark ViGMO on two widely used continuous-control suites: the DMC suite and Robosuite. Representative environments are illustrated in Figure 7.

In the **DMC suite** (Tassa et al., 2018), we consider six visuomotor control tasks: *cartpole-swingup*, where the agent must swing up and balance a cartpole; *finger-spin*, which requires continuous spinning of a planar finger; *walker-walk*, which trains a bipedal walker to move forward stably; *walker-stand*, which focuses on maintaining balance in an upright posture; *cheetah-run*, where the agent learns high-speed forward locomotion; and *reacher-easy*, which involves controlling a 2-DoF arm to reach a target location.

For **Robosuite** (Zhu et al., 2020), we evaluate four robotic manipulation tasks: *door-opening*, where the robot arm must open a hinged door; *nut-assembly*, which requires placing a nut onto a peg; *peg-in-hole*, where a peg must be inserted into a narrow slot; and *lifting*, which involves grasping and lifting a cube.

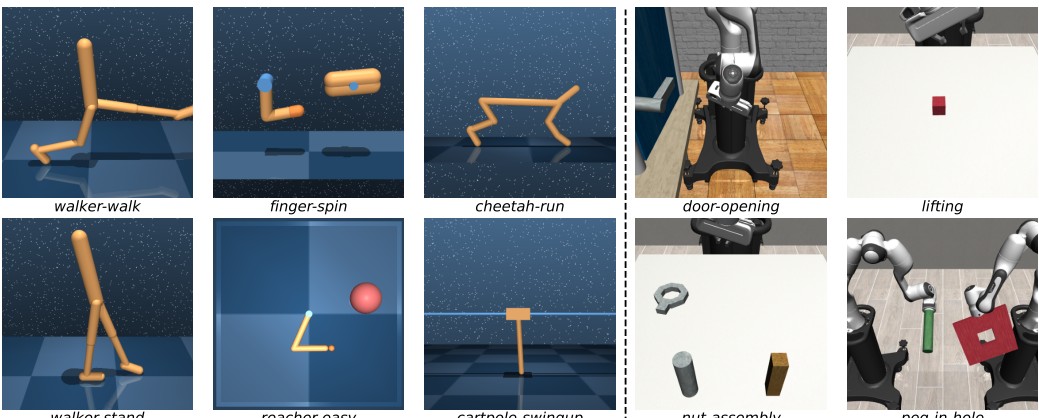

Figure 7: **Environments and tasks.** Six visuomotor control tasks from the DMC suite and four robotic manipulation tasks from Robosuite.

## C.2 VISUAL DISTRACTION CATEGORIES

We consider seven types of distractions, each with two difficulty levels for the DMC suite and three difficulty levels for Robosuite:

- *background-color*: Change the background color of the agent (e.g., terrain grid or background sky color).
    - Uniformly sample the parameters of the color, i.e., *(r,g,b)*, from the pre-defined distribution for each difficulty.
- *cam-pos*: Change the position of the tracking camera's focus by randomly adding noise offset.
    - Let the initial position of the tracking camera's focus be $X_{\text{cam}} = (x_i, y_i, z_i)$ in Euclidean space.
    - Sample a random offset $\delta \in \mathbb{R}^3$ from the uniform distribution with different bounds: $\mathcal{U}(-0.08, 0.08)$ for *easy* and $\mathcal{U}(-0.15, 0.15)$ for *hard* difficulty.

- – Inject the offset to the initial position of the camera; $X_{\text{cam}} = (x_i + \delta_x, y_i + \delta_y, z_i + \delta_z)$.
- *background-video*: Overlay the background with the randomly sampled natural video.
  - – Sample a random video with the same width and height as the original image from a set of natural videos (Stone et al., 2021).
  - – Overlay the video only to the background sky for *easy* and to all backgrounds, including the ground terrain other than the agent, for *hard* difficulty.
- *light-position*: Change the position and orientation of the tracking light of the agent.
  - – Following the approach used in (Stone et al., 2021), the tracking light's coordinate is parameterized as the spherical coordinate; $(\phi, \theta, r)$ where $\phi$ is azimuth, $\theta$ is inclination, and $r$ is the radius of the sphere.
  - – Sample $\phi$ from the normal distribution $\mathcal{N}(\pi/6, 1)$ for *easy* and $\mathcal{N}(\pi/3, 1)$ for *hard* difficulty.
  - – Sample $\theta \sim \mathcal{N}(2\pi, 1)$ and transform the initial pose of the tracking light $X_{\text{light}}$ to $(\phi, \theta, r)$ where $r = \sqrt{X_{\text{light}}}$.
- *light-color*: Change the color of the tracking light of the agent.
  - – Uniformly sample the parameters of the color, i.e., *(r,g,b)*, from the pre-defined distribution for each difficulty.
- *moving-light*: Rotate the tracking light of the agent around the agent.
  - – Likewise in *light_position*, the spherical coordinate of the tracking light is randomly initialized as $(\phi, \theta, r)$.
  - – Let the speed of azimuth rotation as $\Delta_\phi = \pi/200$ for *easy* and $\Delta_\phi = \pi/100$ for *hard* difficulty.
  - – Rotate the tracking light counterclockwise along the azimuth axis at every time-step; $(\phi, \theta, r) \leftarrow (\phi, \theta, r) + (\Delta_\phi, 0, 0)$.
- *object-color*: Change the color of the body color of the agent.
  - – Uniformly sample the parameters of the color, i.e., *(r,g,b)*, from the pre-defined distribution for each difficulty.

### C.3 EVALUATION PROTOCOL

For the DMC suite, each of the seven distraction types listed above includes two difficulty levels: *easy* and *hard*. Robosuite follows the predefined evaluation splits—*eval-easy*, *eval-hard*, and *eval-extreme*—as introduced in RL-ViGen (Yuan et al., 2024).

### C.4 EVALUATION SCENES

Representative evaluation scenes for the DMC suite and Robosuite are shown in Figure 8(a) and Figure 8(b), respectively. Agents are trained in the default *clean* setting and then evaluated *zero-shot* under diverse unseen distractions.

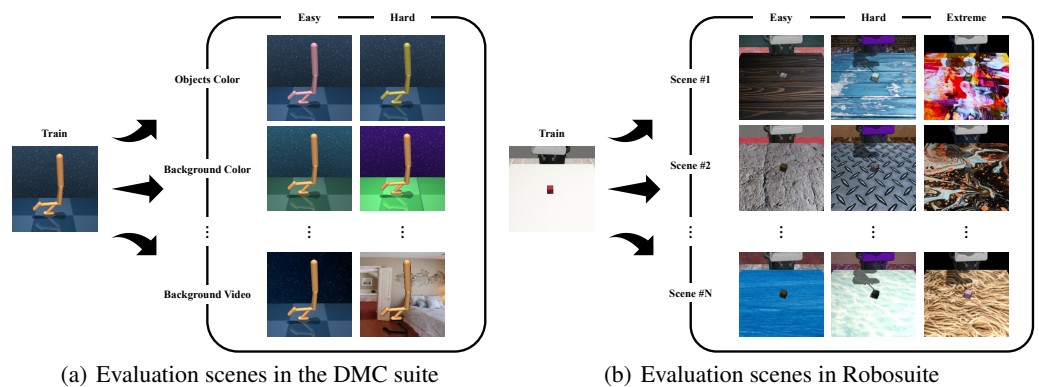

(a) Evaluation scenes in the DMC suite      (b) Evaluation scenes in Robosuite

Figure 8: **Evaluation scenes.** Agents are trained in clean environments and evaluated zero-shot under diverse unseen distractions across all tasks.

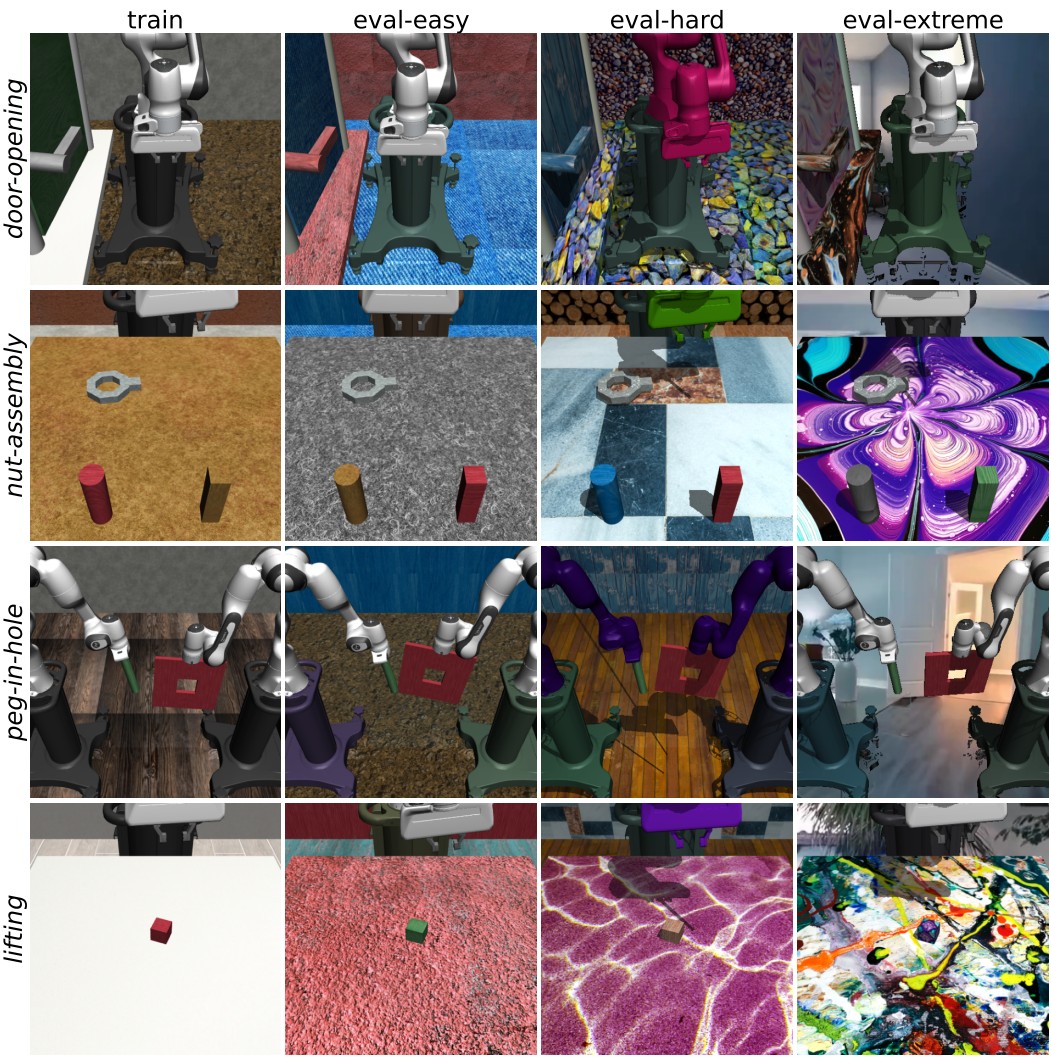

Figure 9: **Visual generalization evaluation in Robosuite.** Visual generalization setup in Robosuite, which includes four robotic manipulation tasks: *door-opening*, *nut-assembly*, *peg-in-hole*, and *lifting*. Each column represents a different level of visual distraction, from left to right: *train*, *eval-easy*, *eval-hard*, and *eval-extreme*. Each row shows a different task.

# D SUPPLEMENTARY RESULTS

This appendix complements Section 4.2 with additional figures, tables, and analyses. Unless otherwise noted, evaluation metrics are averaged over five random seeds; sample-efficiency statistics are averaged over ten seeds.

## D.1 ZERO-SHOT GENERALIZATION

This subsection provides a detailed breakdown of our zero-shot generalization results, complementing the aggregate scores presented in the main paper. We report full results across distraction types and tasks in Figures 10 and 11, respectively, with per-task averages for the DMC suite and Robosuite summarized in Table 6.

**Analysis across distraction types.** Figures 10 report zero-shot generalization performance across fourteen distraction types in the DMC suite and three in Robosuite. Across all methods, we observe a monotonic degradation in returns as the distraction difficulty increases (*original → easy → hard → extreme*), indicating the inherent challenge of distribution shifts in visual control. However, the magnitude of this degradation differs substantially across algorithms.

In the DMC suite, SGQN shows rapid declines in performance even under mild perturbations, confirming its limited robustness to visual variability. TD-MPC2, despite being one of the strongest latent-space MBRL baselines under clean settings, exhibits sharp drops when exposed to hard distractors, highlighting its sensitivity to severe visual shifts. In contrast, ViGMO consistently sustains higher returns across all distraction types and difficulty levels, demonstrating its robustness.

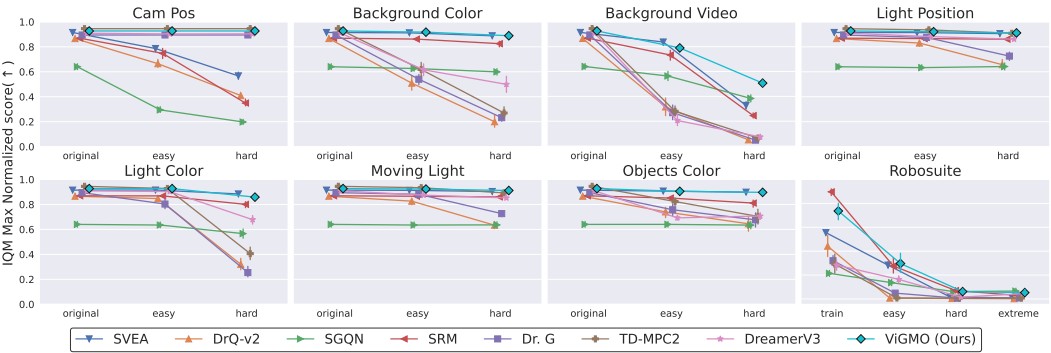

Figure 10: **Zero-shot generalization performance across distraction types.** We evaluate fourteen distraction types in the DMC suite and three in Robosuite. All baselines exhibit a monotonic performance drop as the difficulty increases (e.g., *original → easy → hard → extreme*). TD-MPC2 shows sharp performance drops under heavy perturbations. In contrast, ViGMO yields higher returns, particularly on the challenging Robosuite tasks.

**Per-task analysis.** Figure 11 and Table 6 provide a detailed breakdown of zero-shot generalization performance across all tasks in the DMC suite and Robosuite. While no single method dominates every task, ViGMO consistently ranks among the top performers and achieves the most reliable overall performance across benchmarks.

On the DMC suite, ViGMO achieves the highest average return of 817.7, which represents a relative improvement of approximately 7% over the next best baseline, SVEA (764.1). Task-level analysis shows that ViGMO achieves the strongest performance on *walker-walk* (878.1), *finger-spin* (880.6), and *reacher-easy* (964.5), outperforming the corresponding baselines by margins up to 20%. Although SRM and SVEA occasionally achieve the highest scores on specific tasks (e.g., *cheetah-run*, *walker-stand*, and *cartpole-swingup*), ViGMO's consistently strong results across multiple tasks yield the most reliable aggregate performance.

On the DMC suite, ViGMO achieves the highest average return of 817.7, which represents a relative improvement of approximately 7% over the next best baseline, SVEA (764.1). Task-level analysis

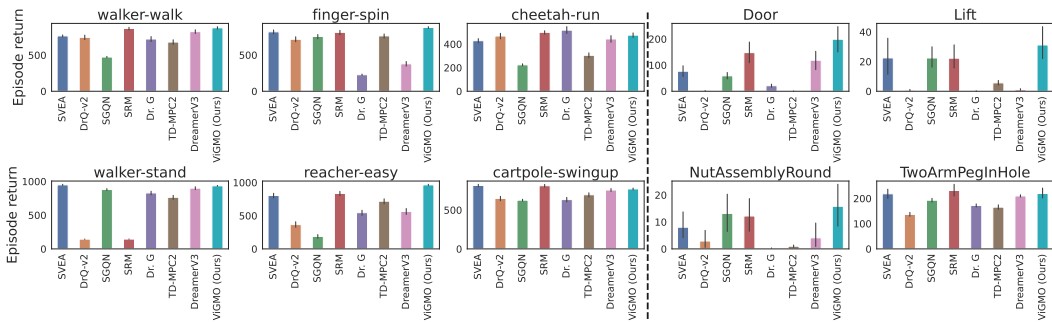

Figure 11: **Zero-shot generalization performance across tasks.** We evaluate six visuomotor control tasks from the DMC suite and four robotic manipulation tasks from Robosuite. While no single method achieves the best score on *every* task, ViGMO attains the most reliable overall performance across tasks.

Table 6: **Zero-shot generalization scores.** Reported values are mean episode returns averaged across all distraction levels for each task in the DMC suite and Robosuite. Boldface denotes the best score per task. ViGMO consistently ranks among the top performers and attains the most reliable overall performance.

| ENV | TASK | SVEA | DRQ-V2 | SGQN | SRM | DR. G | TD-MPC2 | DREAMERV3 | **VIGMO (OURS)** |
|---|---|---|---|---|---|---|---|---|---|
| DMC SUITE | WALKER-WALK | $765.8 \pm 152.2$ | $746.0 \pm 330.6$ | $473.4 \pm 106.9$ | $869.4 \pm 182.6$ | $722.9 \pm 352.1$ | $680.8 \pm 346.8$ | $824.9 \pm 274.4$ | $\mathbf{878.1 \pm 180.4}$ |
| | FINGER-SPIN | $820.2 \pm 298.6$ | $717.4 \pm 355.3$ | $755.7 \pm 313.2$ | $813.0 \pm 318.1$ | $228.0 \pm 99.2$ | $763.8 \pm 300.6$ | $378.5 \pm 327.2$ | $\mathbf{880.6 \pm 126.8}$ |
| | CHEETAH-RUN | $429.3 \pm 176.3$ | $469.1 \pm 255.0$ | $224.4 \pm 92.8$ | $500.2 \pm 154.3$ | $\mathbf{518.9 \pm 316.5}$ | $305.2 \pm 216.0$ | $444.1 \pm 307.5$ | $476.4 \pm 194.4$ |
| | WALKER-STAND | $\mathbf{948.9 \pm 90.9}$ | $144.3 \pm 28.3$ | $880.0 \pm 150.8$ | $145.3 \pm 28.1$ | $829.3 \pm 278.3$ | $764.8 \pm 315.4$ | $900.7 \pm 240.0$ | $935.0 \pm 71.5$ |
| | REACHER-EASY | $804.4 \pm 351.5$ | $366.0 \pm 431.2$ | $187.8 \pm 312.4$ | $835.8 \pm 321.3$ | $545.6 \pm 416.8$ | $718.6 \pm 400.0$ | $564.1 \pm 455.9$ | $\mathbf{964.5 \pm 110.8}$ |
| | CARTPOLE-SWINGUP | $\mathbf{816.3 \pm 164.8}$ | $649.5 \pm 293.8$ | $626.9 \pm 128.5$ | $812.6 \pm 189.1$ | $634.4 \pm 297.5$ | $696.4 \pm 273.7$ | $757.7 \pm 199.1$ | $771.7 \pm 131.1$ |
| | AVERAGE | $764.1 \pm 275.6$ | $515.4 \pm 375.7$ | $524.7 \pm 329.4$ | $662.7 \pm 343.9$ | $579.9 \pm 362.2$ | $654.9 \pm 352.0$ | $645.0 \pm 367.3$ | $\mathbf{817.7 \pm 216.9}$ |
| ROBOSUITE | DOOR-OPENING | $76.6 \pm 100.3$ | $1.8 \pm 3.5$ | $59.0 \pm 62.1$ | $147.8 \pm 184.9$ | $21.2 \pm 29.0$ | $0.9 \pm 0.6$ | $118.6 \pm 169.1$ | $\mathbf{199.0 \pm 235.5}$ |
| | NUT-ASSEMBLY | $8.0 \pm 22.6$ | $2.9 \pm 18.4$ | $13.2 \pm 34.3$ | $12.2 \pm 30.5$ | $0.3 \pm 0.2$ | $1.0 \pm 2.1$ | $4.1 \pm 21.8$ | $\mathbf{15.8 \pm 38.9}$ |
| | PEG-IN-HOLE | $218.5 \pm 87.5$ | $136.8 \pm 36.8$ | $193.0 \pm 36.0$ | $\mathbf{231.6 \pm 114.0}$ | $172.3 \pm 30.5$ | $165.6 \pm 43.0$ | $210.1 \pm 22.9$ | $219.8 \pm 96.5$ |
| | LIFTING | $22.5 \pm 59.8$ | $0.8 \pm 2.1$ | $22.4 \pm 34.2$ | $22.3 \pm 38.2$ | $0.2 \pm 0.5$ | $5.7 \pm 7.6$ | $1.2 \pm 2.5$ | $\mathbf{31.2 \pm 55.0}$ |
| | AVERAGE | $81.4 \pm 111.1$ | $35.6 \pm 62.1$ | $71.9 \pm 84.0$ | $103.5 \pm 143.7$ | $48.5 \pm 75.1$ | $43.3 \pm 74.0$ | $83.5 \pm 122.2$ | $\mathbf{116.5 \pm 161.1}$ |

shows that ViGMO achieves the strongest performance on *walker-walk* (878.1), *finger-spin* (880.6), and *reacher-easy* (964.5), exceeding the second-best methods by $1.0\%$ (vs. SRM, 869.4), $7.4\%$ (vs. SVEA, 820.2), and $15.4\%$ (vs. SRM, 835.8), respectively. Although SRM and SVEA occasionally achieve the highest scores on specific tasks (e.g., *cheetah-run*, *walker-stand*, and *cartpole-swingup*), ViGMO's consistently strong results across multiple tasks yield the most reliable aggregate performance.

On Robosuite, the performance gains are even more pronounced. ViGMO achieves the best average return of 116.5, surpassing the strongest baseline, SRM (103.5), by approximately $13\%$. Moreover, ViGMO achieves state-of-the-art results on three of the four tasks: *door-opening* (199.0), *nut-assembly* (15.8), and *lifting* (31.2). These results demonstrate that ViGMO maintains robustness even under visually challenging manipulation settings, whereas many baselines collapse to near-random performance.

**Summary.** Overall, these findings confirm that ViGMO's three key components—MA, LC, and ER—are essential for mitigating severe distribution shifts. By jointly promoting augmentation-invariant latent dynamics, enforcing temporal consistency, and preventing representational collapse, ViGMO enables reliable zero-shot generalization across both locomotion and robotic manipulation tasks. This detailed per-task analysis further corroborates the main paper's conclusion that conventional latent-space MBRL is fragile to OOD perturbations, whereas ViGMO achieves robust and stable generalization without requiring test-time adaptation.

### D.2 SAMPLE EFFICIENCY

An essential aspect of RL, alongside generalization, is achieving high sample efficiency. In this subsection, we provide a detailed comparison of sample efficiency across tasks, complementing the main results.

**Training curves.** Figure 12 presents the per-task learning curves for both the DMC suite and Robosuite, reporting IQM scores normalized by the maximum return across all methods. While no single method dominates across all tasks, ViGMO consistently ranks among the top performers and provides the most reliable performance overall. Importantly, ViGMO maintains this advantage despite being trained with more challenging objectives that incorporate visually distracting perturbations, confirming that robustness does not come at the cost of efficiency.

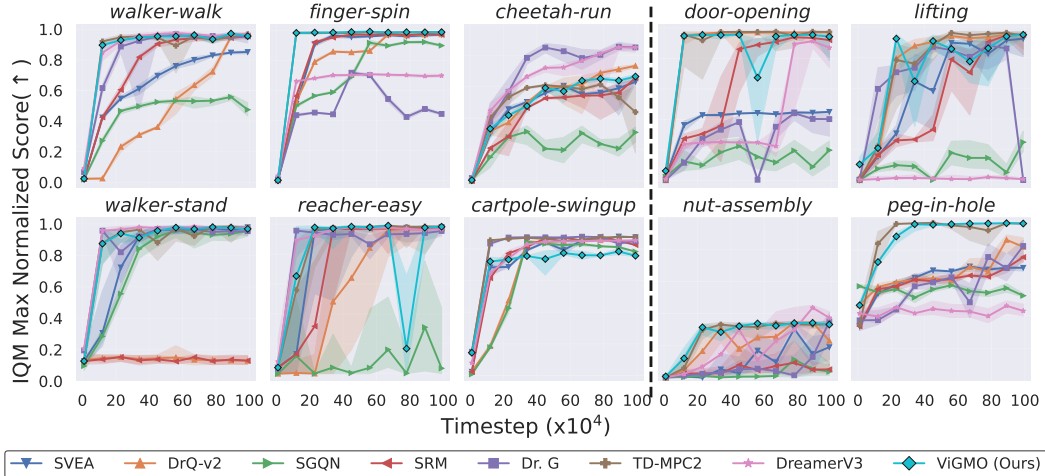

Figure 12: **Sample efficiency results across tasks.** Learning curves are plotted using IQM scores, normalized by the maximum return across all methods. Shaded regions indicate 95% stratified bootstrap CIs.

**Evaluation protocol.** To quantify sample efficiency, we adopt the evaluation protocol of Mai et al. (2022). We define DrQ-v2 as the oracle baseline and measure the number of environment episodes required for each method to reach 25%, 50%, and 75% of the oracle's asymptotic return. A task is considered "solved" once the threshold is crossed. If an agent fails within the training budget, we assign the maximum episode count as a penalty (1,000 for the DMC suite and 2,000 for Robosuite). This protocol provides a consistent and interpretable method for comparing algorithms across environments.

**Quantitative results** As shown in Table 7, ViGMO achieves sample efficiency broadly comparable to its backbone TD-MPC2, while substantially outperforming all other baselines.

On the DMC suite, ViGMO reaches the 50%-oracle threshold in 295 episodes (vs. 187 for TD-MPC2) and requires 423 episodes at the 75% level (vs. 327). At the stricter 25% threshold, ViGMO remains competitive with 160 episodes, close to the best-performing SVEA baseline (152) and substantially better than all other baselines.

On Robosuite, the efficiency gap between ViGMO and TD-MPC2 is more noticeable: ViGMO requires 700, 1085, and 1375 episodes to reach the 25%, 50%, and 75% thresholds, compared with 325, 680, and 945 for TD-MPC2. Nevertheless, ViGMO still dramatically outperforms all other baselines, which demand up to $1.6-2.7\times$ more interactions. Importantly, ViGMO achieves this level of efficiency despite being trained under visually perturbed settings, confirming that MA, LC, and ER preserve the hallmark efficiency of latent-space MBRL while simultaneously enabling robustness to unseen distractions.

**Statistical analysis.** To complement the aggregate metrics, Figure 13 presents empirical cumulative distribution functions (ECDFs) of episode returns across tasks. ECDFs capture the distribution of training efficiency and illustrate how quickly each method reaches target performance thresholds. ViGMO demonstrates ECDF profiles that closely track TD-MPC2, while surpassing all other baselines. This indicates that ViGMO maintains the hallmark efficiency of latent-space MBRL even under visually perturbed settings, further validating its robustness-efficiency balance.

Table 7: **Sample efficiency scores.** Entries report the mean number of episodes (lower is better) required to reach 25 %, 50 %, and 75 % of the oracle baseline (DrQ-v2), following the protocol of Mai et al. (2022).

| ENV | PERCENTILE | SVEA | SGQN | SRM | DR. G | DREAMERV3 | ViGMO (OURS) | TD-MPC2 |
|---|---|---|---|---|---|---|---|---|
| DMC SUITE | 25% | **152** | 415 | 195 | 205 | 220 | 160 | 70 |
| | 50% | 315 | 618 | 415 | 358 | 382 | **295** | 187 |
| | 75% | 573 | 827 | 657 | 430 | 512 | **423** | 327 |
| ROBOSUITE | 25% | 1140 | 1870 | 1335 | 1225 | 1695 | **700** | 325 |
| | 50% | 1560 | 1975 | 1670 | 1805 | 1785 | **1085** | 680 |
| | 75% | 1985 | 1995 | 1895 | 1905 | 1830 | **1375** | 945 |

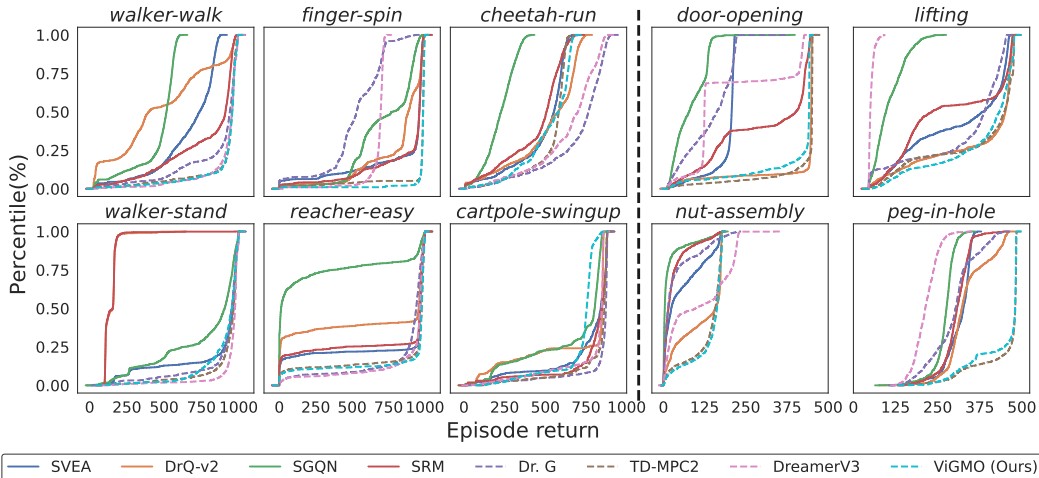

Figure 13: **Statistical comparison of sample efficiency.** Sample efficiency is evaluated across different tasks using ECDFs computed from training statistics. These ECDFs are used to compare the sample efficiency of all baselines reported in Table 7.

**Summary.** Overall, these results confirm that ViGMO's three core components—MA, LC, and ER—enable strong zero-shot generalization while maintaining high sample efficiency. In other words, ViGMO advances beyond conventional latent-space models by achieving superior generalization to unseen distractions while retaining their core advantage in sample efficiency.

### D.3 ABLATION STUDY ON MIXED AUGMENTATIONS

A central design choice in ViGMO is the MA strategy, which applies weak augmentations to all samples and strong augmentations only to a subset, thereby producing a structured mixture of weak-only and weak-to-strong samples. This design balances the stability of weak augmentations with the robustness of strong ones, while avoiding the computational overhead of encoding two full views per sample.

**Ablated variants.** To assess the importance of this mixture, we compare ViGMO against three ablated strategies: (i) a `weak-only` variant (WO), which applies only weak augmentations (*random-shift*) to every sample, (ii) a `strong-only` variant (SO), which applies only strong augmentations (*random-overlay*) to every sample, and (iii) a `weak-to-strong-only` variant (WTSO), which applies both weak and strong augmentations sequentially to every sample. Table 8 summarizes their performance in zero-shot generalization, while Figure 14 presents their sample efficiency curves.

**Results on the DMC suite.** ViGMO achieves the highest overall performance, with an average of $771.7 \pm 131.1$ ($792.9 \pm 104.0$ on *easy* and $750.4 \pm 150.8$ on *hard*). The `weak-only` variant trains stably but generalizes poorly under stronger shifts, averaging $712.7 \pm 238.4$ (92% of ViGMO; $740.9 \pm 218.5$ on *easy*, $684.4 \pm 254.2$ on *hard*). The `strong-only` variant collapses almost entirely, reaching only $67.5 \pm 43.8$ (9% of ViGMO; $68.7 \pm 42.6$ on *easy*, $66.4 \pm 45.1$ on *hard*),

highlighting the destabilizing effect of excessive perturbations. The `weak-to-strong-only` variant shows partial improvement, 241.4±54.2 (31% of ViGMO; 247.5±54.1 on *easy*, 235.2±53.8 on *hard*), but still lags far behind. These results demonstrate that mixing weak-only and weak-to-strong samples, as in ViGMO, is essential to combine stable optimization with robustness.

**Results on Robosuite.** The Robosuite benchmark presents an even clearer separation. The `weak-only` variant fails almost entirely under severe perturbations, averaging $0.8 \pm 0.7$ (1% of ViGMO; $1.0 \pm 1.2$ on *easy*, $0.7 \pm 0.0$ on *hard/extreme*). The `strong-only` variant improves but remains highly unstable, $39.5 \pm 91.6$ (20% of ViGMO; $112.8 \pm 131.9$ on *easy*, $3.9 \pm 4.8$ on *hard*, $1.8 \pm 2.2$ on *extreme*). The `weak-to-strong-only` variant achieves moderate robustness with $91.1 \pm 179.3$ (46% of ViGMO; $266.1 \pm 225.5$ on *easy*, $2.1 \pm 1.9$ on *hard*, $5.0 \pm 16.7$ on *extreme*), but still falls short of ViGMO. In contrast, ViGMO provides substantially stronger generalization with an average of $199.0 \pm 235.5$ ($374.2 \pm 207.3$ on *easy*, $123.5 \pm 204.6$ on *hard*, $99.3 \pm 193.6$ on *extreme*), underscoring that the mixed-augmentation design—rather than relying exclusively on weak or strong augmentations—is critical for achieving robust performance.

**Summary.** Taken together, these results confirm that ViGMO's MA strategy is essential for achieving both efficiency and robustness. The `weak-only` variant provides efficiency but lacks robustness, the `strong-only` variant introduces robustness but sacrifices stability and efficiency, and the `weak-to-strong-only` variant partially improves robustness but remains inefficient. Only the MA strategy successfully combines the strengths of both augmentation types, yielding the best efficiency–robustness trade-off across both the DMC suite and Robosuite.

Table 8: **Zero-shot generalization scores.** Ablation on mixed augmentations. Reported values are mean episode returns averaged across all distraction levels for each task in the DMC suite and Robosuite.

| TASK | DIFFICULTY | WEAK-ONLY | STRONG-ONLY | WEAK-TO-STRONG-ONLY | ViGMO (Ours) |
|---|---|---|---|---|---|
| | EASY | $740.9 \pm 218.5$ | $68.7 \pm 42.6$ | $247.5 \pm 54.1$ | $\mathbf{792.9 \pm 104.0}$ |
| CARTPOLE-SWINGUP | HARD | $684.4 \pm 254.2$ | $66.4 \pm 45.1$ | $235.2 \pm 53.8$ | $\mathbf{750.4 \pm 150.8}$ |
| | AVERAGE | $712.7 \pm 238.4$ (92%) | $67.5 \pm 43.8$ (9%) | $241.4 \pm 54.2$ (31%) | $\mathbf{771.7 \pm 131.1}$ **(100%)** |
| | EASY | $1.0 \pm 1.2$ | $112.8 \pm 131.9$ | $266.1 \pm 225.5$ | $\mathbf{374.2 \pm 207.3}$ |
| DOOR-OPENING | HARD | $0.7 \pm 0.0$ | $3.9 \pm 4.8$ | $2.1 \pm 1.9$ | $\mathbf{123.5 \pm 204.6}$ |
| | EXTREME | $0.7 \pm 0.0$ | $1.8 \pm 2.2$ | $5.0 \pm 16.7$ | $\mathbf{99.3 \pm 193.6}$ |
| | AVERAGE | $0.8 \pm 0.7$ (1%) | $39.5 \pm 91.6$ (20%) | $91.1 \pm 179.3$ (46%) | $\mathbf{199.0 \pm 235.5}$ **(100%)** |

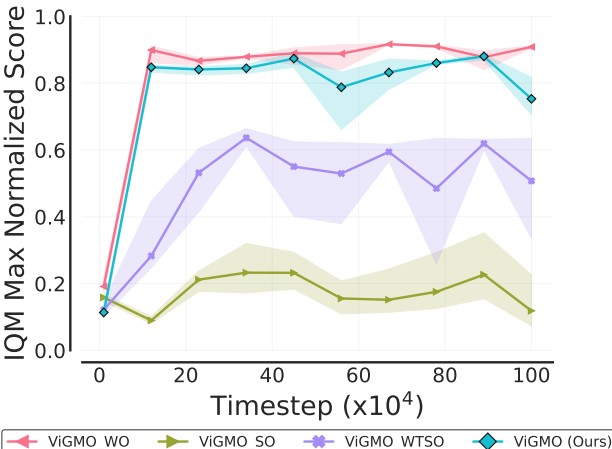

Figure 14: **Sample efficiency performance.** Comparison of ViGMO's MA strategy with `weak-only` (WO), `strong-only` (SO), and `weak-to-strong-only` (WTSO) variants. Both the MA strategy (ViGMO) and WO achieve strong sample efficiency compared with SO and WTSO, while only ViGMO provides superior zero-shot generalization.

## D.4  ABLATION STUDY ON DESIGN CHOICES

To better understand the individual contributions of ViGMO's design choices, we conduct an ablation study across diverse evaluation types and tasks. Figures 15 and 16 present comprehensive quantitative comparisons, illustrating how each design choice affects zero-shot generalization and sample efficiency.

**Ablated variants.**  We evaluate three modified versions of ViGMO: (i) `ViGMO_CONST_AUG`, which applies a fixed augmentation strategy without dynamically resampling augmentations at each step; (ii) `ViGMO_CONV`, which replaces the strong augmentation with a random convolution operator that primarily perturbs color statistics; and (iii) `ViGMO_CURL`, which replaces ViGMO's ER objective with the contrastive CURL loss.

These variants allow us to isolate the effects of dynamic augmentation, the choice of strong augmentation, and the auxiliary loss formulation.

**Zero-shot generalization.**  Figure 15 reports zero-shot generalization performance across all distraction types. Overall, ViGMO exhibits a smoother and more monotonic degradation curve (*original → easy → hard*) compared with the ablated baselines, demonstrating stronger robustness to distribution shifts. Among the variants, `ViGMO_CONV` performs competitively on color-related distractions such as *background-color*, *light-color*, and *object-color*, consistent with the color-invariance bias induced by random convolution. However, its performance drops sharply on tasks such as *background-video*, where perturbations alter spatial or contextual cues rather than color. In contrast, `ViGMO_CURL` consistently underperforms, failing to maintain robustness across most types of distraction.

**Sample efficiency and aggregate metrics.**  Figure 16 provides a comprehensive performance comparison across multiple metrics. The top row, which shows aggregate statistics (median, IQM, mean, and optimality gap), demonstrates that while ViGMO ranks second on the median score, it consistently achieves the best performance across IQM, mean, and optimality gap. This underscores its strong overall advantage over the ablated variants. The bottom row further reinforces these findings. The performance profile (bottom-left) confirms ViGMO's dominance across the full distribution of normalized scores, while the learning curve (bottom-right) clearly shows that ViGMO retains the hallmark sample efficiency of its TD-MPC2 backbone despite training under visually perturbed conditions. These results highlight that our gains in robustness do not come at the cost of efficiency.

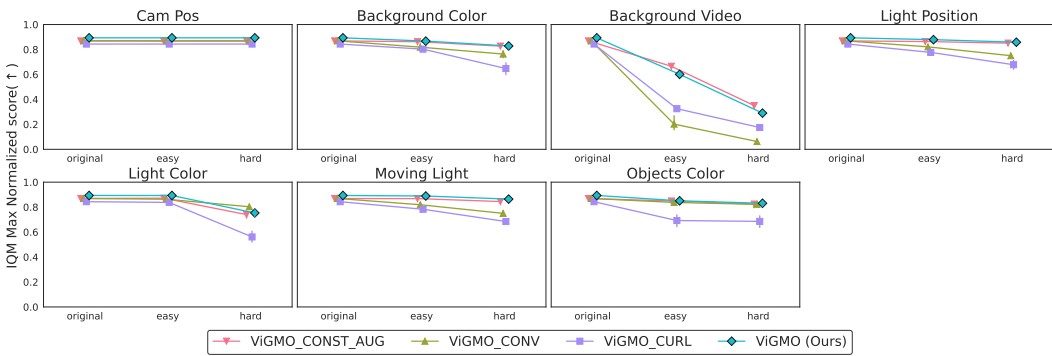

Figure 15: **Zero-shot generalization performance across distraction types for ablated variants.** This figure follows the structure of Figure 10 but compares ablated versions of ViGMO described in Section 4.3. ViGMO shows more stable and monotonic performance across difficulty levels than its ablated counterparts.

**Qualitative comparison of strong augmentations.**  To complement the quantitative comparisons, Figure 17 illustrates the qualitative effects of different strong augmentations in the *walker-walk* task. The *random-overlay* augmentation (used in ViGMO) blends natural images into the background, thereby introducing realistic domain shifts that mimic variations encountered in practice. In

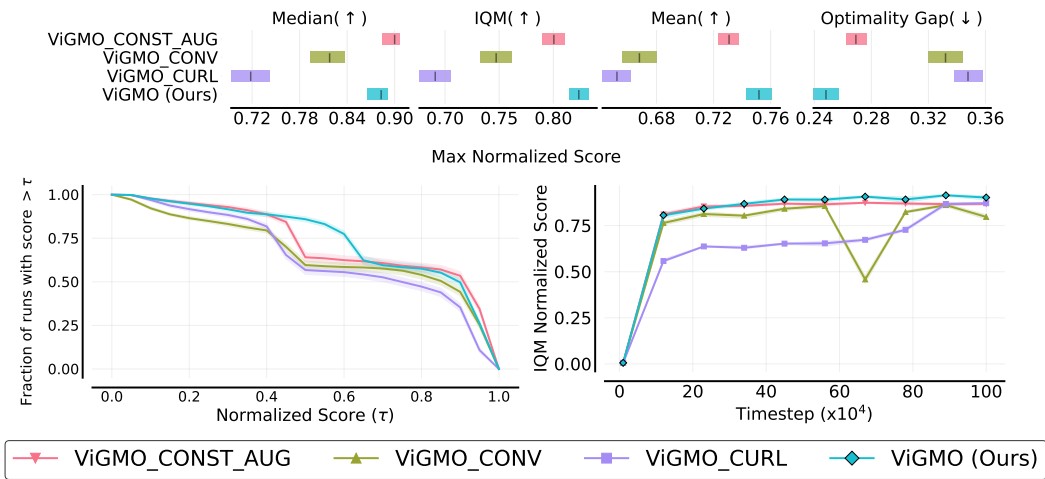

Figure 16: **Ablation study on design choices. Top:** Zero-shot generalization performance. **Bottom Left:** Performance profile. **Bottom Right:** Learning curves (sample efficiency). Across zero-shot generalization and sample efficiency, ViGMO consistently outperforms its ablated variants (`ViGMO_CONST_AUG`, `ViGMO_CONV`, `ViGMO_CURL`), highlighting the importance of each design choice. Shaded regions denote the 95% Stratified Bootstrap CIs.

contrast, the *random-conv* augmentation (used in `ViGMO_CONV`) applies convolutional filters that predominantly alter color statistics without introducing meaningful structural changes. This distinction explains why `ViGMO_CONV` performs well on color-based perturbations but fails to generalize to more complex distribution shifts involving spatial or contextual variations.

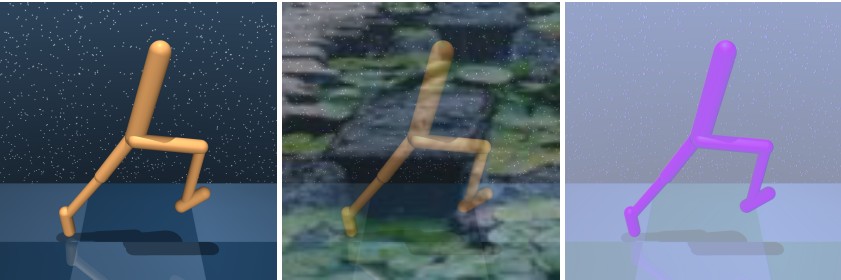

Figure 17: **Example images of strong augmentation methods.** Visualization of augmentation effects in the *walker_walk* task. (Left) Original image; (Center) *random-overlay* augmentation used in ViGMO; (Right) *random-conv* augmentation used in `ViGMO_CONV`.

**Summary.** Overall, these ablations confirm that each design choice—dynamic augmentation, the choice of strong augmentation, and the auxiliary loss formulation—plays an indispensable role. Removing or altering any of them leads to clear performance degradation, demonstrating that these choices collectively enable ViGMO to achieve robust and reliable zero-shot generalization.

## D.5 LATENT-SPACE CONSISTENCY ANALYSIS VIA EMBEDDING VISUALIZATION

A central hypothesis of our work is that robust generalization in MBRL requires the learned world model to produce *consistent* latent trajectories even when observations are perturbed by unseen distractions. If latent rollouts diverge under such perturbations, the learned dynamics can collapse, undermining planning and policy transfer. This subsection provides a detailed analysis of latent-space consistency, complementing the quantitative results reported in the main paper.

**Experimental setup.** We compare ViGMO with its backbone TD-MPC2 and the competitive baseline Dr. G. All agents are trained in clean environments and then evaluated zero-shot under distribution shifts. For each agent, we roll out the respective learned world models from identical initial inputs and record the predicted latent states $z_{t+1} = d_\theta(z_t, a_t)$ along with environment rewards $r_t$. This procedure ensures a fair comparison, as differences in trajectories arise purely from model robustness rather than input variation.

**Results.** To visualize high-dimensional latent rollouts, we apply UMAP (McInnes et al., 2018) to project trajectories into two dimensions, with faded markers indicating earlier time steps. To complement this embedding analysis, we additionally report episode returns and provide environment snapshots across evaluation levels (*original*, *easy*, *hard*). This three-part visualization—embeddings, returns, and snapshots—offers a holistic view of how perturbations affect both internal dynamics and external performance.

Figure 18 presents the results for the *walker_walk* task under *background-color*, *background-video*, and *light-color* perturbations. ViGMO maintains a single, compact latent manifold across all difficulty levels, consistently aligning perturbed rollouts with their clean counterparts. This structural stability directly translates into stable task returns and visually successful executions, as confirmed by the snapshots. In contrast, Dr. G and TD-MPC2 exhibit scattered or divergent manifolds when exposed to perturbations, leading to significant return degradation and frequent task failures. These findings demonstrate that ViGMO not only stabilizes latent-space dynamics but also mitigates cumulative error propagation over long horizons.

**Summary.** Overall, these results provide strong empirical evidence for our hypothesis (Figure 1) that mapping OOD inputs onto the in-domain latent manifold is essential for zero-shot generalization. By jointly integrating MA, LC, and ER, ViGMO enforces both temporal and structural consistency in latent space. This capability, absent in prior MBRL baselines, explains why ViGMO sustains robust performance across unseen perturbations without sacrificing the hallmark sample efficiency of latent-space MBRL.

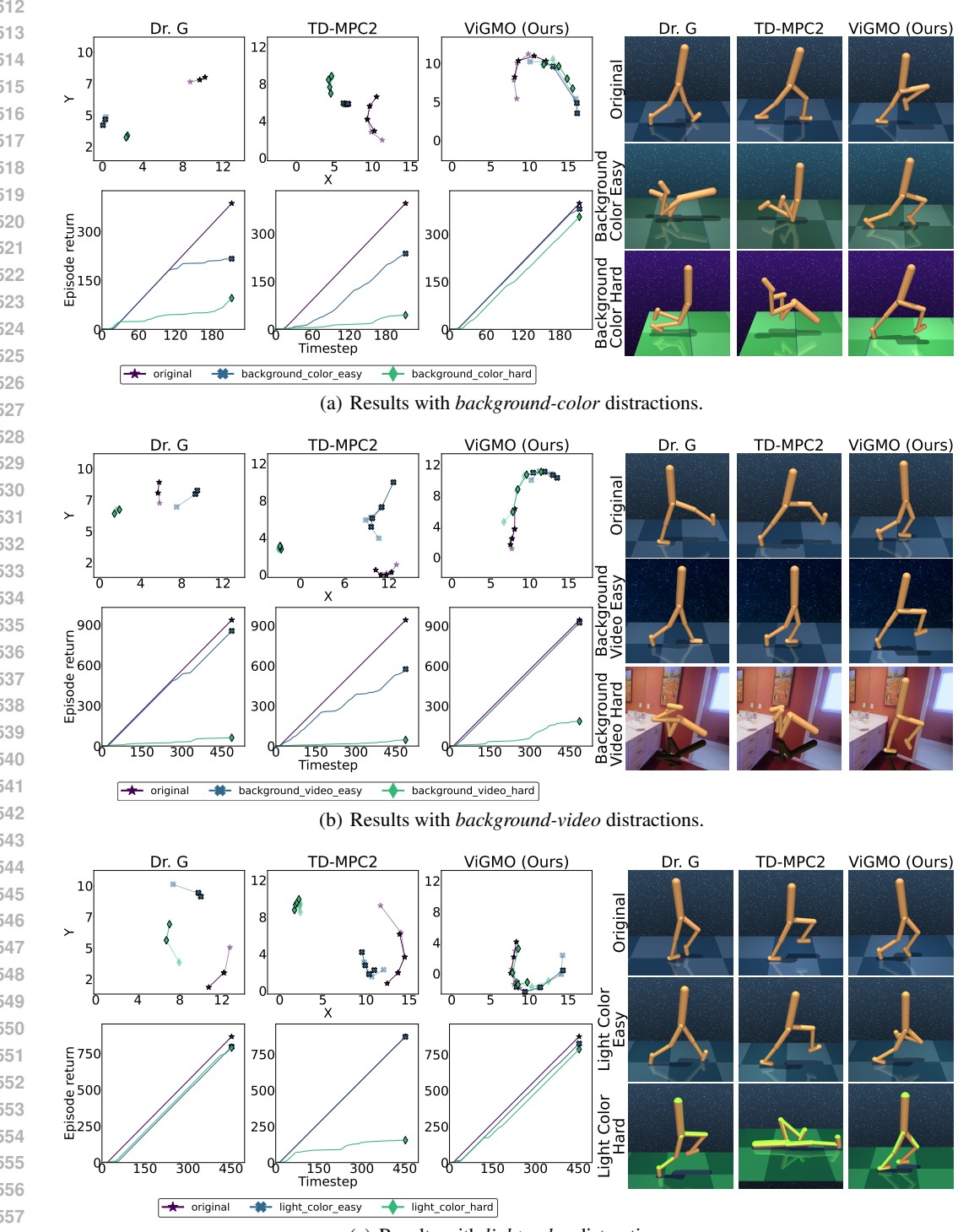

(a) Results with *background-color* distractions.

(b) Results with *background-video* distractions.

(c) Results with *light-color* distractions.

Figure 18: **Latent-space consistency analysis under visual perturbations. Left:** UMAP projections of latent embeddings (top) and episode returns (bottom) for the *walker_walk* task with three distractions, including *background-color*, *background-video*, and *light-color*. **Right:** Environment snapshots for each algorithm (ViGMO, TD-MPC2, Dr. G) across three evaluation types: *original*, *easy*, and *hard*. Markers denote task difficulty: ⋆ for *original*, × for *easy*, and ⋄ for *hard*. Arrows indicate temporal progression within each rollout segment $s_{t:t+H}$. ViGMO maintains consistent latent structures and stable performance across difficulty levels, whereas TD-MPC2 and Dr. G exhibit divergence and degraded task execution under perturbations.

## D.6  WALL-CLOCK TRAINING EFFICIENCY

To complement the sample-efficiency results in the main paper, we evaluate the wall-clock training cost of ViGMO relative to TD-MPC2 under the identical RL-ViGen training protocol. All experiments were conducted on a single NVIDIA A5000 (24GB) GPU using the same dataloaders, environment settings, and hyperparameters described in Appendices B and C.

Across all six DMC tasks and four Robosuite tasks, ViGMO closely matches TD-MPC2 in sample efficiency. As shown in Figure 19 (left), the normalized returns as a function of environment steps are nearly identical, and both methods reach their asymptotic performance at similar sample budgets. This confirms that ViGMO preserves the hallmark sample-efficiency benefits of latent-space MBRL.

Figure 19 (right) analyzes performance as a function of wall-clock time aggregated across all tasks. While TD-MPC2 trains slightly faster during the early stages, the difference remains modest throughout training. Over a 24-hour window, the average normalized-score gap is approximately 0.15, and importantly, this gap steadily decreases as training progresses. The observed overhead is consistent with expectations: strong augmentations introduce an auxiliary preprocessing cost (e.g., background-video composition and external-image loading), whereas LC and ER operate purely in the latent space and thus add negligible computational load.

A per-task breakdown in Figure 20 confirms this trend across all ten environments. TD-MPC2 maintains a small speed advantage in absolute wall-clock time, but ViGMO remains close throughout training, and the difference narrows as both methods converge to the asymptotic performance. Crucially, ViGMO achieves substantially stronger zero-shot robustness despite this minor overhead, demonstrating that the added consistency mechanisms improve generalization without compromising practical training speed.

Table 9 summarizes the difference in IQM normalized returns (TD-MPC2 minus ViGMO) at fixed wall-clock checkpoints. Most values remain small, and the differences diminish toward the later training stages—for example, at hour 21 the gap drops to 0.06—highlighting that ViGMO's overhead becomes less pronounced over time.

Overall, these results show that ViGMO introduces only a small wall-clock overhead relative to TD-MPC2, which continues to shrink as training proceeds. ViGMO therefore preserves practical training efficiency while delivering significantly improved zero-shot robustness.

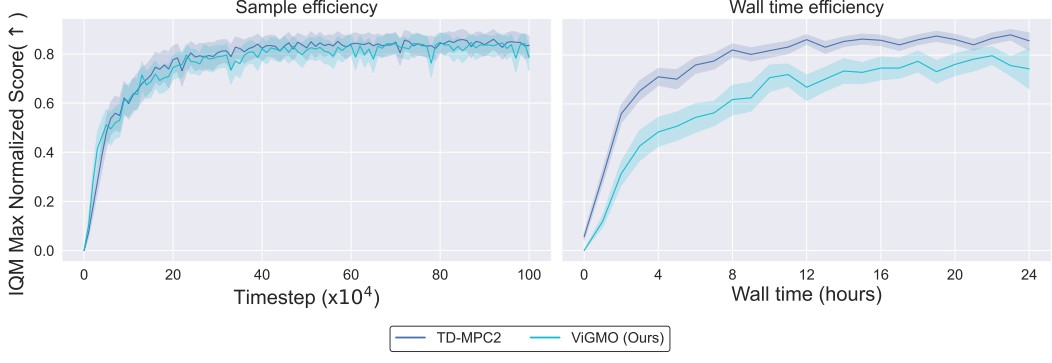

Figure 19: **Sample efficiency and wall-clock training efficiency of ViGMO vs. TD-MPC2 across all DMC and Robosuite tasks. Left:** sample-efficiency learning curves show that ViGMO closely matches TD-MPC2 throughout training. **Right:** wall-clock learning curves indicate that TD-MPC2 is marginally faster, but the gap remains consistently small (approximately 0.15 over 24 hours) and gradually narrows as training progresses, confirming that ViGMO preserves practical training efficiency while providing substantially improved robustness.

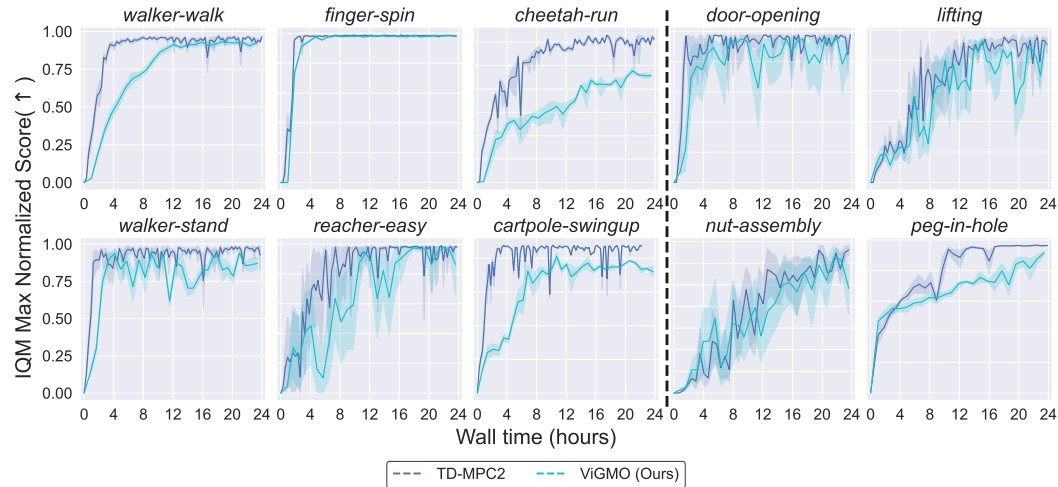

Figure 20: **Per-task wall-clock training efficiency across DMC and Robosuite.** Each subplot reports IQM max-normalized returns as a function of wall-clock time for TD-MPC2 and ViGMO. Across all ten tasks, TD-MPC2 is marginally faster in wall-clock time, but the gap remains consistently small throughout training. Despite this slight overhead, ViGMO achieves substantially improved zero-shot robustness while maintaining practical training efficiency.

Table 9: **Wall-clock training efficiency scores for TD-MPC2 vs. ViGMO.** Each entry reports the difference in IQM normalized returns (TD-MPC2 minus ViGMO) at fixed wall-clock checkpoints across all DMC and Robosuite tasks. Values near zero indicate matched training speed, while positive values represent a slight advantage for TD-MPC2. Although the 24-hour average difference is approximately 0.15 (reflecting a modest overhead for ViGMO), the gap gradually narrows over the course of training.

| WALL-CLOCK TIME (HOURS) | IQM NORMALIZED SCORE DIFFERENCE |
|---|---|
| **0** | **0.06** |
| 1 | 0.18 |
| 2 | 0.24 |
| 3 | 0.23 |
| 4 | 0.23 |
| 5 | 0.19 |
| 6 | 0.21 |
| 7 | 0.21 |
| 8 | 0.20 |
| 9 | 0.18 |
| 10 | 0.11 |
| 11 | 0.11 |
| 12 | 0.19 |
| 13 | 0.13 |
| 14 | 0.12 |
| 15 | 0.14 |
| 16 | 0.11 |
| 17 | 0.10 |
| 18 | 0.09 |
| 19 | 0.15 |
| 20 | 0.10 |
| **21** | **0.06** |
| 22 | 0.07 |
| 23 | 0.13 |
| 24 | 0.11 |
| AVERAGE | 0.15 |

## D.7 Generalization to Unseen Camera Viewpoints

The camera-position shifts included in RL-ViGen alter the viewpoint primarily through translations while preserving the base camera orientation. To investigate whether models trained with appearance-focused perturbations can generalize to fundamentally different geometric changes, we evaluate all methods on previously unseen camera rotations that modify projection geometry and the relative orientation between the agent and the camera.

**Viewpoint-rotation settings.** We consider two rotational viewpoint distractions—*camera-orientation* (`cam-ori`) and *camera-dynamic* (`cam-dyn`)—introduced in DMControl-GB (Hansen & Wang, 2021):

- **`cam-ori` (static rotation).** At the start of each episode, the tracking camera's spherical parameters $c = (\phi, \theta, r, \theta_{\text{roll}})$ are sampled uniformly within bounds controlled by a difficulty parameter $\beta_{\text{cam}} \in [0, 1]$ (easy: 0.3; hard: 0.8). Larger $\beta_{\text{cam}}$ values permit wider ranges of rotation and distance (e.g., $0 \leq \phi, \theta, \theta_{\text{roll}} \leq \frac{\pi}{2}$ and $0.5 \, r_{\text{original}} \leq r \leq 2.5 \, r_{\text{original}}$), producing substantial viewpoint deviations without temporal variation.

- **`cam-dyn` (dynamic rotation).** Camera parameters are initialized as in `cam-ori` and then evolve via a bounded random walk. Camera velocities are sampled as

$$v_0 \sim \mathcal{U}(v_{\min}, v_{\max}), \quad v_n = v_0 + \sum_{j=1}^{n} \mathcal{N}(0, \sigma\Sigma),$$

  generating continuous, temporally coherent viewpoint drift throughout the episode. Motion magnitudes scale with $\beta_{\text{cam}}$, producing time-varying geometric distortions.

As illustrated in Figure 21, these viewpoint perturbations introduce substantial geometric distortions—altering perspective, foreshortening limbs, reordering depth relationships, and in some cases causing the agent to become partially or entirely outside the field of view (e.g., *cartpole-swingup*).

**Zero-shot evaluation.** All models (ViGMO and baselines) are trained strictly under the RL-ViGen protocol—*without* any viewpoint-rotation augmentations—and evaluated zero-shot on the two viewpoint-shift settings.

**Results.** As shown in Figure 22, performance drops sharply across all algorithms. None of the evaluated methods maintains reliable zero-shot control under these rotations. This degradation occurs because rotational viewpoint shifts induce geometric transformations that fundamentally differ from appearance shifts (e.g., lighting, background, color). Such transformations violate the pixel-level invariances learned by current encoder-based MBRL and MFRL frameworks, including ViGMO. Although ViGMO was designed to be robust to appearance-based perturbations (e.g., background video, lighting, color, camera translation), the results here show that geometric viewpoint changes pose a significantly harder challenge. Developing geometry-aware latent representations, multi-view–consistent world models, or scene-structured encoders remains an important direction for future work.

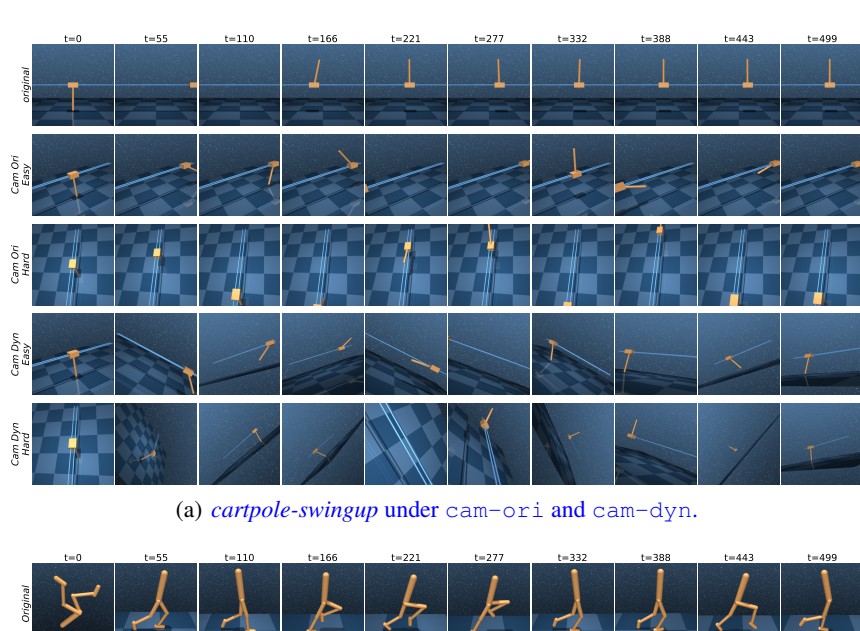

(a) *cartpole-swingup* under `cam-ori` and `cam-dyn`.

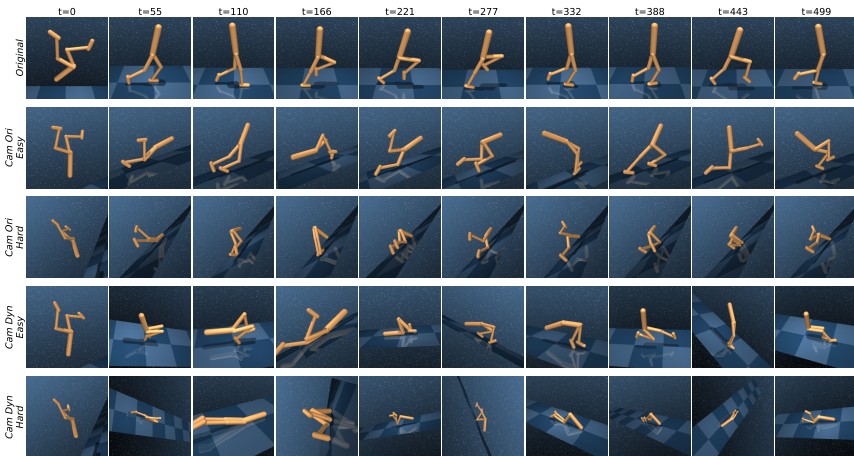

(b) *walker-walk* under `cam-ori` and `cam-dyn`.

Figure 21: **Pixel-level effects of camera-viewpoint distractions.** Example frames from *cartpole-swingup* and *walker-walk* under the `cam-ori` and `cam-dyn` settings. These settings introduce geometric distortions, changes in projection, and in extreme cases remove the agent partially or fully from view. Such shifts create substantially different visual observations from those encountered during training, explaining the difficulty of viewpoint-rotation generalization for pixel-based RL agents.

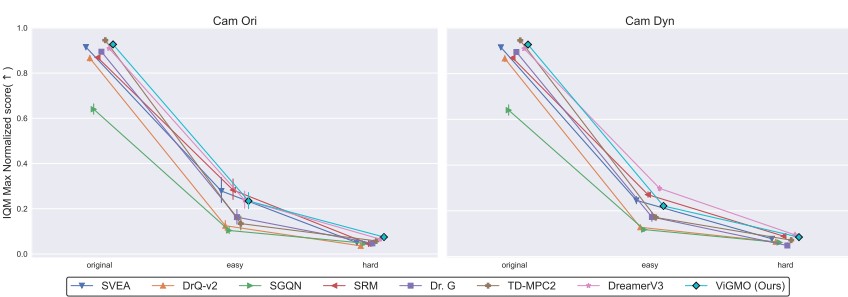

Figure 22: **Zero-shot generalization to unseen sensor noises.** We evaluate ViGMO and baselines on environments rendered under camera-viewpoint rotations using the `cam-ori` and `cam-dyn` settings. All models are trained without viewpoint-rotation augmentations. Performance drops sharply for all methods due to substantial geometric distortions in projection and depth ordering, indicating that reliable zero-shot generalization under viewpoint changes remains a challenging problem.

## D.8 GENERALIZATION TO SENSOR NOISE

Real-world visual sensing often suffers from sensor-level degradations such as blur and pixel noise, which differ fundamentally from the appearance- and context-based perturbations included in RL-ViGen. To evaluate whether ViGMO extends beyond these appearance shifts, we conduct zero-shot evaluations under two canonical sensor-noise corruptions: `normal-blur` and `gaussian-noise`. These perturbations remove high-frequency detail or introduce pixel-wise stochastic distortions, resulting in a distribution shift not encountered during training.

**Sensor-noise settings.**

- **`normal-blur`** (mean filter). Each frame is blurred using a normalized local averaging filter. The difficulty level is controlled by a blur coefficient $\beta_{\text{blur}}$ (easy: 20, hard: 10). The resulting kernel size $(H_K, W_K)$ is computed as

$$H_K = \begin{cases} H/\beta_{\text{blur}}, & \text{if } H/\beta_{\text{blur}} \text{ is odd} \\ H/\beta_{\text{blur}} + 1, & \text{otherwise} \end{cases}, \quad W_K = \begin{cases} W/\beta_{\text{blur}}, & \text{if } W/\beta_{\text{blur}} \text{ is odd} \\ W/\beta_{\text{blur}} + 1, & \text{otherwise.} \end{cases}$$

  Larger kernels progressively remove mid- and high-frequency structure from the input.

- **`gaussian-noise`** (white noise). Zero-mean Gaussian noise is added independently to each pixel, with standard deviation $\sigma$ (easy: 0.5, hard: 1.0). This corruption injects high-frequency sensor-like perturbations without altering the underlying geometry.

**Qualitative effects.** Figure 23 visualizes rollouts on two representative DMC tasks—*cartpole-swingup* and *walker-walk*—across noise types and difficulty levels. `normal-blur` obscures limb boundaries and motion traces, while `gaussian-noise` introduces dense speckle patterns across the image. Although the agent remains visible, the loss of edges and texture alters observable cues required for control.

**Results.** Figure 24 summarizes normalized returns under these noise settings. Three consistent trends emerge:

1. **Generalization-oriented methods degrade sharply.** Methods explicitly designed for visual generalization—SVEA, SGQN, SRM, Dr. G, and ViGMO—show large performance drops under both blur and noise, reflecting the mismatch between sensor-level corruptions and the appearance-based shifts (e.g., background, lighting, color) used during training.

2. **Task-specific methods retain high performance.** Interestingly, DrQ-v2, TD-MPC2, and DreamerV3 remain considerably more robust to blur and noise. This robustness likely stems not from true generalization, but from convolutional inductive biases and task-specific representations that remain functional when corruption does not alter scene geometry or dynamics.

3. **ViGMO is the strongest among generalization methods.** Although all generalization-focused methods degrade, ViGMO consistently achieves the highest performance within this group, indicating that its latent-consistency mechanism provides a degree of robustness even to perturbations outside its intended appearance-shift domain.

Overall, sensor noise represents a fundamentally different category of distribution shift—destroying pixel-level fidelity rather than altering contextual appearance. While ViGMO improves robustness to contextual and appearance perturbations, sensor-level corruptions remain challenging for all generalization methods. Developing latent-space models that unify robustness across appearance, geometry, and sensing degradations constitutes a promising direction for future work.

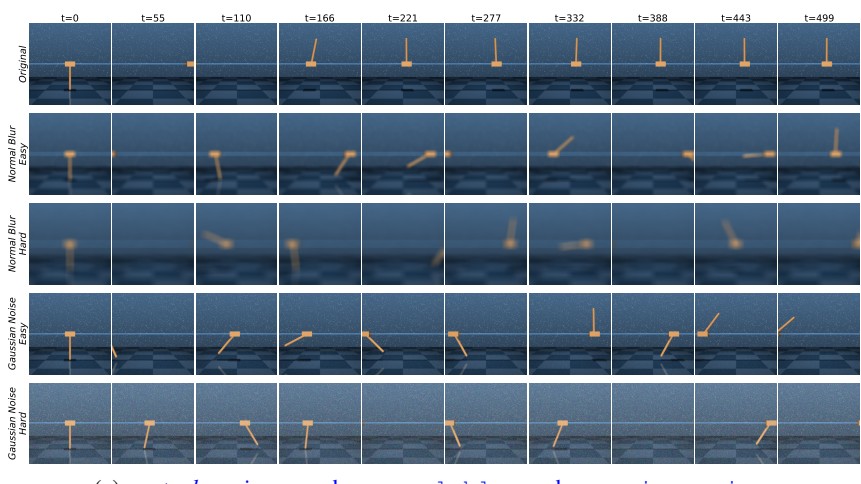

(a) *cartpole-swingup* under `normal-blur` and `gaussian-noise`.

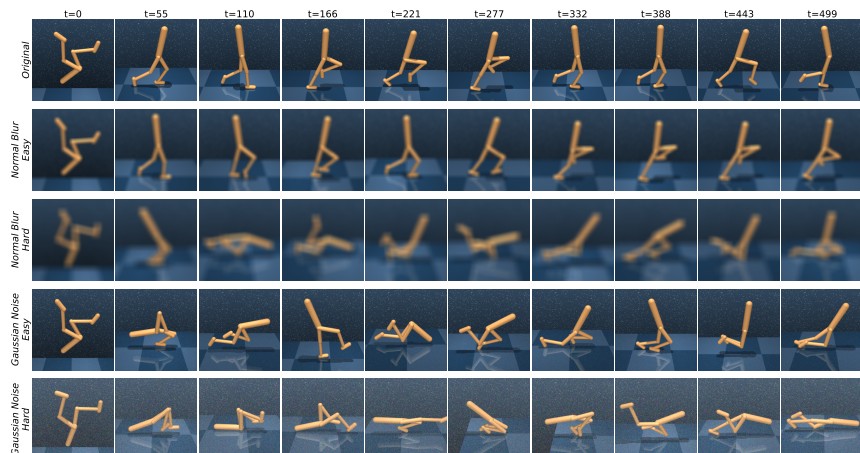

(b) *walker-walk* under `normal-blur` and `gaussian-noise`.

Figure 23: **Pixel-level effects of sensor-noise perturbations.** Example rollouts from *cartpole-swingup* and *walker-walk* under two sensor-noise settings—`normal-blur` and `gaussian-noise`—at *easy* and *hard* severity levels. `normal-blur` removes mid- and high-frequency details, obscuring object boundaries and motion cues, while `gaussian-noise` introduces dense pixel-wise perturbations across the frame. These sensor-level corruptions significantly alter the visual input without altering scene geometry, illustrating a distribution shift not captured by the appearance-based distractions used during training.

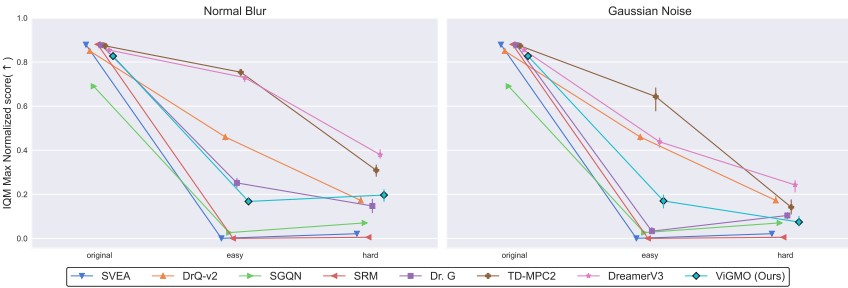

Figure 24: **Zero-shot generalization to unseen sensor noises.** Normalized returns for all baselines under `normal-blur` and `gaussian-noise`. Task-specific methods (DrQ-v2, TD-MPC2, DreamerV3) maintain relatively high robustness, whereas generalization-oriented methods (SVEA, SGQN, SRM, Dr. G, ViGMO) degrade substantially. Among the latter, ViGMO consistently achieves the highest robustness.

## D.9 ROBUSTNESS ACROSS DIVERSE AUGMENTATION FAMILIES

Our main experiments use *random-overlay* as the strong augmentation within the MA+LC+ER mechanism. While overlay-based perturbations match the contextual and background-driven distractions present in RL-ViGen, it remains important to determine whether ViGMO's robustness depends on this specific augmentation family or whether the underlying latent-consistency mechanism provides a more general inductive bias that extends across distinct augmentation types.

**Experimental setup.** To isolate the effect of the augmentation family, we construct two additional ViGMO variants that differ only in their choice of strong augmentation, while keeping the shared encoder, dynamics model, and MA+LC+ER structure identical:

- `ViGMO_CONV`: uses *random-conv* as the strong augmentation, perturbing local texture and color statistics without modifying global scene layout.
- `ViGMO_SRM`: uses *Spectrum Random Masking (SRM)* as the strong augmentation, which masks contiguous Fourier-spectrum bands and removes mid/high-frequency image components.

Figure 25 visualizes the qualitative differences among these augmentations on the *walker_walk* task. Overlay introduces contextual background replacement, random-conv yields spatially smooth color/texture shifts, and SRM injects strong frequency-domain corruption. These visualizations highlight that the three augmentation families differ substantially in the kinds of perturbations they introduce.

**Results.** Figure 26 reports results for all three ViGMO variants, with the left panel showing sample-efficiency learning curves and the right panel presenting zero-shot generalization performance. Across both metrics, the default ViGMO (trained with *random-overlay*) achieves the highest overall performance. However, a crucial observation is that `ViGMO_CONV` and `ViGMO_SRM` also exhibit strong zero-shot robustness despite being trained with distinct augmentation families. All three variants maintain stable learning curves and achieve competitive zero-shot returns, demonstrating that ViGMO's reliable generalization does not hinge on the choice of *random-overlay*. Instead, the latent-consistency mechanism (MA + LC + ER) provides a general inductive bias that stabilizes the latent transition model across spatial, contextual, and spectral perturbation families. This augmentation-agnostic robustness highlights that ViGMO's effectiveness arises primarily from its latent consistency design rather than from any particular augmentation choice.

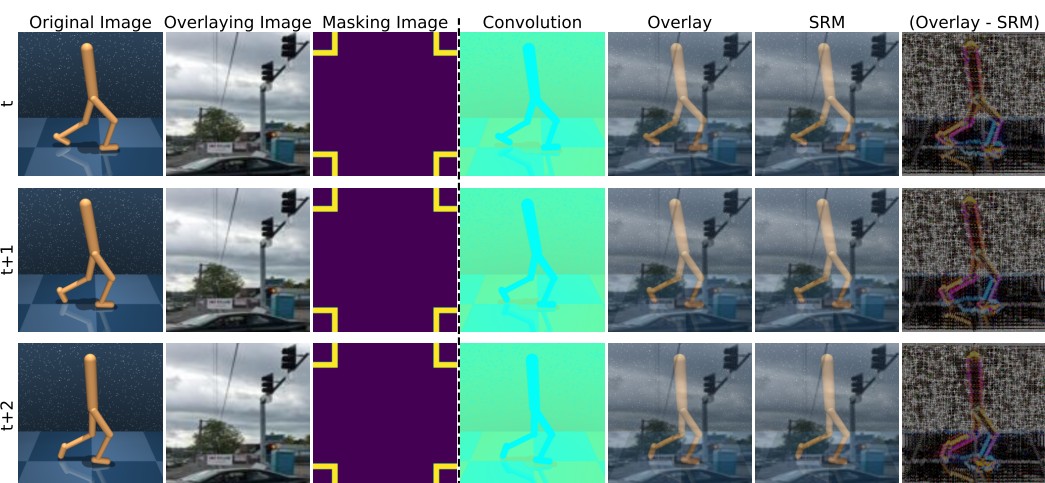

Figure 25: **Example images of strong augmentation methods.** Visualization of the three strong augmentation families—*random-overlay*, *random-conv*, and *SRM*—on the *walker_walk* task. Overlay introduces contextual background replacement, random-conv perturbs local texture and color statistics, and SRM masks contiguous frequency bands in the Fourier domain. The rightmost column visualizes the absolute pixel-wise difference between the overlay and SRM, highlighting the fundamentally different spectral distortions induced by SRM compared to those of contextual overlay-based perturbations.

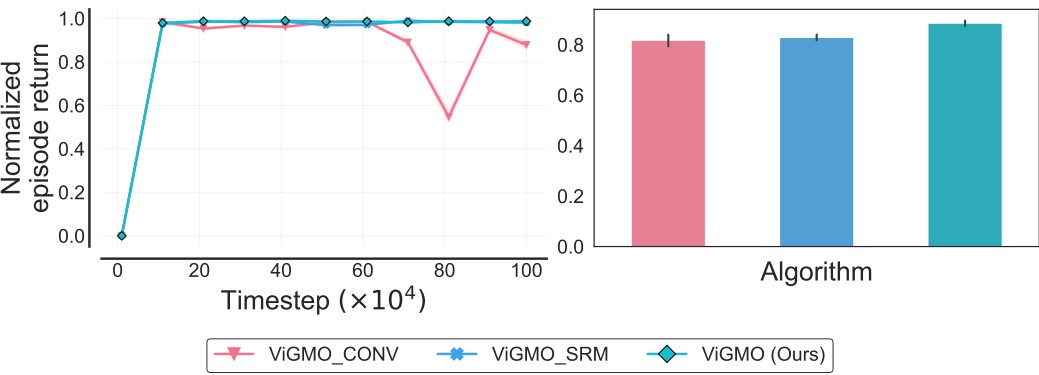

Figure 26: **Zero-shot robustness of ViGMO across diverse augmentation families.** We compare three ViGMO variants trained with different strong augmentations—`ViGMO_CONV` (*random-conv*), `ViGMO_SRM` (*SRM*), and ViGMO (Ours) (*random-overlay*). All variants maintain strong zero-shot robustness despite being trained with distinct augmentation families. These results indicate that ViGMO's latent-consistency mechanism (MA + LC + ER) provides augmentation-agnostic robustness beyond any specific strong augmentation choice.

## D.10 PERFORMANCE UNDER LONGER PLANNING HORIZONS

Our default experiments adopt a short MPC planning horizon of $H = 3$, following standard practice in latent-space MPC where short horizons offer a favorable trade-off between predictive stability, computational cost, and sample efficiency. To examine whether ViGMO's latent-consistency mechanism remains effective when planning over deeper imagined rollouts, we conduct additional comparisons under extended planning horizons.

**Experimental setup.** Evaluating long-horizon MPC substantially increases wall-clock training and inference time because each optimization step requires longer latent rollouts. Running $H = 5$ or $H = 10$ across the entire DMC and Robosuite tasks would therefore incur prohibitively high computational cost. For this reason, we evaluate the three horizons ($H = 3$, $H = 5$, and $H = 10$) on a single representative and challenging DMC task, *finger-spin*.

**Results.** Figure 27 shows learning curves for ViGMO under the three horizons. Across $H = 3$, $H = 5$, and $H = 10$, ViGMO maintains stable optimization and reaches similar asymptotic returns. As expected, increasing the planning horizon slows convergence because compounding prediction errors accumulate more severely over longer imagined trajectories—a well-documented limitation of MBRL. Nevertheless, ViGMO's latent-consistency regularization continues to stabilize the forward dynamics, preventing error accumulation from degrading policy learning even when the planning depth is significantly increased. These findings indicate that ViGMO's latent-consistency mechanism generalizes beyond the short-horizon setting, and remains effective under longer imagined rollouts.

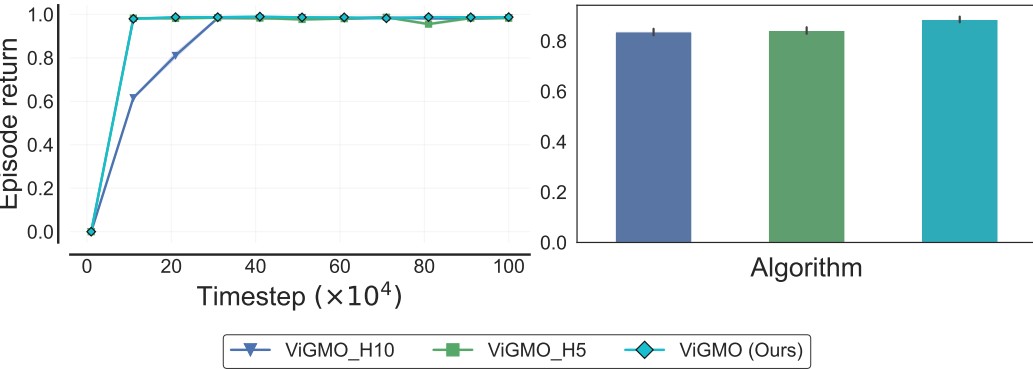

Figure 27: **Performance under extended planning horizons on *finger-spin*.** We compare ViGMO under planning horizons $H = 3$, $H = 5$, and $H = 10$ on the representative DMC task *finger-spin*. Despite increased susceptibility to compounding errors at longer horizons, ViGMO maintains stable learning dynamics and achieves similar asymptotic returns across all settings. These results suggest that latent consistency continues to effectively regularize the forward dynamics even under deeper imagined rollouts.

### D.11 COMPARISON OF RENDERING PIPELINES: RL-VIGEN VS. DMCONTROL-GB

RL-ViGen extends the DMControl-GB benchmark by introducing a broader set of appearance perturbations (e.g., background-video overlays, lighting motion, camera-position shifts, and synchronized object/background color variations). Because several of these perturbations are built on top of the background compositing procedure originally introduced in DMControl-GB, we conduct a detailed comparison of the two visual pipelines to identify potential discrepancies in their underlying rendering behavior.

To isolate the effect of the rendering pipeline alone, we extract identical unmodified foreground frames from the base DMControl environments and apply the exact background-video compositing procedures from both RL-ViGen and DMControl-GB. We then compute pixel-wise difference maps between the resulting composite images.

As shown in Figure 29, the composite frames produced by the two pipelines are nearly identical across time and tasks. The only consistent difference arises from the foreground-mask color employed prior to compositing: DMControl-GB employs a green foreground mask, whereas RL-ViGen uses a black foreground mask. This mask-color difference yields minor deviations in shadow regions (due to color spill) but does not alter the camera intrinsics, viewpoint, scene geometry, articulated motion, or the rendered foreground content. Aside from this masking-color variation, the pixel-level outputs remain almost indistinguishable, as confirmed by the near-zero difference maps.

Figure 28 further visualizes the pre-overlay foreground frames used by both benchmarks, clearly showing that the only visible distinction originates from the mask color. The actual rendered robot, lighting model, shading, and viewpoint remain identical.

Overall, these analyses confirm that RL-ViGen faithfully preserves the underlying rendering pipeline of DMControl-GB, while expanding the diversity and difficulty of appearance perturbations via additional background videos and visual-shift categories. RL-ViGen should therefore be viewed as a benchmark extension rather than a fundamentally different rendering procedure.

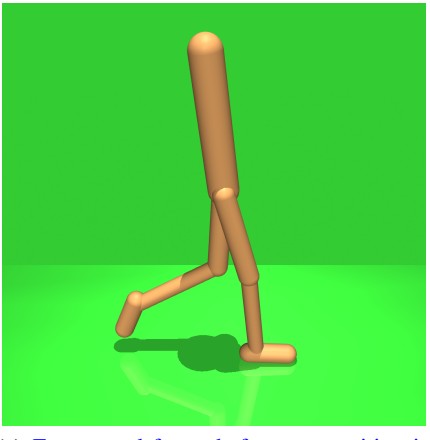 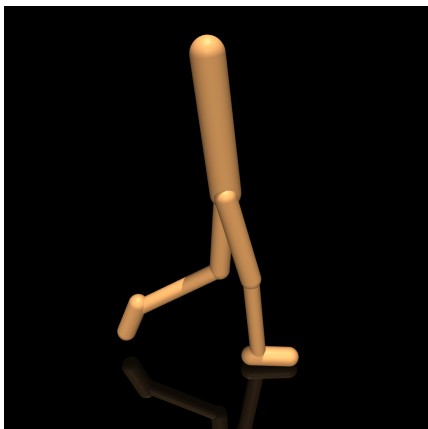

(a) Foreground frame before compositing in DMControl-GB.

(b) Foreground frame before compositing in RL-ViGen.

Figure 28: **Foreground masks used by DMControl-GB and RL-ViGen before background-video compositing.** DMControl-GB employs a green-screen mask, whereas RL-ViGen uses a black mask, resulting in minor differences in shadow regions but leaving the rendered content unchanged.

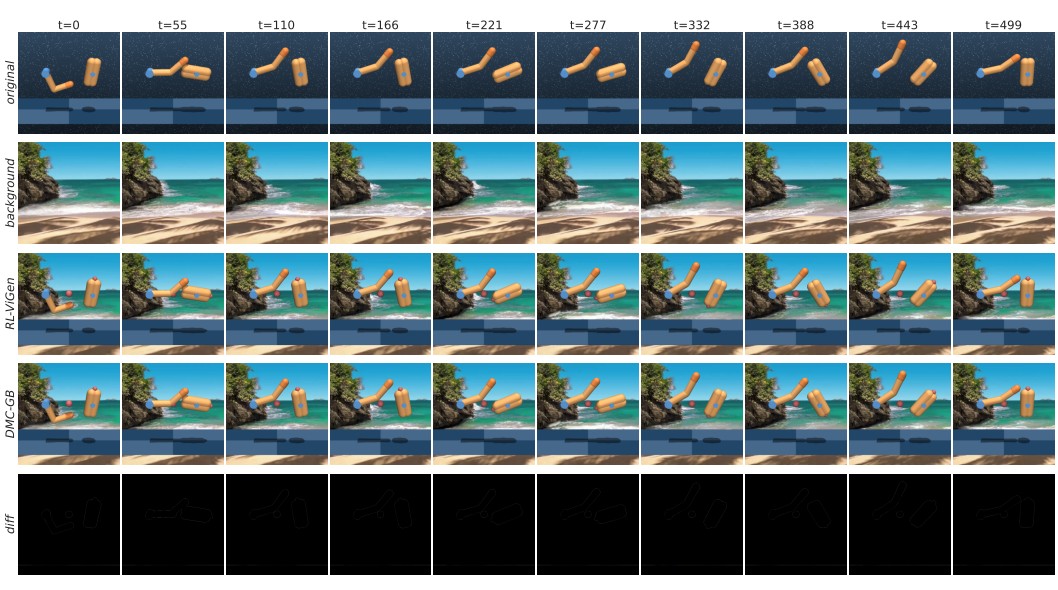

(a) *finger-spin* with *video-easy* distraction.

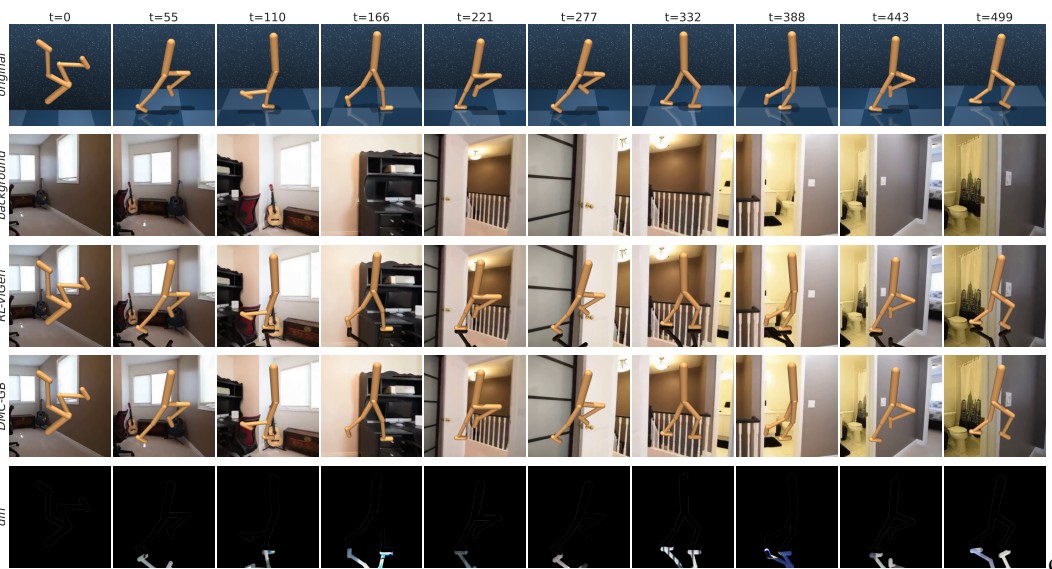

(b) *walker-walk* with *video-hard* distraction.

Figure 29: **Pixel-level comparison of RL-ViGen and DMControl-GB rendering pipelines.** For two representative tasks, the composite images produced by both pipelines are nearly identical. The only systematic difference arises from the foreground-mask color (green in DMControl-GB vs. black in RL-ViGen), which slightly affects shadow appearance but not camera geometry or scene content.

## D.12  ADDITIONAL EXPERIMENTS: COMPARISON WITH SMG (ZHANG ET AL., 2024)

To complement the baselines in the main paper, we also compare ViGMO with SMG (Zhang et al., 2024), a recent model-free method for visual generalization that separates the visual encoder from a lightweight auxiliary predictor. SMG improves representation robustness through this auxiliary prediction mechanism, but—unlike ViGMO—it does not learn a world model, perform latent imagination, or use its predictive module for planning.

**Experimental setup.**  We evaluate SMG using the authors' anonymized reference implementation[5] under the same RL-ViGen training protocol, environment configurations, and zero-shot evaluation settings used for all baselines. Following the evaluation protocol from the SMG paper, we report results on five representative DMC tasks: *cartpole-swingup*, *finger-spin*, *walker-stand*, *walker-walk*, and *cheetah-run*. This matched setup isolates architectural and algorithmic differences rather than variations in rendering or distractor distributions.

**Results.**  Figure 30 presents aggregated results across the five DMC tasks, with the left panel showing normalized sample-efficiency learning curves averaged over all tasks and the right panel reporting the corresponding aggregate zero-shot returns. Under identical RL-ViGen training and zero-shot evaluation settings, ViGMO consistently learns faster and achieves higher aggregate zero-shot robustness than SMG.

These differences arise naturally from the methodological distinction between the two approaches: SMG enhances visual representations within a model-free actor–critic framework, whereas ViGMO's latent-consistency mechanism (MA + LC + ER) stabilizes the planner-used world model and directly supports multi-step latent rollouts. Consequently, ViGMO benefits more from stable latent dynamics when facing the diverse appearance distractions present in RL-ViGen, while SMG remains competitive in its intended model-free regime.

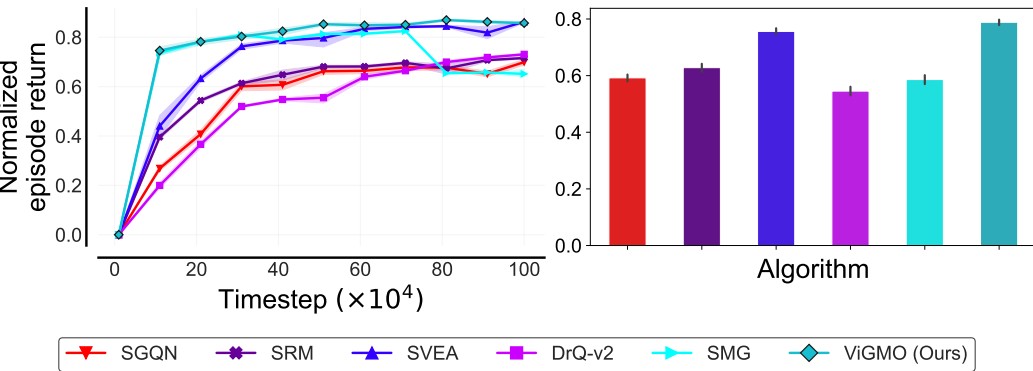

Figure 30: **Aggregated performance comparison with SMG across five DMC tasks—*cartpole-swingup*, *finger-spin*, *walker-stand*, *walker-walk*, and *cheetah-run*.** The left panel shows normalized sample-efficiency learning curves averaged across tasks, and the right panel presents zero-shot returns averaged across the same tasks. Under identical RL-ViGen training and zero-shot evaluation settings, ViGMO learns faster and achieves higher aggregate zero-shot robustness than SMG.

---

[5] https://anonymous.4open.science/r/SMG/

## E  IMPLICATIONS UNDER METHOD

In this section, we provide a detailed discussion of why ViGMO's three components—MA, LC, and ER—are necessary in the MBRL setting, how they are implemented, and how they contribute to both robustness and efficiency. While similar ideas have appeared in MFRL, directly applying them to latent-space MBRL is non-trivial due to the recursive nature of world model rollouts. We therefore highlight the design challenges and clarify how ViGMO overcomes them.

**Mixed weak-to-strong augmentation strategy.** Data augmentation is widely recognized as an effective way to improve robustness and sample efficiency in visual RL. However, most prior results come from value-based MFRL methods, where augmentations are applied one step at a time during Q-learning. In MBRL, rollouts span multiple prediction steps, so naïvely applying both weak and strong transformations to every frame can double computational cost and destabilize latent dynamics. ViGMO addresses this by splitting each mini-batch into two subsets: weak-only and weak-to-strong. Weak augmentations are implemented via *random-shift* (Yarats et al., 2022), while strong augmentations use *random-overlay* (Hansen & Wang, 2021), which blends the input frame with a task-irrelevant image (Zhou et al., 2017). This structured division ensures that the model sees both stable and heavily perturbed views without redundant computation. Empirically, this design maintains efficiency while providing sufficient exposure to diverse visual variations.

**Latent-consistency learning.** Learning reliable dynamics in latent space is especially challenging when targets are derived from noisy or heavily augmented inputs. Prior work has shown that high-variance targets hinder convergence in value learning (Mnih, 2013; He et al., 2020; Hansen et al., 2021). In MBRL, this problem is amplified because Q-functions and dynamics models are both conditioned on encoder outputs. To mitigate this, ViGMO enforces latent-consistency by always computing TD targets using weakly augmented successor states. Weak augmentations produce stable yet non-trivial latents (Yarats et al., 2021a; 2022; Hansen et al., 2022), which serve as reliable anchors for training. By aligning predictions from mixed latents with these weak targets, LC stabilizes rollouts and prevents error accumulation over long horizons. This preserves the strong sample efficiency of the backbone while improving robustness to distractions.

**Encoder regularization.** Even with MA and LC, the encoder itself may learn unstable features if it is not explicitly constrained. This issue arises when the agent encounters unseen distractions at test time: inconsistent encodings corrupt the initial latent states and, by extension, the entire rollout trajectory. Unlike the dynamics model, which is trained to enforce temporal consistency, the encoder lacks a direct mechanism to enforce cross-augmentation invariance. ViGMO introduces an auxiliary ER loss inspired by contrastive learning methods (Hansen & Wang, 2021). The ER loss explicitly aligns weak-only and weak-to-strong latents of the same frame. This regularization encourages the encoder to preserve task-relevant features (e.g., the agent's body or manipulated objects) while discarding nuisance factors (e.g., background textures, lighting variations). As a result, the encoder produces stable, task-focused representations that remain reliable under unseen perturbations, leading to higher-quality rollouts and stronger generalization.

**Summary.** MA provides structured exposure to both weak and strong views, LC stabilizes latent dynamics by anchoring predictions to weak targets, and ER ensures the encoder itself remains consistent under augmentations. These three components are complementary: removing any one of them leads to notable degradation in performance (see ablations in Section 4.3). Together, they allow ViGMO to retain the efficiency of latent-space MBRL while achieving substantially better zero-shot generalization under challenging visual shifts.

## F  MBRL Baseline Analysis

In this subsection, we provide an extended discussion of the MBRL baselines used in our study. Since ViGMO is designed as an augmentation, consistency, and regularization framework that can be integrated into existing latent-space MBRL methods, it is important to understand the characteristics of representative backbones. We therefore focus on two influential families of latent-space MBRL algorithms—**Dreamer** and **TD-MPC**—which are widely regarded as state-of-the-art in terms of sample efficiency and performance on continuous visuomotor control tasks.

**Dreamer family.**  The Dreamer family (Hafner et al., 2020; 2021; 2025) addresses long-horizon control from pixels by learning a *decoder-based* latent world model. Specifically, the model is trained not only on latent dynamics and reward prediction but also on pixel-level reconstruction, forcing the latent state to retain sufficient information for observation recovery. The initial latent state is inferred through a recurrent state-space model conditioned on past states, actions, and current observations. Actions are generated by a learned *actor policy* optimized in latent space using actor–critic RL, with the critic trained on imagined rollouts from the world model. DreamerV2 (Hafner et al., 2021) and DreamerV3 (Hafner et al., 2025)—scale up the architecture, introduce improved optimization strategies, and take advantage of modern accelerators. These advances have made the Dreamer family increasingly competitive on high-dimensional continuous control tasks.

**TD-MPC family.**  The TD-MPC family (Hansen et al., 2022; 2024) takes a different approach by discarding reconstruction losses and adopting a *decoder-free* latent world model. Instead of reconstructing observations, TD-MPC jointly learns latent dynamics, reward, value, and a policy under a temporal-difference (TD) objective, operating entirely in latent space. Unlike Dreamer, which trains a separate actor for long-horizon imagination in the learned world model, TD-MPC integrates its policy into an online planning scheme. At each step, a model predictive controller (MPC) searches over action sequences guided by Q-value predictions while using the learned policy as a prior, and executes the first action of the best trajectory. TD-MPC2 improves upon this formulation with architectural refinements and multi-task scalability, achieving state-of-the-art sample efficiency on continuous visuomotor control benchmarks.

**Comparison of the two families.**  Both Dreamer and TD-MPC families share the same core principle of operating in latent space by learning a transition model over compact representations. Their fundamental difference lies in how the latent model is trained and how actions are selected. Dreamer relies on a *decoder-based* world model and learns a parameterized actor policy via RL. TD-MPC, by contrast, uses a *decoder-free* world model and selects actions through MPC planning guided by Q-values. This distinction reflects two different philosophies: *decoder-based actor learning* (Dreamer) versus *decoder-free latent planning* (TD-MPC), with important implications for sample efficiency and generalization.

**Rationale for backbone selection.**  Since the central goal of ViGMO is to enhance visual generalization without sacrificing sample efficiency, it is essential to build upon a backbone that is already highly sample-efficient. Between the two families, Dreamer's reconstruction-driven formulation tends to increase model complexity and training cost, while often providing less favorable efficiency in practice. In contrast, TD-MPC's decoder-free design avoids unnecessary reconstruction objectives and focuses directly on value learning and planning in latent space, yielding stronger sample efficiency. As shown in Figure 31, our reproduced results are consistent with prior findings (Hansen et al., 2024): TD-MPC2 outperforms both the Dreamer family and its predecessor TD-MPC across diverse tasks in the DMC suite. For this reason, we adopt TD-MPC2 as the backbone of ViGMO.

**Preserving efficiency under unseen visual perturbations.**  Importantly, ViGMO's contribution is not merely inheriting TD-MPC2's efficiency, but demonstrating that this efficiency can be retained even under visually perturbed conditions—an open challenge in latent-space MBRL. The key difficulty lies in simultaneously maintaining high sample efficiency and achieving robustness to unseen distractions, which standard latent-space MBRL models typically fail to balance. ViGMO addresses this gap by introducing MA, LC, and ER, which together extend TD-MPC2 into a framework that preserves its hallmark sample efficiency while substantially improving generalization. Notably, ViGMO attains this robustness in a zero-shot manner, without requiring test-time adaptation.

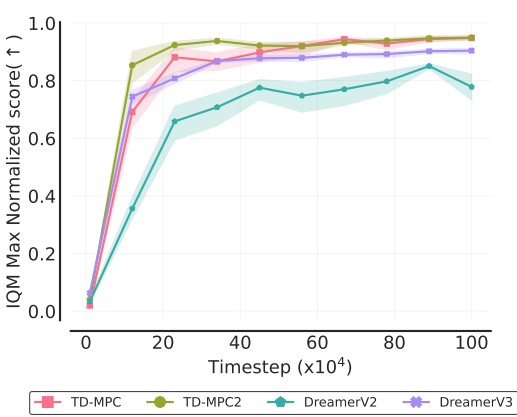

Figure 31: **Comparison of sample efficiency among MBRL baselines.** TD-MPC2 achieves the highest sample efficiency across six tasks in the DMC suite, outperforming other MBRL baselines.

## G   THE USE OF LARGE LANGUAGE MODELS (LLMS)

This paper introduces a novel MBRL framework that achieves strong zero-shot visual generalization while preserving high sample efficiency, improving zero-shot generalization performance by up to 13 % over the strongest baseline. Extensive experiments and carefully designed ablation studies on the DMC suite and Robosuite validate that the integration of core components (MA, LC, and ER) is critical to achieving the best efficiency–robustness trade-off under visual distractions. All reported results, including tables and Figures, are obtained from rigorous experiments and not generated by LLMs. LLMs were only used to refine the writing, ensuring clarity, conciseness, and grammatical correctness, thereby improving the overall readability and coherence of the paper.

