# OpenReview forum: "Zero-Shot Visual Generalization in Model-Based Reinforcement Learning via Latent Consistency"
_ICLR.cc/2026/Conference — Submitted to ICLR 2026_

### Official Review · Reviewer_MxSJ · 2025-10-27

**Soundness:** 3
**Presentation:** 3
**Contribution:** 1
**Rating:** 2
**Confidence:** 5

**Summary:**

This paper proposes ViGMO (Visual Generalization in Model-based RL), a model-based reinforcement learning (MBRL) framework that addresses the problem of visual generalization. To mitigate the severe performance degradation of policies under visual perturbations (e.g., changes in lighting, background, or texture) at test time, ViGMO is designed to learn latent dynamics consistency and is composed of three key components:

* MA (Mixed Weak-to-Strong Augmentation) – applies strong augmentations to only half of each mini-batch to balance sample efficiency and robustness.
* LC (Latent Consistency) – enforces augmentation-invariant transition consistency in latent space by using weak/strong view pairs generated by MA.
* ER (Encoder Regularization) – aligns weak-only and weak→strong latent representations of the same frame to stabilize encoder features.

The framework is implemented on top of TD-MPC2, and experimental results on the RL-ViGen benchmark show that ViGMO outperforms both MFRL methods (e.g., SVEA, DrQ-v2, SGQN) and MBRL methods (e.g., DreamerV3, Dr. G, TD-MPC2), achieving up to 13% improvement in performance.

**Strengths:**

The idea of enforcing latent-space consistency in model-based RL is not new; it was already introduced in Dr. G (Ha et al., 2023) through dual contrastive learning and inverse dynamics. ViGMO’s contribution lies mainly in simplifying this idea into a decoder-free, MSE-based formulation compatible with TD-MPC2.

**Weaknesses:**

### (1) Limited conceptual novelty:
The central idea of enforcing latent dynamics consistency is not conceptually new.
Prior works such as Dr. G (Ha et al., 2023) and SPD (Kim et al., 2022) have already introduced consistency regularization at the latent-dynamics level.
In particular, Dr. G, a model-based framework, combines dual contrastive learning and recurrent inverse-dynamics losses to achieve consistency across visual perturbations.
ViGMO’s ER loss closely resembles the approach in SODA (Hansen & Wang, 2021), with its main novelty being the application of this loss within TD-MPC2.
This constitutes a useful engineering simplification but not a fundamentally new algorithmic insight.

### (2) Reproducibility and evaluation-benchmark ambiguity.
ViGMO is evaluated on the RL-ViGen benchmark, whereas Dr. G was originally assessed on DMControl-GB.
These benchmarks differ in rendering pipeline, camera randomization, and observation normalization,
which likely explains the large performance drop of Dr. G reported in this paper.
Without cross-benchmark replication or normalization of rendering settings,
it remains unclear whether the reported improvements stem from algorithmic advances or from environmental discrepancies.

### Overall assessment.
ViGMO demonstrates solid empirical results but offers only incremental novelty over prior latent-consistency frameworks such as SODA, SPD, and Dr. G. The main weaknesses lie in the unclear theoretical distinction, benchmark inconsistency, and limited mechanistic analysis.

**Questions:**

1. Use of mixed weak & strong augmentations
Many prior works already employ a combination of weak and strong augmentations for visual robustness.
Beyond reusing this common strategy, what specific contribution or new insight does ViGMO introduce?

2. Latent-dynamics consistency
The idea of enforcing consistency in latent dynamics was emphasized in Dr. G (Ha et al., 2023) and SPD (Kim et al., 2022).
How does ViGMO’s Latent Consistency (LC) differ conceptually or functionally from these prior formulations?

3. Encoder Regularization (ER)
The proposed ER component appears very similar to the feature-alignment regularization used in SODA (2021).
Could the authors clarify what is novel or distinct in their formulation?

4. Performance discrepancy of Dr. G
In prior literature, Dr. G consistently demonstrated strong visual robustness on both DMC and Robosuite benchmarks (evaluated on DMControl-GB). In contrast, the reported results here show a severe degradation of Dr. G’s performance.
Could the authors investigate and explain the cause of this gap—e.g., differences in benchmark version, rendering pipeline, or implementation details?

5. Missing comparison with recent state-of-the-art work (SMD, Zhang et al., 2024)
The paper omits comparison with the recent study “Focus on What Matters: Separated Models for Visual-Based RL Generalization (SMD, Zhang et al., 2024)”, which also tackles visual generalization in RL by separating representation and dynamics learning.

---

> ### Author Response · Authors · 2025-11-21
>
> We thank the reviewer for the thoughtful feedback. Below, we address each point with clarifications and additional analyses added in the revised manuscript, where all changes are highlighted in blue.
>
> ### **Weakness 1. “Limited Conceptual Novelty”**
>
> We acknowledge the strong lineage of consistency-based visual generalization methods, including SODA, SPD, and Dr. G. Below, we clarify how ViGMO differs conceptually and methodologically from these prior studies, and why its contribution is not a re-implementation of existing consistency ideas, but a planner-centric restructuring of consistency for latent-space MBRL.
>
> **(1) Prior works regularize representations; ViGMO regularizes *planner-used latent transitions*.**
>
> SODA enforces aug→non-aug feature alignment through an EMA predictor and MSE; SPD performs two-way augmentation, view-gap discrimination, and inverse/forward dynamics alignment; and Dr. G employs dual contrastive learning (reality–reality and dream–reality InfoNCE) together with RSID to jointly shape representations and action-prediction within a decoder-free Dreamer-style world model.
>
> In contrast, **LC (Latent Consistency)** is a *non-contrastive, negative-free, temperature-free* constraint applied *directly to the forward transition* used for MPC imagination. LC enforces that the next latent predicted from a mixed weak-to-strong input must match the weak target. This provides **planner-facing stabilization** rather than generic representation alignment.
>
> **(2) MA (mixed weak-to-strong augmentation) supplies the structure that LC and ER require and preserves TD-MPC2-level efficiency.**
>
> MA (Mixed weak-to-strong) provides role-specific, asymmetric batching: every sample receives a weak view, while only a subset receives strong augmentation. This yields paired weak / weak-to-strong latents within the same minibatch, which (i) maintains TD-MPC2-level sample efficiency by avoiding two-view encodes, and (ii) structurally supplies the cross-view inputs that LC and ER require (Section 3.1). Appendix D.3 (Table 8) shows that mixed batching—not weak-only, strong-only, or weak-to-strong-only-subset—is essential for stability and robustness.
>
> **(3) ER (Encoder Regularization) serves a role specific to latent-space MBRL with a shared encoder.**
>
> Although inspired by SODA, ER is not a generic feature-alignment loss. In ViGMO, ER **prevents LC-induced drift in the shared encoder**, which the world model, Q-functions, and the planner simultaneously use. This stabilization is crucial for maintaining multi-step rollout quality in TD-MPC-style planning (Section 3.3; Appendix E).
>
> **(4) Empirical evidence shows that MA→LC→ER forms a tightly coupled mechanism, not modular reuse of prior ideas.**
>
> Removing LC collapses Robosuite performance to ~1% of full ViGMO (Table 2). Removing ER preserves only partial robustness but induces large variance. On DMC, LC, and ER, respectively, preserve 84% and 96% of full performance (Table 2). Appendix D.3 further shows that mixed augmentation pairing is necessary for stability (Table 8; Figure 14). Combined, MA→LC→ER yield substantial robustness gains without sacrificing sample efficiency, unlike strong augmentation or contrastive pipelines (Table 1; Figure 4).
>
> **(5) Overall novelty.**
>
> ViGMO does not propose “yet another form of representation consistency.” Its novelty lies in **restructuring consistency as a planner-centric constraint for latent-space MBRL**, enabling consistent multi-step imagination *without* sacrificing TD-MPC2’s sample efficiency—an outcome that prior contrastive or augmentation-heavy pipelines (e.g., SPD, Dr. G) do not achieve.

---

> > ### Author Response · Authors · 2025-11-21
> >
> > ### **Weakness 2 & Question 4. “Reproducibility and benchmark ambiguity between RL-ViGen and DMControl-GB”**
> >
> > We carefully analyzed whether differences between RL-ViGen and DMControl-GB could account for the performance drop observed in Dr. G's experiments. Our investigation shows that this is **not** the case.
> >
> > **(1) Rendering-pipeline differences are minimal.**
> >
> > To directly test the reviewer’s hypothesis, we rendered the same original frames using the RL-ViGen and DMC-GB background-overlay pipelines and computed pixel-level differences (Appendix D.11; Figures 28-29). The only systematic discrepancy was the mask color used for foreground extraction (green in DMC-GB vs. black in RL-ViGen), which leads to slightly different shadow rendering. Apart from this shadow effect, the rendered observations were effectively identical. Thus, the benchmark-level differences in the rendering pipeline, camera randomization, and observation normalization have a negligible impact and do not account for the magnitude of Dr. G’s performance drop.
> >
> > **(2) All algorithms—including Dr. G—were evaluated using their official implementations under the same RL-ViGen pipeline.**
> >
> > We emphasize that we did not reuse Dr. G’s DMControl-GB results. Instead, we re-trained Dr. G and all baselines from scratch under identical RL-ViGen rendering, normalization, and camera settings. Therefore, no method was advantaged or disadvantaged by differences in preprocessing or environment design.
> >
> > **(3) Dr. G’s reduced performance reflects algorithmic sensitivity—not benchmark artifacts.**
> >
> > Under matched conditions, Dr. G consistently struggled on RL-ViGen’s video-hard, lighting, and scene-clutter settings—even though ViGMO remained stable. This indicates that the gap is due to algorithmic factors (e.g., sensitivity of contrastive representation learning and recurrent inverse-dynamics losses to high-variance backgrounds), not environmental discrepancies.
> >
> > **(4) RL-ViGen should be viewed as an extension—not a deviation—of DMControl-GB.**
> >
> > RL-ViGen preserves the DMControl-GB rendering mechanics (as confirmed in Appendix D.11) and extends them with additional appearance perturbations (light motion, color synchrony, camera shifts). Thus, RL-ViGen remains **fully comparable** with prior DMControl-GB work while supplying a broader range of perturbations.
> >
> > **Summary.**
> >
> > We verified that RL-ViGen and DMC-GB differ only in foreground-mask color and shadow rendering; we re-trained all baselines under identical RL-ViGen settings; and ViGMO’s gains persist under these matched conditions. Thus, the reported improvements are due to algorithmic robustness, not discrepancies between benchmarks.

---

> > > ### Author Response · Authors · 2025-11-21
> > >
> > > ### **Question 1. “Use of mixed weak & strong augmentations — what is the new contribution?”**
> > >
> > > While many prior works use weak/strong augmentations, **ViGMO’s MA mechanism is not a generic two-view strategy**. Its contribution lies in a *role-specific, asymmetric batching design* that is **tightly integrated with LC and ER** to stabilize latent *dynamics* (not only representations) in latent-space MBRL. The key points are as follows:
> > >
> > > **(1) MA provides asymmetric batching that preserves efficiency (no 2× encoder cost).**
> > >
> > > Unlike standard two-view pipelines—where every sample is encoded twice—MA applies a weak augmentation to **all** samples, applies a strong augmentation to **only a subset**, and **reuses the weak view** for both LC and ER. Thus, **each image is encoded exactly once**, maintaining TD-MPC2-level computational efficiency (Section 3.1; Appendices D.3, D.6), while still creating the structured weak-only and weak-to-strong pairs required for consistency learning.
> > >
> > > **(2) MA produces the paired latents required by LC and ER.**
> > >
> > > From each rollout’s first step we form paired latents $z_t^{w}$, $z_t^{ws}$ and a mixed input $z_t^{m}=z_t^{w} \oplus z_t^{ws}$ (Figure 2). LC directly regularizes the transition by regressing $d_{\theta}(z_t^{m}, a_t)$ to the weak next-latent target $z_{t+1}^{w}$ (decoder-free, non-contrastive MSE), and ER aligns $z_t^{ws}$ to $z_t^{w}$ (l2-normalized MSE) to stabilize the shared encoder used by the world model, Q-functions, and MPC (Sections 3.2–3.3, Appendix E; Figure 2). This supervision is planner-centric (on the dynamics actually used for imagination), rather than a generic representation loss.
> > >
> > > **(3) Ablations show that MA’s structure—not just “using weak+strong”—is essential.**
> > >
> > > Replacing MA with weak‑only, strong‑only, or weak-to-strong-only for every sample degrades sharply: on DMC, strong‑only falls to 9% and weak-to-strong‑only to 31% of ViGMO; on Robosuite, weak‑only is ~1%, strong‑only 20%, and weak-to-strong‑only 46% of ViGMO (Appendix D.3; Table 8; Figure 14). These results also explain why “all‑strong” inputs cause instability, whereas MA’s mixture preserves both robustness and efficiency.
> > >
> > > **(4) Robustness stems from MA→LC→ER, not from a specific augmentation.**
> > >
> > > Appendix D.9 shows that ViGMO variants trained with **random-conv** or **SRM** (frequency masking) also maintain strong zero-shot robustness, despite differing substantially from overlay-based perturbations. LC and ER provide consistent gains across all families, confirming that **the structure of MA→LC→ER—not the augmentation choice—is the primary driver of robustness**.
> > >
> > > **Summary.**
> > >
> > > ViGMO does not simply “use weak and strong augmentations.” Its novelty lies in a batch-partitioned MA mechanism that provides the paired weak / weak-to-strong latents required for transition-level LC and encoder-stabilizing ER, all designed around the needs of latent-space planning. This yields substantial robustness without doubling encoder cost or harming sample efficiency, which we verify extensively in Appendix D.

---

> > > > ### Author Response · Authors · 2025-11-21
> > > >
> > > > ### **Question 2. “How does LC differ from prior latent-dynamics consistency approaches (Dr. G, SPD)?”**
> > > >
> > > > While prior work has explored consistency in latent spaces, **ViGMO’s Latent Consistency (LC) differs both conceptually and functionally** from Dr. G and SPD. LC is designed specifically for **latent-space MPC** and operates directly on the **planner-used transition model**, not on generic representations.
> > > >
> > > > **(1) LC is a *planner-centric, transition-level* constraint.**
> > > >
> > > > LC directly regularizes the forward latent transition used by MPC. At the first step of each rollout, LC takes the next-latent prediction from a mixed weak-to-strong latent input and regresses it to the weak next-latent target (Section 3.2; Figure 2). LC is (i) non-contrastive (no negatives, no temperature), (ii) decoder-free, and (iii) applied inside the same dynamics model used for multi-step imagination and planning. Thus, LC constrains the planner-facing world model rather than only shaping latent representations.
> > > >
> > > > **(2) Difference from Dr. G (Ha et al., 2023).**
> > > >
> > > > Dr. G enforces consistency through: (i) Dual Contrastive Learning (DCL): two InfoNCE losses (reality–reality and dream–reality) and (ii) RSID: an inverse-dynamics loss over imagined states. These objectives operate via contrastive similarity and require negatives and temperatures; they do not directly regress the transition used for planning. In contrast, LC acts directly on the transition model used for planning and uses a straightforward, stable MSE objective without contrastive machinery.
> > > >
> > > > **(3) Difference from SPD (Kim et al., 2022).**
> > > >
> > > > SPD combines: (i) two-way augmented views, (ii) a view-gap discriminator, and (iii) inverse/forward dynamics chaining, but its consistency losses are used to train a shared encoder for an MFRL policy, not the transition model used for MPC rollouts. SPD’s forward-dynamics alignment is an auxiliary representation-learning objective—not a planner-level constraint on the world model itself.
> > > >
> > > > **(4) Core distinction: ViGMO enforces *transition-level consistency* inside the world model.**
> > > >
> > > > Only ViGMO performs transition-level, non-contrastive regression inside the planner-used world model: $d_{\theta}(z_t^{m}, a_t) \rightarrow z_{t+1}^{w}$, providing a stability signal that directly reduces compounding errors under visual shifts.

---

> > > > > ### Author Response · Authors · 2025-11-27
> > > > >
> > > > > ### **Question 5. “Missing comparison with SMG (Zhang et al., 2024)”**
> > > > >
> > > > > We agree that SMG (Zhang et al., 2024) is an important recent contribution to visual generalization, and we have added a direct comparison in the revised manuscript (Appendix D.12). Below, we clarify the methodological differences and summarize the new results.
> > > > >
> > > > > **(1) SMG and ViGMO address fundamentally different problem settings.**
> > > > >
> > > > > **SMG is a *model-free* method.**
> > > > >
> > > > > Its core idea is to separate a representation encoder from an auxiliary latent-prediction module so that the encoder can learn perturbation-invariant features while the predictor captures simple latent updates. As described in SMG, the “dynamics module” is *not* a world model—it does not support latent imagination, multi-step rollouts, or planning, and instead provides auxiliary supervision within an actor–critic model-free pipeline. Thus, SMG is designed to improve representation robustness without explicitly modeling latent dynamics.
> > > > >
> > > > > **ViGMO is a *model-based* method.**
> > > > >
> > > > > ViGMO addresses a failure mode unique to latent-space MBRL: instability of the learned transition model under strong visual shifts. Our latent-consistency mechanism (MA + LC + ER) operates directly inside the world model used for multi-step latent rollouts and MPC planning. This planner-facing stabilization affects not only the encoder but also the entire latent-dynamics pipeline used for control.
> > > > >
> > > > > Thus, while both methods aim to improve visual generalization, SMG enhances representation invariance for model-free RL, whereas ViGMO enforces transition-level consistency for model-based planning.
> > > > >
> > > > > **(2) Newly added comparison with SMG (Appendix D.12; Figure 30).**
> > > > >
> > > > > As requested by the reviewer, we have included an experimental comparison with SMG in the revised manuscript. We evaluate SMG using the authors’ anonymized reference implementation (https://anonymous.4open.science/r/SMG/) under identical RL-ViGen training protocols and zero-shot evaluation settings used for all baselines.
> > > > >
> > > > > Following prior SMG evaluations, we report performance aggregated across five DMC tasks:
> > > > > *cartpole-swingup*, *finger-spin*, *walker-stand*, *walker-walk*, and *cheetah-run*.
> > > > >
> > > > > The results (Appendix D.12; Figure 30) show that ViGMO trains faster and achieves higher aggregate zero-shot returns than SMG under the same conditions. These differences are consistent with the methodological distinction above: ViGMO benefits from stabilized multi-step latent rollouts through MA + LC + ER under diverse appearance shifts, whereas SMG remains competitive within its intended model-free setting but does not leverage planning or a world model.
> > > > >
> > > > > In summary, SMG and ViGMO address different algorithmic challenges—model-free representation robustness vs. latent-dynamics consistency. Our newly added experiments provide a fair, matched comparison under identical training and evaluation protocols, where ViGMO demonstrates stronger aggregate zero-shot generalization across the five DMC tasks.

---

> ### Author Response · Authors · 2025-11-21
>
> ### **Question 3. “Encoder Regularization (ER) resemblance to SODA—what is novel or distinct?”**
>
> While ViGMO’s ER is inspired by the general intuition of aligning augmented and non-augmented representations, its **motivation, placement, and functional role differ fundamentally** from the feature-alignment regularizer in SODA.
>
> **(1) ER solves a problem that does *not* arise in SODA: LC-induced drift in a shared-encoder MBRL framework.**
>
> SODA aligns aug→clean features to stabilize model-free Q-learning, where the encoder feeds only a policy or Q-function. In ViGMO, the encoder simultaneously influences: (i) the latent transition model used for multi-step imagination, (ii) reward/Q estimation, and (iii) MPC planning. LC introduces strong cross-augmentation supervision on the transition model. Without ER, this pressure causes the encoder to drift, which directly destabilizes multi-step rollouts—a failure mode that does not arise in SODA’s architecture. ER is specifically introduced to counteract this LC-induced drift and preserve a stable latent geometry for planning.
>
> **(2) ER is *structurally coupled* to LC and the planner-used dynamics model.**
>
> SODA’s alignment is a standalone representation objective. ER, in contrast: (i) aligns weak vs. weak-to-strong latents of the same frame, (ii) ensures that LC’s transition-level regression has a stable target space, and (iii) stabilizes the entire rollout pipeline used by MPC. Thus, ER is structurally part of the MA → LC → ER mechanism; it is not interchangeable with SODA-style feature alignment.
>
> **(3) Empirically, ER has a unique and essential role in latent-space MBRL.**
>
> Removing ER severely degrades robustness, especially on visually complex Robosuite tasks (≈51% of full ViGMO performance; Table 2), even though LC remains intact. This demonstrates that ER is not a generic alignment loss but a planner-critical stabilizer required for consistent latent dynamics under visual shifts.
>
> **Summary.**
>
> While ER draws inspiration from SODA, its purpose is distinct: SODA stabilizes representations for model-free RL, whereas ViGMO’s ER stabilizes the shared encoder inside a model-based planner so that LC can reliably regularize the forward dynamics used for multi-step imagination. This functional coupling to LC and the world model differs from SODA.

---

### Official Review · Reviewer_kVE1 · 2025-11-01

**Soundness:** 3
**Presentation:** 3
**Contribution:** 3
**Rating:** 6
**Confidence:** 2

**Summary:**

This paper addresses a critical limitation in current Model-Based Reinforcement Learning (MBRL): catastrophic performance degradation when facing unseen visual distractions at test time (zero-shot generalization). The authors propose ViGMO (Visual Generalization in Model-based RL), a framework built atop TD-MPC2. ViGMO introduces three complementary components to robustify latent-space planning without sacrificing sample efficiency.

**Strengths:**

1. Preservation of Sample Efficiency: A major contribution is achieving robustness without the typical sample efficiency tax associated with domain randomization or heavy augmentation in MBRL. By using standard TD-MPC2 as a backbone and employing the Mixed Augmentation (MA) strategy, ViGMO matches the learning curves of non-robust MBRL baselines on clean tasks. This addresses a key pain point in applying robust RL to real-world scenarios where data is expensive.
2. Effective Decomposition of the Problem: The separation of concerns between Encoder Regularization (ER) and Latent-Consistency (LC) is well-motivated for MBRL. The authors correctly identify that in MBRL, it is insufficient for just the observations to be encoded invariantly; the transition dynamics must also be robust to perturbations to prevent compounding errors during multi-step planning. The ablation studies (Table 2) clearly vindicate this design, showing that removing LC leads to catastrophic failure in severe shift scenarios (Robosuite), while ER is necessary for stability.
3. Insightful Analysis of Latent Space: Figure 6 and Appendix D.5 provide compelling qualitative evidence that ViGMO successfully maintains a compact, aligned latent manifold under perturbation, whereas baselines like Dr. G and TD-MPC2 show significant latent state divergence when exposed to distractions.

**Weaknesses:**

1. Incremental Technical Novelty: While the combination and application to MBRL are effective, the individual components are largely adaptations of existing techniques. The MA strategy resembles techniques used in semi-supervised learning or robust MFRL (like SVEA), and ER is explicitly identified as an adaptation of SODA. The primary novelty lies in the LC loss specifically targeting the dynamics model $(d_\theta)$ using mixed augmentations.
2. Dependency on Specific Augmentations: The framework relies heavily on the choice of "random-overlay" as the strong augmentation. While the ablation in Figure 15 shows it outperforms "random-conv" for these specific benchmarks, this might be an artifact of the benchmarks themselves (which often involve background replacements). The paper might overclaim "general" robustness when it is specifically highly robust to overlay-style distractions.
3. Lack of Computational Overhead Analysis: The paper claims MA avoids computational overhead by splitting the batch13. While it avoids doubling the batch size, computing strong augmentations (especially overlays requiring external datasets) likely incurs a wall-clock time penalty compared to simple weak augmentations. A wall-clock time comparison is missing to fully substantiate the "efficiency" claim beyond just sample counts.

**Questions:**

1. Does ViGMO generalize to spectrally different shifts (e.g., sensor noise, realistic lighting physics) not well-modeled by the random-overlay training augmentation?
2. Why are dynamic augmentations superior over short horizons ($H=3$) when real-world distractions often exhibit temporal consistency?
3. Does the effectiveness of Latent-Consistency (LC) learning degrade at longer planning horizons ($H>3$) where MBRL typically suffers from compounding errors?
4. What is the actual wall-clock training time penalty of ViGMO compared to TD-MPC2, accounting for complex augmentations and auxiliary losses?

---

> ### Author Response · Authors · 2025-11-21
>
> We thank the reviewer for the thoughtful and constructive feedback. Below, we address each concern and summarize the clarifications, experiments, and analyses added in the revised manuscript, where all modifications are highlighted in blue.
>
> ### **Weakness 1. “Incremental Technical Novelty”**
>
> While MA and ER draw inspiration from prior augmentation frameworks, their **roles, structures, and interactions within latent-space MBRL** are fundamentally different from those of existing methods.
>
> **(1) MA: Role-specific, asymmetric batching**
>
> MA is not simply a combination of weak and strong augmentations. Its asymmetric design, where only a subset of samples receives strong augmentation, serves a specific structural role in latent-space MBRL:
>
> - Every sample is processed through a weak augmentation, ensuring that the encoder, dynamics model, and planner remain grounded in stable weak latents—preserving the sample-efficiency characteristic of latent MBRL.
> - Only a subset of samples is further transformed using a strong augmentation, producing paired weak-only and weak-to-strong latents within each minibatch.
> - These paired latents yield the precise supervision structure needed by LC and ER: (i) LC applies a transition-level constraint by taking mixed weak-to-strong inputs and regressing to a weak next-latent target, and (ii) ER stabilizes the shared encoder by aligning weak-only vs. weak-to-strong latents of the same frame.
> - Because strong augmentation is applied only to a subset and reuses weak latents, MA avoids the computational overhead of full two-view encoding and prevents the instability observed when the entire batch is strongly augmented, thereby maintaining both computational efficiency and optimization stability.
>
> Ablation studies confirm the necessity of this structured asymmetry: weak-only, strong-only, and weak-to-strong-only configurations all degrade performance markedly (Appendix D.3; Table 8), demonstrating that MA’s subset-based pairing is essential for achieving both robustness and efficiency in ViGMO.
>
> **(2) ER: Stabilizing the shared encoder under LC**
>
> Although ER is inspired by aug→clean alignment ideas such as SODA, its function in ViGMO is unique to latent-space MBRL:
>
> - TD-MPC2 employs a single shared encoder whose latents feed the world-model transition function, value networks, and the MPC planner.
> - LC introduces strong cross-augmentation supervision by training the dynamics model on mixed weak-to-strong latents while anchoring predictions to weak next-latent targets.
> - Without ER, this cross-augmentation pressure can distort the shared encoder, causing the latent geometry to drift and propagating errors through multi-step imagined rollouts.
>
> Thus, ER is not a generic MFRL alignment loss but a **structural stabilizer enabling LC** to operate without collapsing latent geometry.
>
> The novelty of ViGMO lies not in isolated components but in a **latent-consistency framework** tailored for latent-space MBRL. Ablations and analyses (Appendices D.3, D.4, D.9) show that **MA + LC + ER collectively reshape the inductive bias of the latent transition model**, producing robustness unattained by prior approaches.

---

> > ### Author Response · Authors · 2025-11-21
> >
> > ### **Weakness 2. “Dependency on random-overlay augmentation”**
> >
> > Our primary experiments use *random-overlay* as the strong augmentation because it is known to be the most robust and visually generalizable for prior visual RL baselines (e.g., SVEA, SODA, SGQN). However, to verify whether ViGMO’s robustness depends on this specific augmentation family—or whether its latent-consistency mechanism generalizes more broadly—we conducted additional analyses in Appendix D.9.
> >
> > **Comparison with alternative strong augmentations (Appendix D.9).**
> >
> > We constructed two additional ViGMO variants that differ *only* in their strong augmentation, while keeping the encoder, world model, planner, and MA+LC+ER structure identical:
> >
> > - **ViGMO_CONV** — uses *random-conv*, which perturbs local texture and color statistics without altering global scene layout.
> > - **ViGMO_SRM** — uses *Spectrum Random Masking (SRM)*, which masks contiguous Fourier-spectrum bands and removes mid/high-frequency components.
> >
> > These augmentations belong to **distinct augmentation families**—spatial/texture (random-conv) and frequency-based (SRM)—and differ substantially from overlay-style contextual perturbations.
> >
> > All three ViGMO variants exhibit stable learning curves and strong zero-shot robustness despite using different augmentation families (Appendix D.9; Figure 26). While the default ViGMO (*random-overlay*) achieves the highest overall performance, both ViGMO_CONV and ViGMO_SRM achieve competitive zero-shot performance.
> >
> > These results demonstrate that ViGMO’s robustness is **augmentation-agnostic**: the MA + LC + ER framework stabilizes latent transitions across spatial, contextual, and spectral perturbation families. The specific choice of strong augmentation modulates which distraction types are handled best (e.g., contextual vs. color-based), but the effectiveness of ViGMO does not rely on any single augmentation family.
> >
> > ### **Weakness 3 & Question 4. “Lack of computational overhead analysis”**
> >
> > We appreciate the reviewer’s concern regarding the practical cost of ViGMO’s dual augmentations and auxiliary losses. Appendix D.6 provides detailed wall-clock analyses, and the key findings are as follows.
> >
> > **First**, ViGMO’s design explicitly avoids the **2× encoder cost** that would occur if both weak and strong views were encoded for every sample. As described in Section 3.1, MA applies the weak augmentation to the full batch and applies the strong augmentation only to a **subset**, reusing the weak view. This ensures that **each image is encoded exactly once**, preventing the effective encoder workload from doubling.
> >
> > **Second**, the two additional losses introduced by ViGMO—LC and ER—are implemented as lightweight latent-space MSE losses (Sections 3.2–3.3). These operations are negligible compared to TD-MPC2’s dynamics, Q-function, and planner updates, and therefore contribute virtually no measurable overhead.
> >
> > In the meantime, we agree with the reviewer that strong augmentations, such as *random-overlay,* introduce some wall-clock overhead due to background-video compositing and external image loading. To quantify this, we added explicit wall-clock training time comparisons in the revised manuscript (Appendix D.6), using identical hardware (single NVIDIA A5000), hyperparameters, dataloaders, and environment settings. Across 10 tasks in the DMC suite and Robosuite, we observe that: (i) ViGMO introduces a small wall-clock overhead relative to TD-MPC2, which continues to shrink as training proceeds (Appendix D.6; Figures 19-20, Table 9).
> >
> > In summary, Appendix D.6 demonstrates that ViGMO preserves TD-MPC2's **practical training efficiency** while providing substantially improved zero-shot robustness. This efficiency–robustness balance is one of the central advantages of ViGMO.

---

> ### Author Response · Authors · 2025-11-21
>
> ### **Question 1. “Generalization to spectrally different shifts (e.g., sensor noise, realistic lighting physics)”**
>
> We agree that sensor-level corruptions represent a qualitatively different class of shifts than the appearance-based distractions modeled by random-overlay. To evaluate this, we conducted additional experiments in **Appendix D.8** using two canonical *spectral* corruptions—**normal-blur** and **gaussian-noise**—that remove high-frequency detail or inject pixel-wise stochastic noise.
>
> **(1) Sensor noise introduces a fundamentally different distribution shift.**
>
> Blur and white-noise corruption alter the image's frequency structure by suppressing edges or adding high-frequency perturbations. These distortions do not resemble contextual shifts (e.g., background, lighting, color) used during training and therefore create a substantially different OOD regime.
>
> **(2) All generalization-oriented methods degrade under sensor noise.**
>
> As shown in Appendix D.8 (Figures 23–24), methods explicitly designed for visual generalization—SVEA, SGQN, SRM, Dr. G, and ViGMO—experience large performance drops. This is expected because none of these methods incorporates spectral corruptions during training. Interestingly, task-specific methods (DrQ-v2, TD-MPC2, and DreamerV3) remain more robust to blur and noise. This robustness likely stems not from true generalization, but from convolutional inductive biases and task-specific representations that remain functional when corruption does not alter scene geometry or dynamics. Although all generalization-focused methods degrade, ViGMO consistently achieves the highest performance within this group, indicating that its latent-consistency mechanism provides a degree of robustness even to perturbations outside its intended appearance-shift domain.
>
> Overall, sensor noise represents a fundamentally different category of distribution shift—destroying pixel-level fidelity rather than altering contextual appearance. While ViGMO improves robustness to contextual and appearance perturbations, sensor-level corruptions remain challenging for all generalization methods. Developing latent-space models that unify robustness across appearance, geometry, and sensing degradations constitutes a promising direction for future work.
>
> ### **Question 2. “Why use dynamic augmentations under short horizons (H=3)?”**
>
> Our use of dynamic augmentations is motivated by how LC/ER supervise the world model over short rollout segments, not by an assumption that real‑world distractors are temporally random.
>
> **How LC/ER are supervised over a short horizon.**
>
> - For LC, at the first step of each length‑*H* rollout, we form mixed latents from paired weak‑only and weak-to-strong views and regress the model’s next‑latent prediction to a weak target; subsequent steps roll out from that mixed input within the same segment (Sections 3.1–3.2; Figure 2). ER aligns weak‑only vs. weak-to-strong latents at every step to keep the shared encoder stable (Section 3.3).
> - In Section 4.3, we explicitly define dynamic vs. consistent augmentation across the horizon *t*:*t+H* and evaluate a ViGMO_CONST_AUG variant that holds the augmentation fixed within each segment (Appendix D.4).
>
> **Why dynamic augmentations help when H=3.**
>
> With a short horizon, keeping a single nuisance fixed across the whole segment provides LC/ER with only one pattern per rollout. Dynamically resampling weak/strong views across the steps *t*, *t+1*, *t+2* exposes the transition model and encoder to multiple nuisance realizations within each segment, which (i) strengthens LC’s transition‑level regression signal and (ii) reduces overfitting of ER to a single artifact. This design yields more informative supervision for the quantities the planner actually uses (next‑state predictions), even when test‑time shifts are temporally consistent. (Sections 3.2–3.3; Section 4.3 “Dynamic vs. consistent augmentations”)
>
> **Empirical evidence.**
>
> Appendix D.4 shows that the full ViGMO (dynamic resampling) consistently outperforms ViGMO_CONST_AUG across distraction types and in aggregate metrics; see the per‑type curves in Figure 15 and the aggregate/efficiency comparisons in Figure 16 (more stable monotonic degradation and better IQM/mean/optimality‑gap), confirming that dynamic sampling provides stronger supervision for robust latent dynamics.

---

> > ### Author Response · Authors · 2025-11-21
> >
> > ### **Question 3. “Does LC degrade at longer planning horizons?”**
> >
> > Our default experiments adopt a short MPC planning horizon of H = 3, following standard practice in latent-space MPC where short horizons offer a favorable trade-off between predictive stability, computational cost, and sample efficiency. To examine whether ViGMO’s latent-consistency mechanism remains effective when planning over deeper imagined rollouts, we conduct additional comparisons under extended planning horizons (Appendix D.10 of the revision).
> >
> > Across these extended horizons, we observe the following: ViGMO maintains stable optimization and reaches similar asymptotic returns. As expected, increasing the planning horizon slows convergence because compounding prediction errors accumulate more severely over longer imagined trajectories—a well-documented limitation of MBRL. Nevertheless, ViGMO’s latent-consistency regularization continues to stabilize the forward dynamics, preventing error accumulation from degrading policy learning even when the planning depth is significantly increased. These findings indicate that ViGMO’s latent-consistency mechanism generalizes beyond the short-horizon setting and remains effective under longer imagined rollouts.

---

### Official Review · Reviewer_2G6K · 2025-11-02

**Soundness:** 2
**Presentation:** 2
**Contribution:** 2
**Rating:** 6
**Confidence:** 4

**Summary:**

This paper introduces ViGMO (Visual Generalization in Model-based Reinforcement Learning), a novel framework designed to achieve zero-shot visual generalization in model-based reinforcement learning (MBRL). While conventional MBRL methods are highly sample-efficient, they often fail when facing unseen visual distractions such as background, lighting, or viewpoint changes. ViGMO addresses this limitation by incorporating three key components into the latent-space world model: a mixed weak-to-strong augmentation (MA) strategy that balances efficient learning with robustness, a latent-consistency learning (LC) module that enforces consistent latent transitions across augmented views to stabilize predictions under distribution shifts, and an encoder regularization (ER) term that preserves task-relevant features and prevents representational collapse. These components can be easily integrated into existing MBRL backbones such as TD-MPC2 without changing the planner or optimization procedure.

Experiments on the DeepMind Control Suite and Robosuite show that ViGMO significantly improves robustness under unseen distractions, achieving up to 13% higher zero-shot generalization than the strongest baseline while maintaining similar sample efficiency. Ablation studies further verify that both LC and ER are essential for stable latent dynamics and representation robustness, and visual analyses demonstrate that ViGMO maintains well-aligned latent manifolds under perturbations, unlike competing methods. Overall, ViGMO provides a simple yet effective solution for enabling model-based agents to generalize to unseen visual conditions without fine-tuning, achieving a strong balance between robustness and efficiency in visual RL.

**Strengths:**

On both the DeepMind Control Suite and Robosuite, ViGMO achieves state-of-the-art zero-shot generalization while maintaining similar sample efficiency to TD-MPC2. As reported in Table 1, ViGMO improves mean returns by up to 13% over the strongest baseline and remains stable across three difficulty levels (easy, hard, and extreme). These results demonstrate that ViGMO effectively balances robustness and efficiency.

The paper includes detailed ablations on the contribution of each component (Table 2) and design choice (Figure 5), clearly showing that both LC and ER are crucial for performance. The analysis in Section 4.4 and Figure 6 further provides visual evidence that ViGMO preserves latent-space consistency under distribution shifts.

**Weaknesses:**

While the paper reports up to 13% improvement in zero-shot generalization (Table 1), the absolute gains over strong baselines such as TD-MPC2 and SVEA are modest on some simpler DMC tasks. The per-task results (Appendix D.1) suggest that ViGMO’s advantage is most evident under extreme visual distractions, but less consistent in easier settings.

ViGMO introduces additional training costs through dual augmentations (weak and strong) and two auxiliary losses (LC and ER). However, the paper does not provide quantitative comparisons of training time or GPU usage relative to the TD-MPC2 baseline. Without such data, it is difficult to assess the efficiency–robustness trade-off in practical use.

The experiments are mainly conducted on a few tasks from the DeepMind Control Suite and Robosuite (Figure 3). Although these benchmarks include some distractions such as background color, object color, and video overlays, they do not fully represent more complex or real-world domain shifts. The method’s ability to generalize to unseen dynamics changes, camera viewpoints, or object appearances remains unclear.

**Questions:**

The experiments mainly consider visual distractions (e.g., color and texture changes). Have you tested ViGMO under non-visual domain shifts, such as changes in dynamics or camera viewpoint? How well would the method generalize in those cases?

Why did you choose random-overlay and random-shift specifically as the strong and weak augmentations? Did you compare with other candidates like color jitter, random convolution, or domain randomization (e.g., from Figure 5)? How sensitive is ViGMO to these augmentation choices?

The experiments are performed entirely in simulation. Do you anticipate any challenges when applying ViGMO to real-robot settings where visual distractions and dynamics changes occur simultaneously?

---

> ### Author Response · Authors · 2025-11-21
>
> We thank the reviewer for the careful reading and constructive feedback. Below, we respond to each point and summarize the analyses and experiments added in the revised manuscript, where all modifications are highlighted in blue.
>
> ### **Weakness 1. “Modest gains on simpler DMC tasks”**
>
> The variation in per-task gains is expected given (i) the strength of our baselines and (ii) the diversity of the RL-ViGen benchmark.
>
> **First**, our experimental setup includes a stronger, more comprehensive set of baselines than much prior work on visual generalization. Specifically, we evaluate against (i) specialized visual-generalization MFRL baselines (SVEA, SGQN, SRM), state-of-the-art visuomotor MBRL agents (TD-MPC2, DreamerV3), and a recent state-of-the-art visual-generalization MBRL method (Dr. G). Because we compare against such strong baselines, per-task margins naturally become tighter, even though ViGMO achieves the strongest average zero-shot performance.
>
> **Second**, we evaluate ViGMO on the full RL-ViGen testbed, which includes diverse and challenging perturbations such as background-video, camera/viewpoint variation, lighting color/position shifts, and texture/color changes. As reported in the RL-ViGen benchmark, no single method achieves uniformly best performance across all tasks and all distractions—performance varies significantly by perturbation type and difficulty. Some algorithms (e.g., SVEA, Dr. G) achieve the best performance on isolated tasks, but ViGMO consistently provides **the best aggregate robustness**, especially on the severe visual distractions.
>
> **Finally**, an important contribution of ViGMO is that it improves zero-shot visual generalization without sacrificing sample efficiency, a trade-off that has been challenging for prior work. Many visual-generalization methods improve robustness at the cost of slower or less stable training, whereas ViGMO maintains TD-MPC2-level sample efficiency (Figure 4; Appendix D.2) while providing significant robustness gains—especially on the more severe shift categories where existing methods degrade sharply.
>
> In summary, while per-task improvements may be modest on simpler tasks with strong baselines, ViGMO achieves the most reliable robustness overall across the full RL-ViGen benchmark, particularly under severe distractions, while maintaining TD-MPC2-level efficiency. This balance of robustness and efficiency is exactly what prior approaches have struggled to achieve.
>
> ### **Weakness 2. “Lack of quantitative overhead analysis”**
>
> We appreciate the reviewer’s concern regarding the practical cost of ViGMO’s dual augmentations and auxiliary losses. Appendix D.6 provides detailed wall-clock analyses, and the key findings are as follows.
>
> **First**, ViGMO’s design explicitly avoids the **2× encoder cost** that would occur if both weak and strong views were encoded for every sample. As described in Section 3.1, MA applies the weak augmentation to the full batch and applies the strong augmentation only to a **subset**, reusing the weak view. This ensures that **each image is encoded exactly once**, preventing the effective encoder workload from doubling.
>
> **Second**, the two additional losses introduced by ViGMO—LC and ER—are implemented as lightweight latent-space MSE losses (Sections 3.2–3.3). These operations are negligible compared to TD-MPC2’s dynamics, Q-function, and planner updates, and therefore contribute virtually no measurable overhead.
>
> In the meantime, we agree with the reviewer that strong augmentations, such as *random-overlay,* introduce some wall-clock overhead due to background-video compositing and external image loading. To quantify this, we added explicit wall-clock training time comparisons in the revised manuscript (Appendix D.6), using identical hardware (single NVIDIA A5000), hyperparameters, dataloaders, and environment settings. Across 10 tasks in the DMC suite and Robosuite, we observe that: (i) ViGMO introduces a small wall-clock overhead relative to TD-MPC2, which continues to shrink as training proceeds (Appendix D.6; Figures 19-20, Table 9).
>
> In summary, Appendix D.6 demonstrates that ViGMO preserves TD-MPC2's **practical training efficiency** while providing substantially improved zero-shot robustness. This efficiency–robustness balance is one of the central advantages of ViGMO.

---

> > ### Author Response · Authors · 2025-11-21
> >
> > ### **Weakness 3 & Question 1. “Unclearness to generalize unseen dynamics changes, camera viewpoints, or object appearances”**
> >
> > Our work focuses specifically on **zero-shot visual generalization** in latent-space MBRL, and our experiments therefore target a broad and diverse set of *visual* distribution shifts. As detailed in Section 4.1 and Appendices C.2–C.4, RL-ViGen includes background-video overlays, lighting color/position/motion, camera-position changes, and object/background color variations—substantially richer than the traditional DMControl-GB and color-jitter settings commonly used in prior work.
> >
> > That said, we fully agree that **camera-viewpoint rotations**, **dynamics changes**, and **object-appearance variations** introduce qualitatively different domain shifts that go beyond appearance changes. To address the reviewer’s concern, we added a new evaluation on **unseen camera-viewpoint rotations** (Appendix D.7; Figs. 21–22), which were not part of the original RL-ViGen benchmark:
> >
> > - These viewpoint perturbations introduce substantial geometric distortions—altering perspective, foreshortening limbs, reordering depth relationships, and in some cases causing the agent to become partially or entirely outside the field of view (e.g., *cartpole-swingup*).
> > - All methods, including ViGMO, degrade significantly under these viewpoint rotations, and none of the evaluated methods maintains reliable zero-shot generalization performance. This degradation occurs because rotational viewpoint shifts induce geometric transformations that fundamentally differ from appearance shifts (e.g., lighting, background, color).
> >
> > Developing geometry-aware latent representations, multiview–consistent world models, or scene-structured encoders remains an important direction for future work.
> >
> > Finally, as stated in our Conclusion’s “Limitations and Future Work,” extending ViGMO to handle **non-visual shifts,** such as dynamics changes, or task-level distribution shifts—and validating the approach on **real robotic systems**—is an important direction for future research.
> >
> > ### **Question 2. “Choice of Augmentations and Sensitivity”**
> >
> > Our choices of *random-shift* (weak) and *random-overlay* (strong) follow the requirements of latent-space MBRL, but our robustness does **not** rely on overlay-specific properties. Appendix D.9 provides a direct sensitivity analysis.
> >
> > **Weak augmentation — random-shift.**
> >
> > We adopt *random-shift* as the weak augmentation because it is known to be the most stable and sample-efficient for visual RL (e.g., DrQ-v2, TD-MPC2). As shown in prior work and confirmed in our experiments (Section 3.1), *random-shift* preserves geometric structure while providing mild invariance—properties well-suited for TD-MPC2’s latent dynamics and planning pipeline.
> >
> > **Strong augmentation — random-overlay.**
> >
> > We adopt *random-overlay* as the default strong augmentation because it is known to be the most robust and visually generalizable for visual RL (e.g., SVEA, SODA, SGQN) (Section 3.1; Appendix C). Overlay introduces spatial/contextual disturbances that strongly challenge latent dynamics models, making it an effective source of supervision for LC and ER.
> >
> > **Comparison with alternative strong augmentations (Appendix D.9).**
> >
> > We constructed two additional ViGMO variants that differ *only* in their strong augmentation, while keeping the encoder, world model, planner, and MA+LC+ER structure identical:
> >
> > - **ViGMO_CONV** — uses *random-conv*, which perturbs local texture and color statistics without altering global scene layout.
> > - **ViGMO_SRM** — uses *Spectrum Random Masking (SRM)*, which masks contiguous Fourier-spectrum bands and removes mid/high-frequency components.
> >
> > All three ViGMO variants exhibit stable learning curves and strong zero-shot robustness despite using different augmentation families (Appendix D.9; Figure 26). While the default ViGMO (*random-overlay*) achieves the highest overall performance, both ViGMO_CONV and ViGMO_SRM achieve competitive zero-shot performance.
> >
> > These results demonstrate that ViGMO’s robustness is **augmentation-agnostic**: the MA + LC + ER framework stabilizes latent transitions across spatial, contextual, and spectral perturbation families. The specific choice of strong augmentation modulates which distraction types are handled best (e.g., contextual vs. color-based), but the effectiveness of ViGMO does not rely on any single augmentation family.

---

> ### Author Response · Authors · 2025-11-21
>
> ### **Question 3. “Real-Robot Applicability and Simultaneous Visual-Dynamics Shifts”**
>
> Our work specifically targets zero-shot visual generalization in latent-space MBRL, and our experiments are therefore limited to controlled visual distribution shifts in simulation. As we note in the paper, ViGMO is not designed to address dynamics changes, and we do not claim robustness to simultaneous visual–dynamics shifts.
>
> We anticipate that applying ViGMO to real-robot settings could introduce additional challenges, as real environments often exhibit both visual distractions (e.g., lighting changes, viewpoint shifts) and dynamic variability (e.g., friction changes, actuator imperfections). While ViGMO already demonstrates robustness to several real-world–relevant visual factors—such as camera position changes and lighting variation (Appendix C.3)—addressing dynamics shifts would require extending the method beyond its current formulation.
>
> As stated in our Conclusion’s “Limitations and Future Work,” extending ViGMO to broader dynamics or task-level distribution changes, and validating the method on real robotic systems, remain important directions for future research.

---

### Author Response · Authors · 2025-12-01
**Summary of Novelty and Contributions**

We provide a concise summary of ViGMO’s novelty and contributions to support the Area Chair’s final assessment.

ViGMO introduces a **planner-centric consistency framework for latent-space MBRL** that achieves **state-of-the-art zero-shot visual generalization** while preserving the hallmark **sample efficiency** of TD-MPC2.

Its core innovation lies not in isolated components, but in the structural coupling of MA, LC, and ER, which collectively reshape the inductive biases of the latent transition model under visual shifts.

**1. Efficiency-Preserving Asymmetric Augmentation (MA)**

Unlike generic augmentation pairs, MA is an asymmetric batching scheme specifically designed for latent MBRL efficiency. By encoding each image exactly once and applying strong augmentations only to a subset used for consistency training, MA provides the required weak-only and weak-to-strong latent pairs for LC/ER without doubling the encoder workload, thereby preserving TD-MPC2–level computational efficiency. Ablations confirm that such variants (e.g., weak-only, strong-only, or weak-to-strong-only) compromise stability or robustness.

**2. Planner-Centric Latent Consistency (LC)**

Unlike prior methods (e.g., Dr.G, SPD) that rely on auxiliary contrastive or inverse-dynamics losses on latent representations, LC provides a transition-level, non-contrastive constraint applied directly inside the world model. By regressing predictions from mixed weak-to-strong inputs toward weak next-latent targets, LC directly stabilizes the planner-facing transition dynamics, thereby reducing compounding errors during multi-step imagination—an aspect that prior representation-focused approaches do not explicitly address.

**3. Structural Stability via Encoder Regularization (ER)**

ER is not a generic alignment objective like SODA. In the context of latent MBRL with a single shared encoder, ER serves as a critical structural stabilizer: it counteracts the latent-geometry drift induced by LC’s strong cross-augmentation pressure, ensuring that the shared encoder remains stable for both the planner and the value functions. Without ER, the shared encoder becomes unstable on visually complex tasks.

**4. Key Empirical Contributions**

Across RL-ViGen (DMC + Robosuite) with diverse and challenging unseen distractions:

- **State-of-the-art aggregate zero-shot visual generalization:** ViGMO achieves the best aggregate zero-shot performance across all tasks, outperforming strong model-free (SVEA, SGQN, SRM, DrQ-v2, SMG) and model-based (Dr.G, TD-MPC2, DreamerV3) baselines.
- **No efficiency compromise:** ViGMO maintains TD-MPC2–level sample efficiency with only modest wall-clock overhead (Appendix D.6).
- **Augmentation-agnostic robustness:** ViGMO generalize robustly across diverse augmentation families (overlay, convolution, spectral masking), demonstrating that robustness stems from the joint MA→LC→ER mechanism rather than any specific augmentation choice (Appendix D.9).
- **Stable under longer planning horizons:** ViGMO remains stable as the MPC horizon increases, which confirms LC’s stabilizing effect on multi-step predictions (Appendix D.10).

**Summary**

ViGMO fills a critical gap left by prior methods (Dr.G, SPD, SODA) by reframing visual generalization as a **transition-level, planner-centered consistency problem**. This integrated design yields a robust yet efficient MBRL agent that maintains performance under severe visual distractions, in settings where prior approaches struggle.

---

> ### Author Response · Authors · 2025-12-01
> **Summary of Authors’ Rebuttal and Additional Experiments**
>
> During the rebuttal and discussion period, we added new experiments and analyses that directly address all weaknesses and questions raised by the reviewers. The newly added materials are provided in **Appendices D.6–D.12**. Below, we summarize the key updates for the Area Chair.
>
> ### **1. Efficiency–Robustness Trade-off and Computational Overhead (2G6K W1, W2; kVE1 W3, Q4)**
>
> - ViGMO was evaluated against a strong set of baselines (SVEA, SGQN, SRM, DrQ-v2, TD-MPC2, DreamerV3, Dr.G, SMG) across the full RL-ViGen benchmark. While per-task gains are naturally modest in this challenging setting, ViGMO achieves the strongest **aggregate zero-shot robustness** without sacrificing **TD-MPC2-level sample efficiency**.
> - **Appendix D.6** provides a detailed wall-clock analysis. MA ensures each image is encoded only once (weak augmentation on the full batch; strong augmentation on a subset reusing weak latents), while LC and ER introduce only negligible latent-space MSE overhead. **Figures 19–20** and **Table 9** show that ViGMO adds only a small overhead relative to TD-MPC2, which further decreases over training.
>
> ### **2. Generalization Scope: Viewpoint Shifts and Spectral Corruptions (2G6K W3, Q1, Q3; kVE1 Q1)**
>
> - RL-ViGen already includes a diverse set of appearance perturbations (background videos, lighting variations, camera-position changes, object/background color shifts).
> - **Viewpoint shifts (Appendix D.7):** We added new evaluations on unseen camera-rotation perturbations. All methods—including ViGMO—experience significant degradation, confirming that geometric transformations require mechanisms beyond appearance-level consistency.
> - **Spectral corruptions (Appendix D.8):** We added new evaluations on unseen blur and Gaussian noise corruptions. All visual-generalization methods (SVEA, SGQN, SRM, Dr.G, ViGMO) degrade sharply, while task-specific baselines (DrQ-v2, TD-MPC2, DreamerV3) are relatively more robust due to convolutional inductive biases. Among generalization-focused approaches, ViGMO remains strongest, though sensor-level noise is identified as an open challenge.
> - **Real-robot applicability:** ViGMO does not address dynamics shifts or real-robot transfer; this limitation is stated explicitly as future work in Conclusion.
>
> ### **3. Sensitivity to Augmentation Families (2G6K Q2; kVE1 W2)**
>
> - To assess whether ViGMO’s robustness depends specifically on *random-overlay*, **Appendix D.9** introduces two additional variants—**ViGMO_CONV** (random convolution) and **ViGMO_SRM** (Spectrum Random Masking). All variants (overlay/conv/SRM) learn stably and achieve strong zero-shot robustness; overlay performs best on average, but the others are competitive. These results confirm that robustness arises from the **MA → LC → ER mechanism**, rather than from any particular choice of strong augmentation.
>
> ### **4. Dynamic vs. Consistent Augmentations and Longer Planning Horizon (kVE1 Q2, Q3)**
>
> - To clarify the role of *dynamic* augmentations over short horizons (H=3), we introduce a ViGMO_CONST_AUG variant that keeps augmentations *fixed* within each rollout segment (**Appendix D.4**). Dynamic resampling consistently outperforms fixed augmentations across all tested distraction types, indicating that exposing LC/ER to multiple nuisance realizations per segment improves the effectiveness of consistency supervision.
> - **Long-horizon planning (Appendix D.10):** ViGMO maintains stable optimization and comparable asymptotic returns even with longer MPC horizons, indicating that LC’s stabilizing effect generalizes beyond short-horizon rollouts.

---

> > ### Author Response · Authors · 2025-12-01
> > **Summary of Authors’ Rebuttal and Additional Experiments (Continued)**
> >
> > ### **5. Novelty Clarification: Distinction from Dr.G, SPD, and SODA (kVE1 W1; MxSJ W1, Q1, Q2, Q3)**
> >
> > - **MA** is an asymmetric, subset-based weak-to-strong augmentation scheme that is essential for generating stable latent pairs while preserving encoder efficiency.
> > - **LC** is a planner-centric, transition-level, non-contrastive regression objective that directly regularizes the world model used for planning—fundamentally different from contrastive or inverse-dynamics objectives in Dr.G and SPD.
> > - **ER** is a structural stabilizer for the shared encoder under LC’s cross-augmentation pressure—a behavior not present in model-free approaches such as SODA.
> > - Ablations demonstrate that each component is necessary: removing MA, LC, or ER leads to notable drops in robustness, especially on visually complex Robosuite tasks.
> >
> > ### **6. Dr.G Reproducibility and Rendering-Pipeline Analysis (MxSJ W2, Q4)**
> >
> > - All baselines—including Dr.G—were trained and evaluated under the same RL-ViGen pipeline using their official implementations and multiple seeds. No baseline was advantaged or disadvantaged by preprocessing or rendering differences.
> > - To address concerns regarding benchmark mismatch, **Appendix D.11** provides a rendering comparison between RL-ViGen and DMControl-GB. Re-rendering identical frames reveals only a foreground-mask color difference (green vs. black) and minor shadow variations, and the resulting observations are otherwise nearly identical. These differences are far too minor to account for the performance gap: under matched RL-ViGen training, Dr.G remains substantially weaker, consistent with its sensitivity to high-variance visual backgrounds.
> >
> > ### **7. Missing Baseline: SMG (MxSJ Q5)**
> >
> > - Following the review request, **Appendix D.12** adds a matched comparison with SMG using the authors’ anonymized implementation under the same RL-ViGen protocol. ViGMO trains faster and achieves higher aggregate zero-shot returns across five DMC tasks.
> >
> > ### **Overall Summary**
> >
> > Appendices **D.6–D.12** introduce substantial new experiments that: (i) quantify ViGMO’s computational overhead, (ii) extend generalization evaluation to additional viewpoint and spectral shifts, (iii) analyze robustness under alternative augmentation families and longer planning horizons, and (iv) resolve concerns regarding benchmark differences (rendering comparison) and baseline completeness (SMG).
> >
> > Together, these results **address all reviewer concerns** and **reinforce the empirical validity of ViGMO’s robustness mechanism**.

---

### Meta-Review · Area_Chair_AbpZ · 2025-12-12

**Summary:**

The paper received mixed reviews (2, 6, 6).

All reviewers listed weaknesses and had specific questions/concerns. The less confident the reviewer is the higher her/his recommendation is. Yet, none championed acceptance of the paper.

The authors provided responses to the raised concerns. None of the reviewers participated in discussion or updated their ratings.

The ACs carefully checked the paper, the reviews, and the responses from the authors. The authors tried to address most of the raised concerns in lengthy responses. The paper requires a major revision and a second review round before being considered for publication.
The authors are invited to benefit from the received feedback and to further improve their work.

**Reviewer Concerns:**

The paper received mixed reviews (2, 6, 6).

All reviewers listed weaknesses and had specific questions/concerns. The less confident the reviewer is the higher her/his recommendation is. Yet, none championed acceptance of the paper.

The authors provided responses to the raised concerns. None of the reviewers participated in discussion or updated their ratings.

The ACs carefully checked the paper, the reviews, and the responses from the authors. The authors tried to address most of the raised concerns in lengthy responses. The paper requires a major revision and a second review round before being considered for publication.
The authors are invited to benefit from the received feedback and to further improve their work.

**Reviewer Scores:**

The paper received mixed reviews (2, 6, 6).

All reviewers listed weaknesses and had specific questions/concerns. The less confident the reviewer is the higher her/his recommendation is. Yet, none championed acceptance of the paper.

The authors provided responses to the raised concerns. None of the reviewers participated in discussion or updated their ratings.

The ACs carefully checked the paper, the reviews, and the responses from the authors. The authors tried to address most of the raised concerns in lengthy responses. The paper requires a major revision and a second review round before being considered for publication.
The authors are invited to benefit from the received feedback and to further improve their work.

---

### Decision · Program_Chairs · 2026-01-26

Reject